# Controlling opioid receptor functional selectivity by targeting distinct subpockets of the orthosteric site

Rajendra Uprety[1†], Tao Che[2,3,4†], Saheem A Zaidi[5†], Steven G Grinnell[6], Balázs R Varga[3,4], Abdelfattah Faouzi[3,4], Samuel T Slocum[2], Abdullah Allaoa[1], András Varadi[1], Melissa Nelson[6], Sarah M Bernhard[3], Elizaveta Kulko[6], Valerie Le Rouzic[1], Shainnel O Eans[7], Chloe A Simons[7], Amanda Hunkele[1], Joan Subrath[1], Ying Xian Pan[1,8‡], Jonathan A Javitch[6], Jay P McLaughlin[7], Bryan L Roth[2]*, Gavril W Pasternak[1§], Vsevolod Katritch[5]*, Susruta Majumdar[1,3,4]*

[1]Department of Neurology and Molecular Pharmacology, Memorial Sloan Kettering Cancer Center, New York, United States; [2]Department of Pharmacology, University of North Carolina, Chapel Hill, United States; [3]Center for Clinical Pharmacology, St. Louis College of Pharmacy and Washington University School of Medicine, St. Louis, United States; [4]Department of Anesthesiology, Washington University in St. Louis School of Medicine, St. Louis, United States; [5]Department of Quantitative and Computational Biology, Department of Chemistry, Bridge Institute, Michelson Center for Convergent Bioscience, University of Southern California, Los Angeles, United States; [6]Division of Molecular Therapeutics, New York State Psychiatric Institute and Departments of Psychiatry, Pharmacology, Columbia University Vagelos College of Physicians & Surgeons, New York, United States; [7]Department of Pharmacodynamics, University of Florida, Gainesville, United States; [8]Department of Anesthesiology, Rutgers New Jersey Medical School, New Jersey, Newark, United States

*For correspondence:
bryan_roth@med.unc.edu (BLR);
katritch@usc.edu (VK);
susrutam@email.wustl.edu (SM)

†These authors contributed equally to this work

Present address: ‡Department of Anesthesiology,, Rutgers New Jersey Medical School, New Jersey , Newark, United States

§Deceased

**Abstract** Controlling receptor functional selectivity profiles for opioid receptors is a promising approach for discovering safer analgesics; however, the structural determinants conferring functional selectivity are not well understood. Here, we used crystal structures of opioid receptors, including the recently solved active state kappa opioid complex with **MP1104**, to rationally design novel mixed mu (MOR) and kappa (KOR) opioid receptor agonists with reduced arrestin signaling. Analysis of structure-activity relationships for new **MP1104** analogs points to a region between transmembrane 5 (TM5) and extracellular loop (ECL2) as key for modulation of arrestin recruitment to both MOR and KOR. The lead compounds, **MP1207 and MP1208,** displayed MOR/KOR Gi-partial agonism with diminished arrestin signaling, showed efficient analgesia with attenuated liabilities, including respiratory depression and conditioned place preference and aversion in mice. The findings validate a novel structure-inspired paradigm for achieving beneficial in vivo profiles for analgesia through different mechanisms that include bias, partial agonism, and dual MOR/KOR agonism.

## Introduction

Pain affects almost every person at some point in their lives, and it has been estimated that more than 25 million people in the United States suffer daily from severe pain (*Nahin, 2015*). Drugs targeting MOR are effective analgesics, but they retain high addiction potential and cause potentially

lethal side effects including respiratory depression. As the use of opioid analgesics increased and then came under greater restrictions, so too has their diversion, misuse, and switch to illicit opioids, as ~80% of opioid addicts reported initiating their habit through prescription opioids. The epidemic of opioid abuse caused more than 47,600 deaths in 2017 alone, (*Overdose Death Rates, 2019*) making drug overdose the leading cause of accidental death in the US. As effective analgesics are essential to minimize the pain and suffering of many diseases, identification of safer analgesic molecular entities with diminished side effects and abuse potential is critical to breaking the vicious cycle fueling the opioid epidemic.

Biased signaling is an important concept in G protein-coupled receptor (GPCR) functional mechanisms, by which distinct downstream signaling pathways can be preferentially activated by agonists working through the same receptor (*Law et al., 2013*; *Pradhan et al., 2012*). It has been proposed that opioid ligands showing a preference (bias) for activating only G protein-mediated signal transduction pathways, or against recruiting βarrestin-2, will demonstrate diminished side effects (*Raehal and Bohn, 2011*; *Chiang et al., 2016*; *Majumdar and Devi, 2018*; *Faouzi et al., 2020a*). However, studies on G protein biased opioid ligands have shown mixed results so far. The first biased ligand, a MOR biased agonist oliceridine (TRV130) (*DeWire et al., 2013*) has been recently approved by the FDA (*FDA Approves New Opioid for Intravenous Use in Hospitals, 2020*). It is important to note, however, that TRV130 displays weak G-protein bias in vitro (*Schmid et al., 2017*) and mixed safety results in rodent models (*Austin Zamarripa et al., 2018*; *Altarifi et al., 2017*). Other MOR ligands with greater bias, such as SR17018 (*Schmid et al., 2017*), show diminished respiratory depression in rodents compared to fentanyl (*Gillis et al., 2020a*), while PZM21 (*Manglik et al., 2016*; *Kudla et al., 2019*) and mitragynine(s) (*Váradi et al., 2016*; *Kruegel et al., 2016*; *Kruegel et al., 2019*; *Chakraborty and Majumdar, 2020*) display reduced abuse liability (*Yue et al., 2018*; *Hemby et al., 2019*). Similarly, some KOR-selective G protein biased ligands such as HS666 (*Spetea et al., 2017*), 6'GNTI (*Rives et al., 2012*), and triazole 1.1 (*Brust et al., 2016*) show a promising separation of place aversion from analgesia, unlike balanced KOR agonists. On the other hand, other KOR ligands such as RB64 (*White et al., 2015*), HS665 (*Spetea et al., 2017*), and collybolide (*Gupta et al., 2016*) retain the aversive properties of balanced KOR agonists despite being G protein biased.

Together, these data suggest that the 'classical model' of bias or activation of a single opioid receptor subtype may not be sufficient for achieving an optimal pharmacological profile in vivo. Here, we tested the new hypothesis that dually selective agonist ligands working through the G protein pathway at both MOR and KOR could be synergistically analgesic while mitigating the common liabilities of conventional opioids. Prior evidence shows that simultaneous activation of MOR and KOR may produce synergistic analgesia, while the contrasting side-effects offset respective liabilities (*Sutters et al., 1990*; *Pan, 1998*). For example, KOR agonists such as U50,488h, while showing no sign of respiratory depression on their own, *Matthes et al., 1998* have been reported to reduce the respiratory depression mediated by *icv* administration of DAMGO (*Dosaka-Akita et al., 1993*; *Haji and Takeda, 2001*). Likewise, nalbuphine, a moderate efficacy MOR partial agonist and high efficacy KOR partial agonist (*Schmidt et al., 1985*; *Peng et al., 2007*), is similar in analgesic efficacy to morphine, but shows negligible respiratory effects (*Schmidt et al., 1985*), suggesting that partial agonism coupled to mixed agonism at MOR/KOR may attenuate MOR-induced respiratory depression.

In this study, we used structure-based computational modeling to facilitate the design of compounds with G protein biased activity at both MOR and KOR. Comparative structural analysis and docking to opioid receptors, including the recent nanobody-stabilized active state KOR complexed with **MP1104** (*Che et al., 2018*; *Váradi et al., 2015a*), suggested that specific receptor-ligand interactions at the TM5-ECL2 region in the orthosteric ligand pocket may reduce arrestin recruitment at both *MOR* and *KOR* (*Figure 1A*). In vitro characterization and structure-activity relationship studies (SAR) of a variety of new morphinan and fentanyl analogs reported here (*Figure 1B*), strongly support this hypothesis. In vivo studies of the most potent ligands with dual selectivity to MOR and KOR, partial agonism, and reduced arrestin recruitment, **MP1207/MP1208**, also show receptor-mediated analgesic actions in mice while negating classical side-effects of opioids, suggesting a new approach toward generating effective analgesics with attenuated opioid-induced adverse effects.

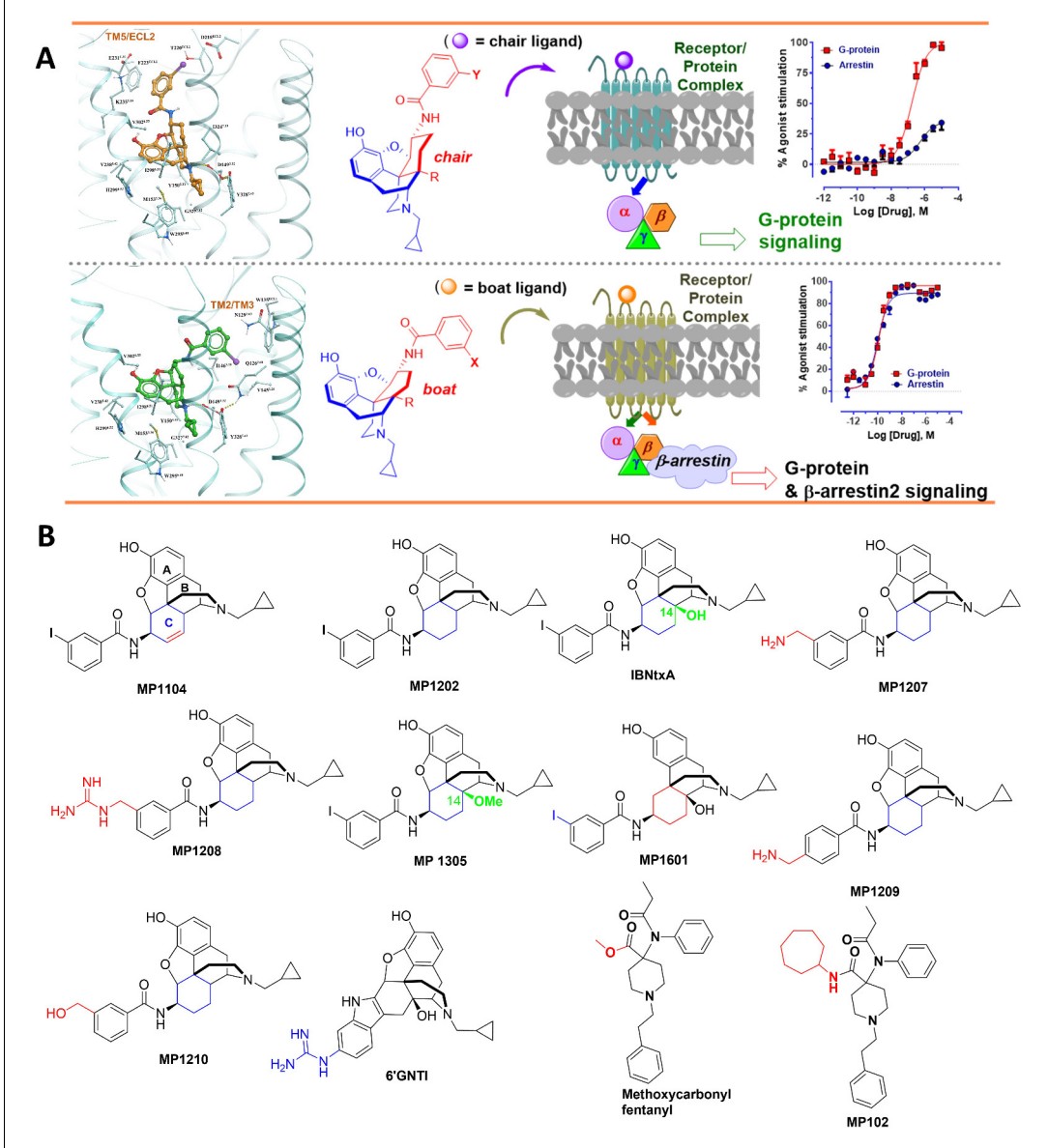

**Figure 1.** The design concept, structures of ligands synthesized and evaluated at opioids receptors. (A) Overview of the key hypothesis, suggesting that TM5-ECL2 engagement by morphinan ligand with ring-C chair form leads to preferred G protein signaling, whereas TM2-TM3 engagement and ring-C boat form leads to balanced G protein and arrestin signaling. (B) Structures of the studied m and p-substituted arylamidoepoxymorphinans (MP1104, MP1202, IBNtxA, MP1305, MP1207, MP1208, MP1209, MP1210), m-iodoarylamidomorphinan, (MP1601), methoxycarbonylfentanyl, methoxycarbonyl fentanyl amide MP102 and 6'GNTI.

## Results

### MP1104 shows arrestin recruitment at both MOR and KOR subtypes

Previously, we used the amidoepoxymorphinan ligand **MP1104** (*Váradi et al., 2015a*) to obtain the human KOR crystal structure in a nanobody-stabilized active state conformation (PDB code 6B73). Functional studies (cAMP and $\beta$-arrestin2 recruitment assays) showed that **MP1104** robustly recruits $\beta$-arrestin2 at mu and kappa opioid receptors (*Che et al., 2018*). Its bias factors at hMOR and hKOR were determined to be 0.58 (*Figure 2A–B* and *Appendix 1—figure 4B*, *Appendix 1—table 10*) and 0.15 (*Figure 2C–D* and *Appendix 1—figure 4A*, *Appendix 1—table 10*), respectively. Bias factors reported throughout this work quantify preferred activation of a signaling pathway (G protein or

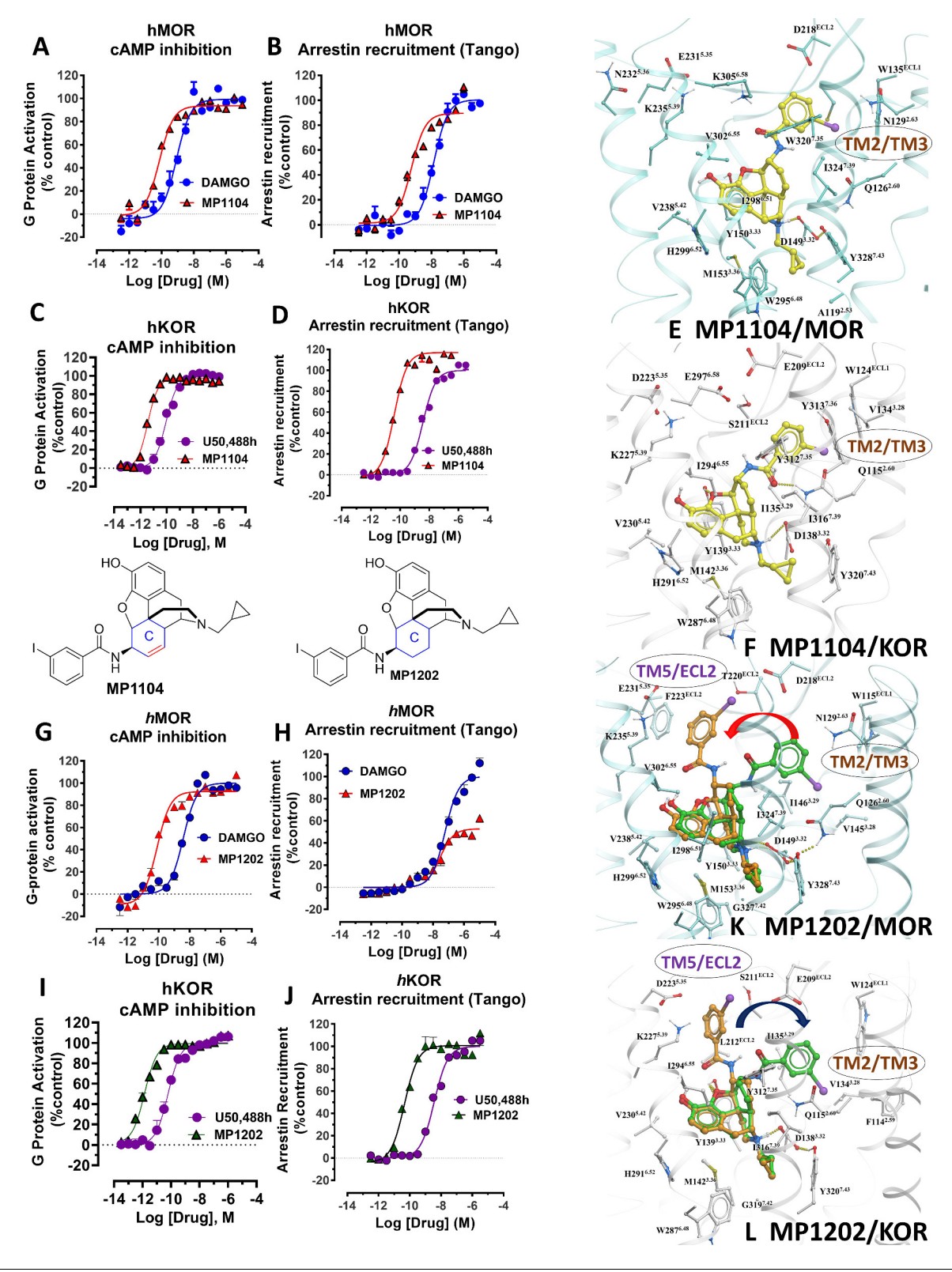

**Figure 2.** MP1104 at both MOR and KOR and MP1202 at KOR targets the TM2-TM3 region while MP1202 at MOR targets the TM5-ECL2 region and show distinct signaling properties. (A-B) MP1104 (red) is a full agonist in hMOR in cAMP inhibition (N = 5) and Tango-arrestin recruitment assays (N = 5) compared to DAMGO (blue). (C–D) MP1104 (red) is a full agonist in hKOR in cAMP inhibition (N = 3) and Tango-arrestin recruitment assays (N = 3) compared to U50,488H (purple). (E–F) The preferred docking pose of MP1104 (boat form, yellow stick) at MOR and MP1104 (boat form, yellow stick) in

*Figure 2 continued on next page*

*Figure 2 continued*

the active state of KOR. Ring-C of MP1104 in boat form forces the iodophenyl moiety to reside in a region between TM2-TM3 at MOR/KOR. (**G–H**) MP1202 (red) is a full agonist in hMOR in cAMP inhibition (N = 3) and partial agonist in Tango-arrestin recruitment assays (N = 3) compared to DAMGO (blue). (**I–J**) MP1202 (green) is a full agonist in hKOR in cAMP inhibition (N = 3) and in Tango-arrestin recruitment assays (N = 3) compared to U50,488h (purple). (**K–L**) The docking poses of MP1202 (chair form, brown stick) and (boat form, green stick) at an active states of MOR and KOR are shown. At MOR, the saturated ring C in MP1202 leads to interaction of the ligand in the ECL2 and TM5 region leading to a preference of chair form shown by a red arrow. At KOR, MP1202 behaves similar to MP1104 and the flip of ring C conformation from chair to boat is shown by a blue arrow. See *Appendix 1—table 7* for values, *Appendix 1—figure 4* and *Appendix 1—table 10* for bias calculations.

βarrestin-2) by the ligand as compared to the prototypic balanced agonists, DAMGO and U50,488h, for MOR and KOR, respectively, A bias factor of >1 signifies compounds with G protein bias, a bias factor of <1 signifies arrestin bias, while compounds with bias factor lacking statistically significant deviation from one are deemed unbiased. The bias factor for each ligand was determined using functional data obtained from cAMP inhibition vs Tango assays or BRET assays, when applicable, by following previously reported methods of calculation (*Kenakin et al., 2012*).

Here, we evaluated the potency and efficacy at G protein activation as well as arrestin recruitment of **MP1104** using BRET assays which afford a more precise interrogation of transducer bias (*Kruegel et al., 2016*) in HEK-293T cells transfected with rodent opioids receptors (mMOR and rKOR). Confirming the cAMP inhibition assay and $\beta$-arrestin2 recruitment results obtained for human opioid receptors, **MP1104** was found to be arrestin biased at both mMOR (*Appendix 1—figure 1A* and *4C*, *Appendix 1—table 11*) as well as rKOR (*Appendix 1—figure 2A* and *4C*, *Appendix 1—table 11*) with bias factors of 0.11 and 0.23, respectively.

## Synthesis and evaluation of MP1202: a MOR G biased and KOR agonist recruiting arrestin

In contrast to **MP1104**, the closely related ligand **IBNtxA** (*Majumdar et al., 2011a*; *Figure 1B*) with a hydroxyl (C14-OH) group and a saturated ring C, showed reduced $\beta$-arrestin2 recruitment at hMOR while recruiting $\beta$-arrestin2 through hKOR activity (*Appendix 1—figure 3A-D* and *4A-B* and *Appendix 1—table 10*, bias factor = 0.1 at hKOR and 24 at hMOR) (*Che et al., 2018*). At rodent receptors, we presently confirmed that IBNtxA trended toward reduced potency in the arrestin pathway versus the G protein pathway compared to DAMGO. Interestingly, while we were not able to determine its potency in the arrestin pathway its efficacy ($E_{max}$ = 75%) was greater than seen in the G protein pathway ($E_{max}$ = 59%) at mMOR (*Appendix 1—figure 3F-G* and *4C*, *Appendix 1—table 11*) and IBNtxA showed no bias at rKOR (*Appendix 1—figure 3H-I* and *4C*, *Appendix 1—table 11*). As a continuation of SAR studies on 6β-amidoepoxymorphinans to identify the structural features responsible for $\beta$-arrestin2 recruitment over G protein activation, we here synthesized **MP1202** (C14-H instead of C14-OH in IBNtxA) with the saturated cyclohexyl ring C (see synthesis in *Appendix 1—scheme 1A*). Evaluation of **MP1202** in radioligand binding assays in opioid receptor transfected cell lines showed that it retained pan opioid sub-nM binding (*Appendix 1—table 1*) and sub-nM potency in the G-protein pathway in GTPγS assays (*Appendix 1—table 2*). At human opioid receptors, **MP1202** was similar to **IBNtxA** and was found to be G protein biased at hMOR (bias factor = 31, *Figure 2G–H*, *Appendix 1—figure 4B* and *Appendix 1—table 10*) while showing $\beta$-arrestin2 recruitment and arrestin bias at hKOR (*Figure 2I–J*, *Appendix 1—figure 4A*, and *Appendix 1—table 10*).

At rodent receptors in BRET assays, **MP1202** retained $\beta$-arrestin2 recruitment at rKOR (*Appendix 1—figure 2C* and *4C* and *Appendix 1—table 11*), although showing a diminished $\beta$-arrestin2 potency at mMOR relative to DAMGO. Similar to IBNtxA a greater efficacy was seen in the arrestin pathway ($E_{max}$ = 57%) compared to the G protein pathway ($E_{max}$ = 70%) (*Appendix 1—figure 1C*).

## Predicted engagement of the TM5-ECL2 region promotes G protein bias

Differences in the bias profiles of **MP1104**, **IBNtxA**, and **MP1202** suggested distinct interaction modes of these ligands at MOR and KOR. We relied on both the ligand-based and the receptor-based structure design approaches to understand the observed pharmacology. In terms of ligand structure, the three ligands have two structural variations among them: the presence/absence of

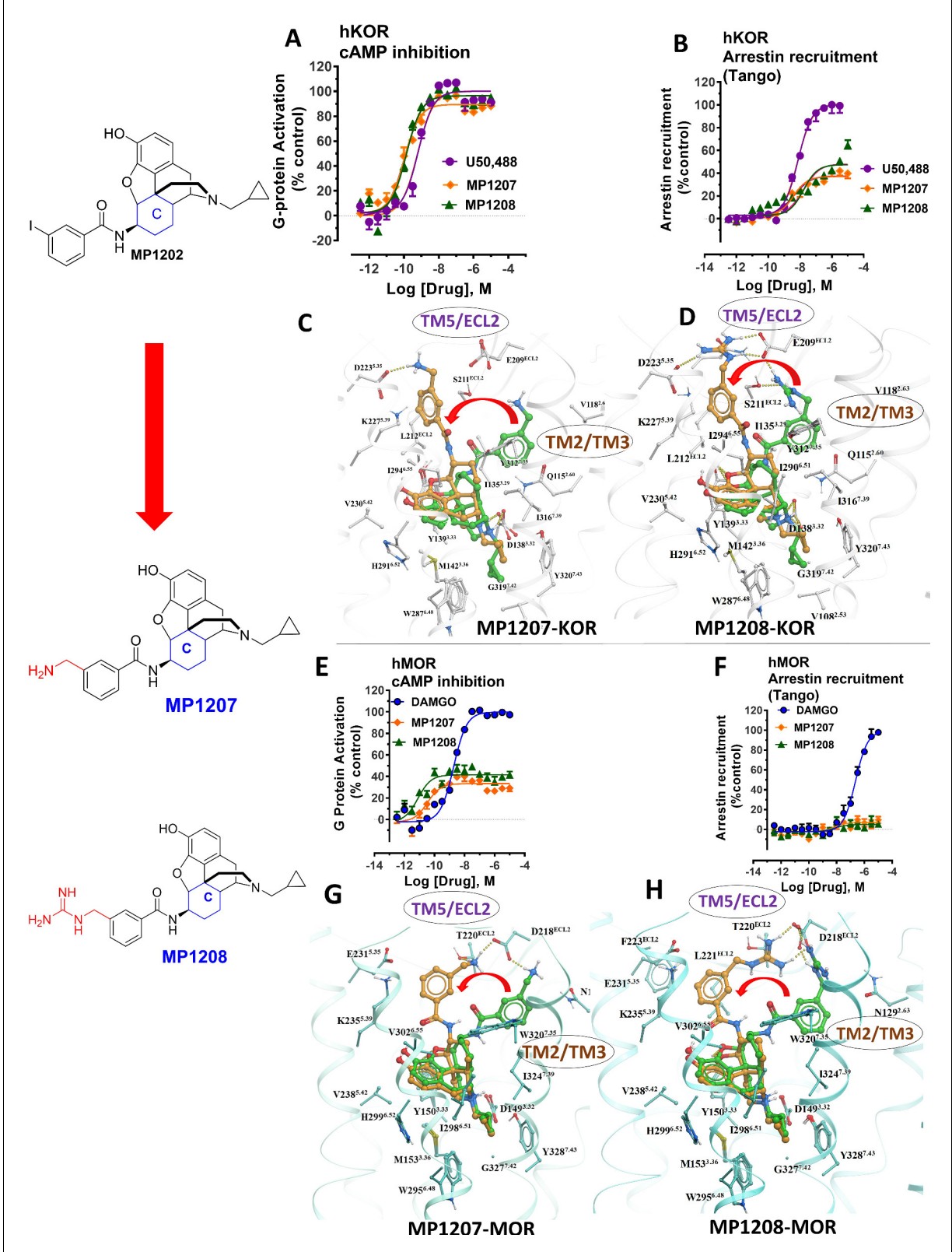

**Figure 3.** meta-Amino (MP1207) and meta-guanidino (MP1208) analogs prefer the chair conformation and target the TM5-ECL2 region and are G protein biased agonists at KOR and at MOR show no measurable arrestin recruitment. (A-B) MP1207 (orange) and MP1208 (green) are full agonists at hKOR in cAMP inhibition (N = 3) and partial agonists in Tango-arrestin recruitment assays (N = 3) compared to U50, 488 (purple). (C–D) Docking results showed that amino methyl (MP1207) or guanidino (MP1208) moieties (replacing the iodo group in MP1202) forced these compounds in chair form

*Figure 3 continued on next page*

*Figure 3 continued*

preferred confirmation at *h*KOR (chair form in brown stick and boat form in green stick). Unlike boat MP1202, chair MP1207 at *h*KOR may form a new salt bridge interaction between amino group and D223[5.35] and E209[ECL2] pulling amidophenyl moiety away from the hydrophobic pocket between TM2 and TM3 (C). Likewise, chair MP1208 forms salt bridge interactions between guanidino group and D223[5.35] as well as with E209[ECL2] (D) The flip in conformation of ringC from boat to chair for both MP1207 and MP1208 is shown by a red arrow. (E–F) MP1207 (orange) and MP1208 (green) are partial agonists at hMOR in cAMP inhibition (N = 3). No arrestin recruitment was observed for both agonists in Tango-arrestin recruitment assays (N = 3) compared to DAMGO (blue). (G–H) At *h*MOR chair forms of MP1207 and MP1208 introduce additional interactions between amino or guanidino group and D218[ECL2] and T220[ECL2]. Thus, biased signaling events of MP1207 and MP1208 are dictated from chair favored binding via the involvement of their m-amino or m-guanidino group with TM5 -ECL2 region. See *Appendix 1—table 7* for values, *Appendix 1—figure 4* and *Appendix 1—table 10* for bias calculations.

C14-OH and the saturation/unsaturation of ring C (*Figure 1B*). The similarity of the bias profiles for **IBNtxA** and **MP1202** suggested that C14-OH is not important for bias. However, the contrast between the strong preference of **IBNtxA** and **MP1202** for the G protein pathway compared to **MP1104** at MOR suggested a useful SAR between the conformation of the C ring and ligand function. Indeed, we found a strong preference for the unsaturated ring C of **MP1104** to be in the boat conformation, based on analysis of similar ligands in the Crystallography Open Database and our quantum mechanics (QM) energy calculations. At the same time, the saturated C rings of both **IBNtxA** and **MP1202** could assume both chair and boat conformations with only a modest preference for chair conformation (*Appendix 1—tables 3–4*). To analyze the differences between the boat and chair conformations of the C rings in the context of ligand-receptor interactions, we performed energy-based docking studies for these ligands in all-atom models of the receptors, based on recently solved active-state crystal structures for MOR (*Huang et al., 2015*) and KOR (*Che et al., 2018*). Due to the boat-form restriction for the unsaturated ring C, the amidophenyl moiety of the best-scored docking poses for **MP1104** in both MOR and KOR occupied a sub-pocket between TM2 and TM3 (*Figure 2E and F*). In the absence of such conformational limitation for the saturated ring C in **IBNtxA** and **MP1202**, both chair and boat conformations of the ring were observed among the top 10 poses ranked by energy score. However, we found that in MOR the best-scored **IBNtxA** (*Appendix 1—figure 3E*) and **MP1202** (*Figure 2K*) docking poses consistently adopted the chair conformation of ring C, while in KOR the best-scored poses adopted the boat conformation (*Appendix 1—figure 3J* and *Figure 2L*).

These reproducible differences between KOR and MOR can be explained by different physical properties of their TM2/TM3 sub-pockets, which accommodate the hydrophobic amidophenyl arm of these ligands. This sub-pocket is more hydrophobic in KOR because of the presence of the non-conserved V118[2.63] residue and a conformational change in the conserved Q115[2.60] residue. Although the MOR sub-pocket does share some hydrophobic residues, namely W[ECL2], V[3.28] and I[3.29], exposed polar groups of the N129[2.63] and Q126[2.60] side chains increase the polarity. Therefore, while in KOR the hydrophobic amidophenyl arm of the ligand retains its preference for binding the TM2-TM3 sub-pocket, in MOR it is preferentially redirected toward the TM5-ECL2 region. Apparently, this binding interaction preference of the amidophenyl arm of **IBNtxA** and **MP1202** is reflected in the switching of the ring-C from the chair conformation when bound in MOR to the boat conformation when interacting with KOR. Notably, this concerted switch of ring-C conformation and the amidophenyl 'arm' position correlates with the observed signaling bias. Specifically, whereas the chair conformation of **IBNtxA** and **MP1202** in MOR results in the 'arm' interactions with the TM5-ECL2 region and favors G protein activation, the boat conformation in KOR results in TM2-TM3 sub-pocket interactions and enhanced preference for balanced agonism or the arrestin pathway. This observation is further corroborated by the activation of the arrestin pathway in both MOR and KOR found for the boat-restricted conformation of the **MP1104** ligand.

## Structure-inspired design of MP1207 and MP1208 as dual MOR/KOR agonists showing reduced arrestin recruitment

Based on the above analysis, we hypothesized that a structure-inspired design of **MP1202** analogs that are G protein biased not only at MOR, but also at KOR would require a switch in preference of amidophenyl arm substituents from the TM2-TM3 sub-pocket to the TM5-ECL2 region in both receptors. To test this hypothesis, we proposed to redesign the **MP1202** ligand by introducing a

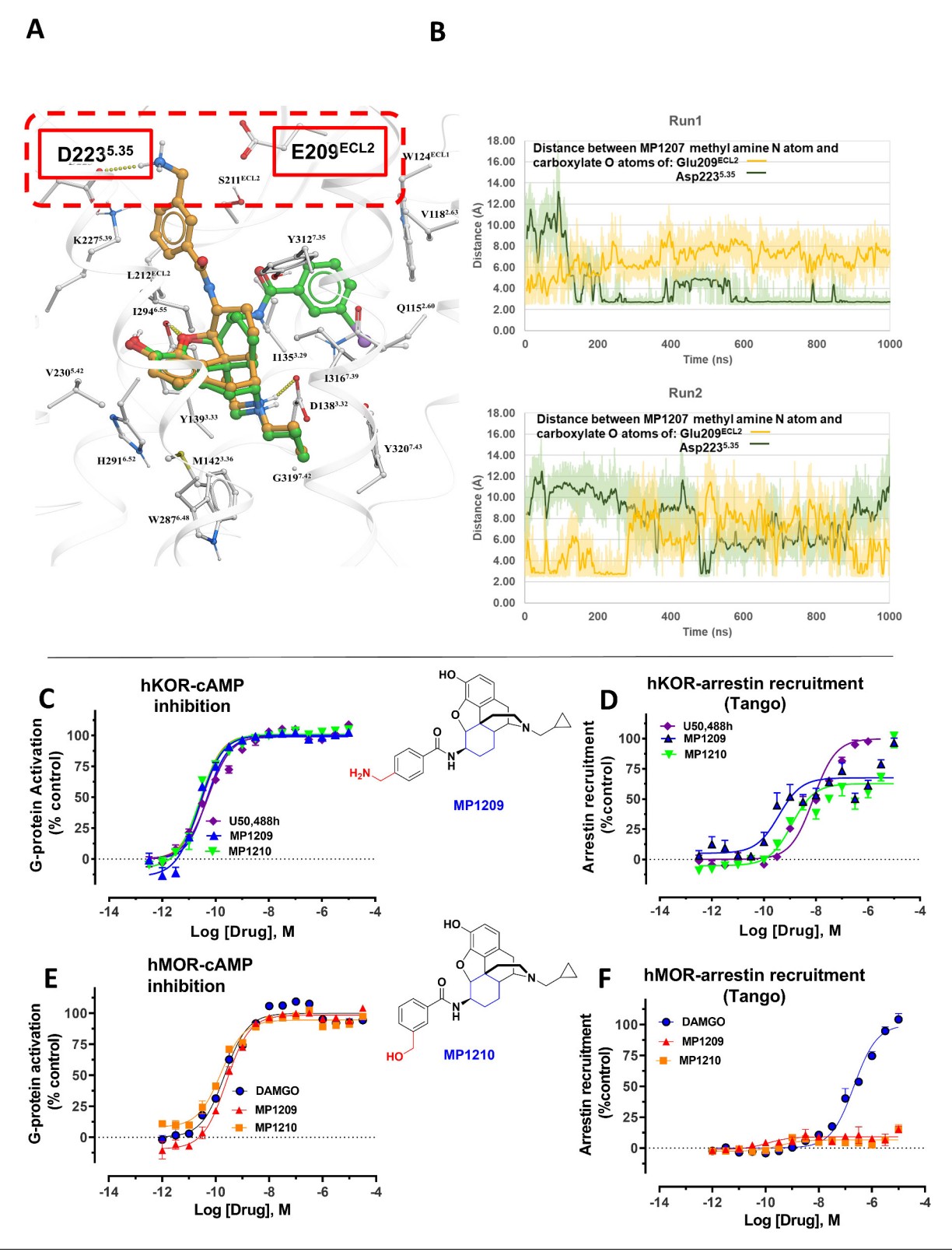

**Figure 4.** MD simulations in hKOR with MP1207 show the guanidine group interacts with E209/D223 and analogs of MP1207 not oriented toward TM5-ECL2 region suggest G protein bias of MP1207/08 is dependent on salt-bridge formation in this region. (A) Docking of MP1202 (green sticks) and MP1207 (yellow sticks) in wild type KOR showing MP1207 chair form engaging D223 and E209 residues in TM5-ECL2 region while MP1202 boat form not engaging this region. (B) Graph plotting distances between methyl nitrogen atom of MP1207 and side-chain carboxylate atoms of Glu209[ECL2] and

*Figure 4 continued on next page*

*Figure 4 continued*

Asp223$^{5.35}$ during two 1000 ns long MDs runs of MOR-MP1207 complex. Distances of each trajectory frame and running average are plotted for Glu209$^{ECL2}$ (light orange and orange) and Asp223$^{5.35}$ (light green and green). (**C–D**) MP1209 (blue) and MP1210 (light green) are full agonists at hKOR in cAMP inhibition (N = 3) and Tango-arrestin recruitment assays (N = 3) compared to U50,488h (purple). (**E–F**) Similarly, MP1209 (red) and MP1210 (orange) are partial agonists at hMOR in cAMP inhibition (N = 3). No arrestin recruitment was observed for both agonists in Tango-arrestin recruitment assays (N = 3) compared to DAMGO (blue). Regioselectivity of ring substituent is important for kappa bias. The p-methyl amino and m-methyl alcohol do not form salt-bridge within TM5-ECL2 unlike the m-methyl amino group of MP1207 as a result similar to MP1202 with respect to bias at KOR and lower arrestin signaling at MOR. See *Appendix 1—table 7* for values, *Appendix 1—figure 4* and *Appendix 1—table 10* for bias calculations.

polar or charged moiety at the amidophenyl 'arm' to make its interactions at the hydrophobic TM2-TM3 pocket of KOR unfavorable. Further, since the TM5-ECL2 region in KOR is lined by acidic residues D223$^{5.35}$ and E209$^{ECL2}$, the presence of basic moieties at the amidophenyl arm would be expected to provide additional favorable interactions to shift its preference toward the TM5-ECL2 region. Interestingly, in the docking pose for 6'GNTI in KOR, a known G protein biased ligand (*Rives et al., 2012*; *Schmid et al., 2013*), the guanidino group also occupies the TM5-ECL2 region (*Appendix 1—figure 5*).

As a part of the computer-assisted design, we proposed a library of analogs where the *m*-iodo group was swapped with polar groups such as OH, $NH_2$ $NMe_2$, $(CH_2)_nNH_2$ and $(CH_2)_n$-guanidine (n = 1–5 for amine and guanidine modification). Docking of these derivatives into the active state KOR structural model allowed computational predictions of their binding scores and conformational preferences (*Appendix 1—table 5*). Two analogs, calculated to have the greatest preference for the ring C chair form and interactions with the TM5-ECL2 region at both MOR and KOR MP1207 and MP1208 (*Figures 1B*, *3C–D and G–H*) were selected and synthesized (*Appendix 1—scheme 1B*).

Functional studies using cAMP inhibition and Tango assays at human opioid receptors showed that both **MP1207** and **MP1208** are G protein biased agonists at hKOR with bias factors of 8 and 22 (*Figure 3A–B*, *Appendix 1—figure 4A* and *Appendix 1—table 10*), respectively. At hMOR, recruitment of arrestin was completely abolished for both ligands; however, reduction of cAMP inhibition was also observed, with $E_{max}$ values of = 33% for **MP1207** and 42% for **MP1208** compared to DAMGO (*Figure 3E–F*). In the cAMP assay, the prototypic MOR agonist morphine acted as a full agonist ($E_{max}$ = 97%) whereas buprenorphine was a partial agonist ($E_{max}$ = 75%) compared to DAMGO (*Appendix 1—table 7*). In binding assays, both **MP1207** (*m*-amine group) and **MP1208** (*m*-guanidine group) showed similar affinities at mMOR ($K_i$ = 0.23 nM and 0.34 nM) and mKOR ($K_i$ = 0.39 and 0.28 nM) with substantial selectivity over mDOR ($K_i$ = 15.62 and 19.28 nM) (*Appendix 1—table 1*). In [$^{35}$S]GTPγS binding assays both **MP1207** and **MP1208** were partial agonists at both mKOR and mMOR, still maintaining very high potency at mKOR ($EC_{50}$ = 1.5 and 1.4 nM) and at mMOR ($EC_{50}$ = 1.3 and 1.1 nM). At mDOR both **MP1207** and **MP1208** show only weak efficacy (*Appendix 1—tables 2 and 6*). Similar results of partial agonism at mMOR (*Appendix 1—figure 1D*) and rKOR were seen in BRET assays (*Appendix 1—figure 2D*). The controls morphine ($E_{max}$ = 110%) and buprenorphine (59%) were found to produce full and partial agonism at mMOR in the same assay, respectively (*Appendix 1—table 7*). No recruitment of βarrestin-2 was seen with either **MP1207** or **MP1208** when rodent opioid receptors were used in BRET assays. The arrestin recruitment signal for both **MP1207** and **MP1208** at hMOR, mMOR and rKOR were too low for the bias factors at these receptors to be calculated.

To investigate specificity, **MP1207** was counter-screened for agonism against 330 other GPCRs using a Presto-Tango assay. Activity at 3 µM was observed at some other targets; however, no potent activity was confirmed with a full concentration–response experiment at these targets (*Appendix 1—figure 6*). Since this assay relies on arrestin recruitment and our ligand was found to show limited *β*-arrestin2 recruitment at its primary targets, MOR and KOR, we further counter-screened **MP1207** using radioligand binding assays. **MP1207** had a $K_i$ >10 µM affinity at all screened targets except SERT and $α_2$C-adrenoreceptors, which displayed $K_i$ ($pK_i±$ SEM) of 356 (6.45 ± 0.091)nM and 2979 (5.53 ± 0.16) nM respectively (*Besnard et al., 2012*). At MOR, KOR, and DOR the respective values were 0.39 (9.4 ± 0.042), 0.39 (9.4 ± 0.056), and 60.1 (7.22 ± 0.059) showing a 900-fold selectivity for MOR and KOR over the nearest non-opioid target.

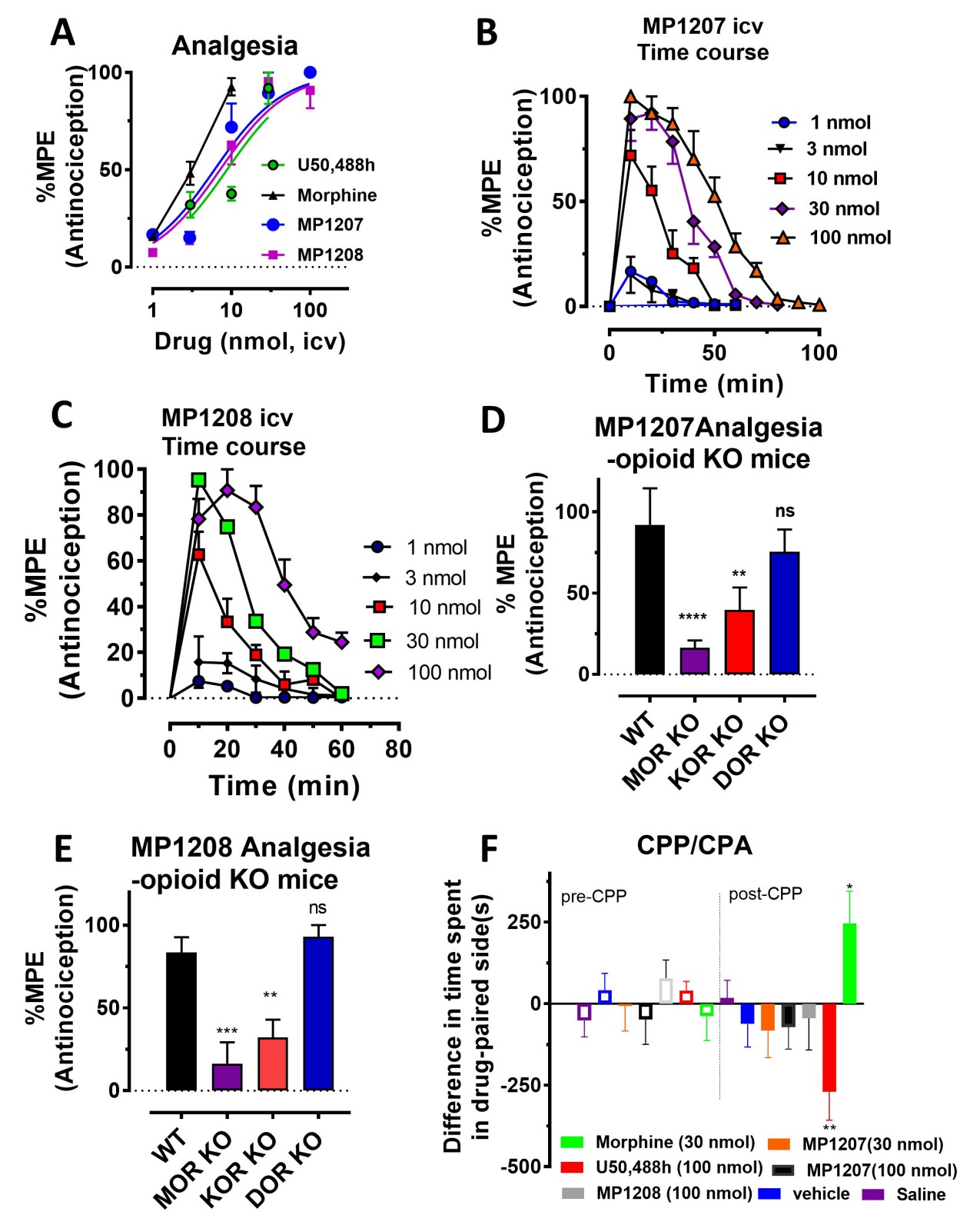

**Figure 5.** MP1207/1208 show MOR/KOR-mediated antinociception without showing place preference or aversion. (**A**) Dose-response curve: Groups of C57BL/6J mice were supraspinally (*icv*) administered MP1207, MP1208, morphine and U50,488h and antinociception measured using the 55°C tail withdrawal assay at peak effect. Data are shown as mean % antinociception (MPE) ± SEM. $ED_{50}$ (with 95% CI) values = 6.1 (4.1–8.9) nmol, 7.2 (5.0–10.2) nmol, 4.77 (1.49–28.8) nmol, and 8.8 (5.7–13.5) nmol were calculated for MP1207, MP1208, morphine and U50,488h respectively. (**B–C**) Antinociceptive

*Figure 5 continued on next page*

Figure 5 continued

time course: Effect of MP1207 (**B**) and MP1208 (**C**) at doses of 1, 3, 10, 30, and 100 (*n* = 8 each group, with *n* = 16 for MP1208 at 30 nmol) with repeated measures over time. (**D–E**) MP1207/08 analgesia in KO mice: Analgesic effect of MP1207 (30 nmol *icv*, **D**) and MP1208 (35 nmol icv, **E**) was evaluated in groups (n = 8) of WT (C57BL/6J), MOR KO, KOR KO, and DOR KO mice. Antinociception of MP1207/08 remained intact in DOR KO mice while it was attenuated in MOR KO and KOR KO mice. Results for MP1207 were analyzed with one-way ANOVA followed by Dunnett's post hoc test; $F_{3,28}$=10.11, p=0.0001.**p=0.005 relative to WT, ****p<0.0001 relative to WT, ns = p>0.05 relative to WT. Similarly, MP1208 results were analyzed with one-way ANOVA followed by Dunnett's post hoc test, $F_{3,28}$=15.35, p<0.0001.**p=0.005 relative to WT, ****p<0.0001 relative to WT, ns = p>0.05 relative to WT. All values are expressed as the mean MPE ± SEM. (**F**) Conditioned place preference or aversion (CPP/CPA): Place conditioning evaluation of MP1207, 1208, morphine, U50,488h, saline and vehicle in C57BL/6J mice after *icv* administration. Following determination of initial preconditioning preferences (pre-CPP), mice were place-conditioned daily for 2 days with MP1207 (30 nmol, n = 23 and 100 nmol, *n* = 24), MP1208 (100 nmol, *n* = 26), U50,488 (100 nmol, *n* = 19) or morphine (30 nmol, *n* = 12) and examined the fourth day for final conditioned place preference (post-CPP). Mean differences in time spent on the drug-paired side ± SEM are presented. *p=0.03 (morphine) or **p=0.003 (U50,488h), significantly different from matching pre-conditioning preference; unpaired t-test with Welsch's correction. Both MP1207/08 were neither reinforcing nor aversive in mice.

## Molecular dynamics and additional MP1207 analog design support TM5-ECL2 region role in signaling bias

**MP1207** and **MP1208** were designed to target the TM5-ECL2 region of KOR via their amidophenyl arm accessing additional interactions with polar residues D223[5.35] and E209[ECL2] (See *Figure 3C–D*), capitalizing on the structure-based modeling predictions. We carried out MD simulations on **MP1207** in KOR (*Figure 4A–B*). Over 50% of the cumulative trajectory frames from the two 1000 ns long MD runs had distances between carboxylate oxygens of these acidic residues within 4.5 Å of methyl amine nitrogen atom of **MP1207**. The methyl amine of **MP1207** can form salt bridge interactions with both E209[ECL2] and D223[5.35] residues, and during MD simulations a fluctuating switch between these two residues was observed. The MD simulations indicate that these salt bridges are possible and moderately stable. These results support our key design hypothesis which posits an interaction between the basic amino and guanidine moieties of **MP1207** and **MP1208** with the negatively charged D223[5.35] and E209[ECL2] side chains in TM5-ECL2 region, and an important role of these interactions in reducing preference for the arrestin pathway and/or recruitment.

Conversely, the design hypothesis also suggests that the relocation of the amidophenyl arm can be achieved by hampering its interactions in the TM2/TM3 subpocket. Indeed, we had previously shown that a mutation of Y312W[7.35] in the TM2-TM3 region of KOR reduces βarrestin-2 recruitment for IBNtxA, which has a similar amidophenyl arm (*Che et al., 2018*). While the residues in this position do not make direct contact with the receptor in our models, this mutation is likely to impact binding indirectly (See *Appendix 1—figure 7I* showing Bu72-MOR TM2-TM3 and *Appendix 1—figure 7J* showing **MP1202**-KOR TM2-TM3 region interactions).

In the present study, we tested the effects of the Y312W[7.35] mutation on **MP1202**, **MP1207** and **MP1208**. As expected, this mutation in KOR reduced arrestin recruitment by **MP1202** to the level observed for MOR, resulting in strong G protein bias (bias factor = 34, *Appendix 1—figure 7A&C and L*) as opposed to robust β-arrestin2 recruitment in the wild-type KOR (*Appendix 1—figure 7B and D*). A similar trend was found with **MP1207** and **1208**, revealing a further reduction of arrestin recruitment at the mutant KOR receptor (*Appendix 1—figure 7E-G*), down to the level seen in wild type MOR (*Appendix 1—figure 7H*). These results suggest that similar to IBNtxA, the Y312W[7.35] mutation in KOR changes the conformational character of the sub-pocket to a MOR-like shape, possibly by changing the conformation of Q115[2.60], and also leads to loss of water-mediated hydrogen bonding with the amido group. Therefore, Y312W[7.35] provides an environment that favors the chair conformation, leading to a shift in ligand bias toward G protein activity (*Appendix 1—figure 7K*).

To further evaluate the role of salt bridges in facilitating ligand conformations with the amidophenyl arm in the TM5-ECL2 region, we synthesized two **MP1207** analogs by swapping the *m*-$CH_2NH_2$ with *p*-$CH_2NH_2$ (**MP1209**) and *m*-$CH_2OH$ (**MP1210**) (*Figure 1B*, *Appendix 1—scheme 2*). Consistent with our predictions (*Appendix 1—table 4*), the *para*-substituted and more planar analog, **MP1209** and the *meta* substituted methyl hydroxyl analog (**MP1210**) which are incapable of forming a salt bridge with D223[5.35] and E209[ECL2], lost their G protein bias in hKOR and showed no bias (*Figure 4C–D*, *Appendix 1—figure 4A* and *Appendix 1—table 10*) while retaining hMOR null arrestin recruitment (*Figure 4E–F*). Thus, only when ideal orientation/distances are maintained (i.e. *meta*-amino/guano) and the amidophenyl arm is accommodated in the TM5-ECL2 region of KOR, is

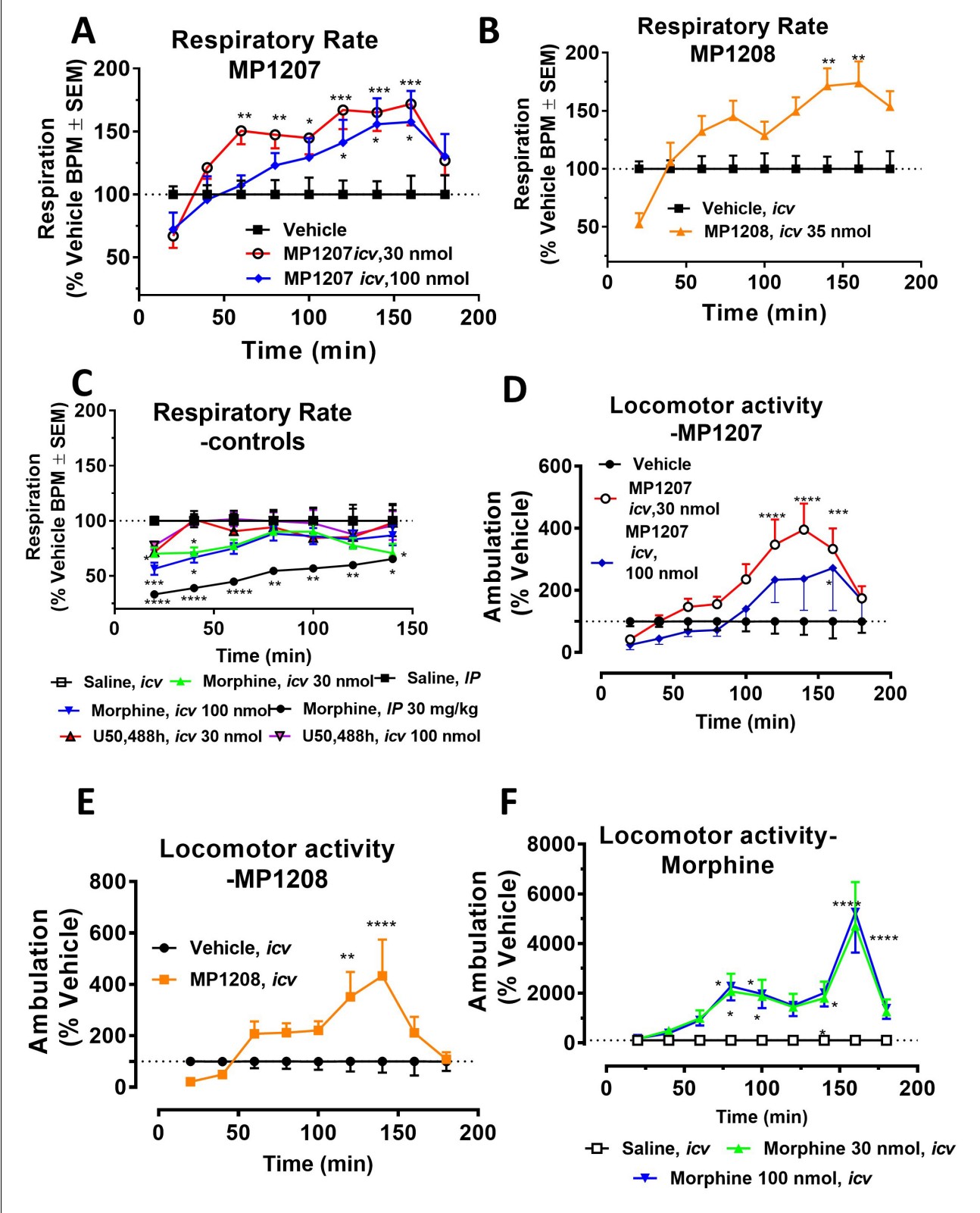

**Figure 6.** MP1207/08 show attenuated respiratory depression and locomotor effects compared to morphine in mice. Mice were administered either saline (n = 15), vehicle (n = 24), morphine (30 mg/kg, *IP; n = 12* or 30 nmol; n = 18 or 100 nmol, *icv*; n = 16), MP1207 (30 nmol; n = 26 or 100 nmol *icv*, n = 10), MP1208 (100 nmol *icv*, n = 10) and the breaths (**A–C**) or ambulations (**D–F**) measured every minute and averaged in 20 min segments. Data presented as % vehicle response ±.SEM; **A–C**, breaths per minute (BPM) or (**D–F**): ambulation (XAMB). (**A**) MP1207 (30 nmol *icv*) increased breathing
*Figure 6 continued on next page*

**Figure 6 continued**

rates at 60 min (\*\*p=0.005), 80 min (\*\*p=0.008), 100 min (\*p=0.01), 120 min (\*\*\*p=0.0001), 140 min (\*\*\*p=0.0002), and 160 min (\*\*\*p=0.0002) compared to vehicle. MP1207 (100 nmol *icv*) showed increased respiration at 120–160 min (\*p=0.02) as determined by two-way ANOVA followed by Dunnett's multiple-comparison test. (B) MP1208 (35 nmol icv) increased respiratory rate similar to MP1207 and significantly different from vehicle at 140 min (\*\*p=0.0018) and 160 min (\*\*p=0.0028) as determined by two-way ANOVA followed by Sidak's multiple comparison test. (C) Morphine (30 mg/kg, *IP*) depressed respiration compared to saline, *IP* at 20–60 min (\*\*\*\*p<0.0001), 80 min (\*\*p=0.0011), 100 min (\*\*p=0.0021), and 120 min (\*p=0.02). Respiration after U50,488h (30 or 100 nmol *icv*) did not significantly differ from that of saline, *icv*. Morphine (30 nmol, *icv*) showed respiratory depression at 20 min (\*p=0.03), 40 min (\*p=0.04) while morphine (100 nmol, *icv*) showed respiratory depression at 20 min (\*\*\*p=0.0009) and 40 min (\*p=0.02) compared to saline, *icv*. (D–F) Locomotor effect: (D) MP1207 (30 nmol, *icv*) significantly increased forward ambulations at 120–140 min (\*\*\*\*p<0.0001) and 160 min (\*\*\*p=0.01) and at 160 min (\*p=0.01), whereas the 100 nmol, *icv* dose did not as determined by two-way ANOVA followed by Dunnett's multiple-comparison test in comparsion to vehicle. (E) MP1208 (35 nmol icv) increased ambulatory activity similar to MP1207 but less than morphine, and significantly different from vehicle at 120 min (\*\*p=0.002) and 140 min (\*\*\*\*p=0.0001) as determined by two-way ANOVA followed by Sidak's multiple comparison test. (F) Morphine at 30 nmol *icv* showed significant hyperlocomotion (note axis scale) compared to saline at 80 min (\*p=0.02), 100 min (\*p=0.05), and 160 min (\*\*\*\*p<0.0001). Similarly morphine at 100 nmol was significantly different at 80 min (\*p=0.01), 100 min (\*p=0.0395), 140 min (\*p=0.034) and 160 min (\*\*\*\*p<0.0001).

a bias for G protein activity observed. Taken together, the described MD analysis combined with assessment of epoxymorphinan analogs targeting in the TM5-ECL2 region further corroborated our hypothesis that interactions in this region can be critical for G protein bias at KOR.

## Design of other morphinan and fentanyl based analogs engaging TM5 support its role in signaling bias

To further explore the hypothesis that TM5 interactions promote G protein bias for scaffolds beyond epoxymorphinans we next examined the fentanyl template. We have published a series of methoxyl-carbonyl fentanyl amides previously (*Váradi et al., 2015b*). We docked a published compound **MP102** at MOR and found that the cycloheptyl group is predicted to line up in the TM5 region, a region not accessed by methoxylcarbonyl fentanyl (*Appendix 1—figure 8E*). We evaluated G protein/arrestin activity of both compounds and found that as expected, methoxylcarbonyl fentanyl showed strong arrestin recruitment and arrestin bias (bias factor = 0.09 at hMOR, *Appendix 1—figure 8A-B, C-D and 8* and *Appendix 1—table 10*) while **MP102** targeting TM5 was unable to recruit arrestin at human (*Appendix 1—figure 8F-G*) and mouse receptors (*Appendix 1—figure 8H-I*).

Continuing with epoxymorphinan/morphinan SAR, *m*-iodo analogs **MP1305** (methylated OH as C14-OCH$_3$), and **MP1601** from morphinan template (devoid of the etheral bridge linking rings A and C) were synthesized (*Appendix 1—schemes 3* and *4*). Our computational docking studies suggested that ring C of **MP1305** prefers the chair form at both MOR and KOR (See *Appendix 1—figures 9I, K* for details), while **MP1601** favors the chair form at MOR and boat form at KOR (*Appendix 1—figure 9J,L*). Consistent with our docking predictions, **MP1305** was found to be G protein biased at MOR and KOR at human receptors with a bias factor of 11 and 4, respectively (*Appendix 1—figure 9A-D* and *4A-B* and *Appendix 1—table 10*) while **MP1601** behaved similar to **IBNtxA** and **MP1202** and showed a preference for the arrestin pathway at hKOR (bias factor = 0.5, *Appendix 1—figure 9E-F* and *4A* and *Appendix 1—table 10*) while being G protein biased at hMOR (Bias factor = 4, *Appendix 1—figure 9G-H* and *4B* and *Appendix 1—table 10*).

At rodent receptors, both compounds showed sub-nM affinity and high potencies in [$^{35}$S]GTPγS assays (*Appendix 1—tables 1–2*). Due to very low arrestin signal, it was impossible to calculate the bias factor at rodent receptors from the BRET assays at MOR for both **MP1305** and **1601**, and for **MP1601** at KOR (*Appendix 1—figures 1E, 2E*). **MP1601** was not biased at rKOR, but did demonstrate arrestin bias at hKOR (*Appendix 1—figures 1–2E* and *4C* and *Appendix 1—table 11*). These results are again consistent with our hypothesis correlating G protein bias (and reduced recruitment of arrestin) of the morphinan derivatives with their C-ring chair conformation and the resulting positioning of the amidophenyl arm in the TM5-ECL2 region. At the same time, the abolished arrestin recruitment could also be a consequence of partial agonism (see discussion, below).

## In vivo pharmacology of MP1207 and MP1208

Antinociception of **MP1207** and **MP1208** was evaluated in vivo in mice using a standard 55°C warm-water tail withdrawal assay, with the compounds administered supraspinally (*icv*) since both compounds were not expected to show systemic activity when administered *IP* because of being positively charged at physiological pH. The antinociceptive $ED_{50}$ (and 95% CI) values of **MP1207** and **MP1208** were 6.1 (4.1–8.9) nmol (*Figure 5A*), and 7.2 (5.0–10.2), nmol respectively, comparable to that of the kappa agonist U50,488h, 8.8 (5.7–13.5) nmol, although slightly higher than the $ED_{50}$ of morphine, 4.77 (1.49–28.8) nmol, *icv*. Both drugs showed antinociceptive responses that peaked at 10 min, returning to baseline values 90 min later for **MP1207** (*Figure 5B*) and 60 min for **MP1208** (*Figure 5C*).

MP1207 and MP1208 were characterized in detail, studying opioid receptor selectivity and opioid mediated potential adverse effects. We used transgenic knock out (KO) mice lacking MOR, KOR, or DOR to examine the selectivity of **MP1207/08's** analgesic actions (*Figure 5D–E*). **MP1207/08** antinociception was found to be significantly attenuated in both MOR KO and KOR KO mice, but remained intact in DOR KO mice. The results were in line with the >40 fold selectivity of MOR and KOR over DOR in our binding assays. Both **MP1207 and MP1208** failed to show either rewarding or aversive behavior in mice in a conditioned place preference paradigm at the highest dose that could be tested given their solubility (100 nmol, *icv*,~15 x analgesic $ED_{50}$ dose) (*Figure 5F*). In contrast, morphine (tested at ~5 x the analgesic $ED_{50}$ dose) and U50,488h (~15 x analgesic $ED_{50}$ dose) as expected showed place preference and place aversion, respectively (*Figure 5F*).

MP1207 and **MP1208** were further tested for respiratory effects (*Figure 6A* for **MP1207** and **6B** for **MP1208**). As expected, administration of morphine *icv* or *IP* decreased respiratory rate while the KOR agonist U50,488h (*icv*) was without effect (*Figure 6C*). In contrast, **MP1207** at 30 nmol *icv* and **MP1208** at 35 nmol *icv* (doses selected to be 5x the $ED_{50}$ dose) each stimulated respiration. At a higher dose of 100 nmol *icv*, **MP1207** still stimulated respiration, although the effect was less than at the lower dose of 30 nmol (*Figure 6A*). Finally, we examined locomotor activity in the same mice. In WT mice, MP1207 (*Figure 6D*) and MP1208 (*Figure 6E*) stimulated locomotor activity at the lower dose of 30 nmol (*icv*) compared to 100 nmol (*icv*). As expected, morphine induced hyperlocomotion (*Figure 6F*), and this hyperlocomotor effect was significantly greater than seen with either **MP1207** or **MP1208** at equianalgesic doses (*Appendix 1—figure 11*). 'Together, these results support that dual MOR and KOR agonism may offset the liabilities characteristic of receptor-selective agonists'.

Overall, these results demonstrate that **MP1207 and MP1208** produce potent antinociception predominantly mediated by KOR and MOR, yet shows a separation of analgesia from some classic opioid side effects such as respiratory depression, conditioned place preference, and aversion, in contrast with the canonical mu and kappa-opioid receptor selective agonists, morphine and U50,488h.

## Discussion

This study employs a new structure-based concept for controlling the functional profile of opioid ligands which allows design of biased ligands at KOR and MOR. Agonists so generated show efficient analgesia in vivo, and lack the respiratory depression and aversion/reward liabilities of classical opioid analgesics. Over the last ~15 years, the discovery of G protein biased opioid ligands has been widely considered as a strategy for the development of potent but safer opioid analgesics. In spite of TRV130's clinical approval, the results pointing to the ability of MOR-specific G protein biased ligands to alleviate opioid side effects has recently been challenged (*Kliewer et al., 2019*; *Hill et al., 2018*; *Faouzi et al., 2020b*). Among the most important recent findings is that the respiratory depressant effects of morphine appear to be $\beta$-arrestin 2-independent. Moreover, mice possessing mutations in the MOR C-tail that prevent phosphorylation by GRK and greatly impair recruitment of $\beta$-arrestin two retained opioid-induced respiratory depression, constipation, and withdrawal effects. These results contrast with previous data from $\beta$-arrestin 2 KO mice (*Raehal et al., 2005*). Consistent with past results, tolerance was attenuated, and the analgesic duration of action was prolonged in these mutant mice (*Kliewer et al., 2020*; *Bachmutsky et al., 2020*; *Kliewer et al., 2019*). Similarly, putative biased ligands such as PZM21, TRV130 and SR17018 (*Gillis et al., 2020a*) have recently been reported to have low intrinsic efficacy at G protein signaling when evaluated in

assays without receptor reserve, raising the idea that partial agonism, and not arrestin bias, may be critical to the design of improved drugs.

It is clear that ligands with more precisely tuned selectivity and functional profiles are needed to more definitively interrogate the pharmacological mechanisms for insulating opioid analgesia from their notorious side effects. Using recently solved active-state structure of KOR in complex with **MP1104** and computational modeling studies of close analogs **MP1202** and **IBNtxA**, we identified two key sites in the binding pockets of both MOR and KOR: (1) a primarily hydrophobic sub-pocket between TM2-TM3 and (2) a region between TM5-ECL2 lined with acidic residues. We also showed that boat or chair conformations of ring C in the **MP1104** scaffold can control the switch of the rigid amidophenyl arm between these two sites. Most importantly, the predicted interactions of the ligand amidophenyl arm in the TM2-TM3 sub-pocket correlated with either $\beta$-arrestin2 bias or unbiased signaling, while the switch to theTM5-ECL2 site correlated with G protein biased agonism in the opioid receptors. To test the applicability of this observation, we designed **MP1207** and **MP1208** with basic moieties that are predicted to facilitate interactions with acidic residues in TM5-ECL2. While the ligands retained high-affinity binding and G protein-mediated signaling, they showed dramatically either reduced potency and/or recruitment at arrestin pathway for both MOR and KOR, thus providing support for our rational design strategy for G protein biased agonists. MD simulations coupled with synthesis of other **MP1207** analogs (**MP1209/1210**-polar and uncharged) that did not engage this region further corroborated the critical role of that region in the G-biased signaling of **MP1207/08** at KOR. Notably, our design strategy also led to partial agonism for G protein signaling at both MOR and KOR.

Interestingly, the TM5-ECL2 key role in bias is also in line with the docking pose for a known biased kappa ligand 6'GNTI, which has its guanidine group align within the TM5-ECL2 region similar to the amidophenyl arm of our compounds. The TM5-ECL2 region has also been proposed as a region dictating bias at other GPCRs such as 5HT$_{2B}$ serotonin (**McCorvy et al., 2018**) and D$_2$ dopamine (**Chun et al., 2018**) receptors, although the specific mechanisms may differ between these receptors.

Interpretation of bias analysis in vitro has its limitations, as discussed recently (**Luttrell et al., 2015**; **Gundry et al., 2017**). For most compounds, lack of measurable arrestin signal in one assay (e.g. BRET) usually was corroborated by strong G protein bias measured with another more amplified assay (e.g. Tango or in different species). For some others, the absence of measurable arrestin recruitment precluded calculation of bias factor in both the BRET and Tango assays. In human KOR Tango assays, dose-dependent curves showed a biphasic shape indicating a second wave of signaling at high concentrations of drugs (see **Figure 3B**, MP1208, **Figure 4D**, MP1209/10 and **Appendix 1—figure 7C**, **MP1202**). It is unlikely this effect was the result of non-specific interactions, as it was not observed in parallel MOR Tango assays, or with the KOR agonist U50,488H. The most plausible explanation for the observed biphasic response could be compounds at high concentrations hitting intracellular receptors, either with a basal pool of internal receptors or with receptors that have been internalized in response to agonist addition. Although further testing of this hypothesis is beyond the scope of this study, it is supported by previous studies showing that large amounts of intracellular GPCRs exist (**Che et al., 2020**; **Stoeber et al., 2018**).

It is additionally important to note that **MP1207** and **MP1208** are partial agonists for G protein signaling, but that the G protein assays that we have employed are more amplified than the arrestin assays, which can lead to apparent increases in G protein signaling efficacy. Morphine is a full agonist (E$_{max}$ = 109%) and buprenorphine has higher efficacy (E$_{max}$ = 59%) in our BRET-based amplified assays in mMOR (**Appendix 1—table 7**) compared to E$_{max}$ = 70% and 25% for morphine and buprenorphine respectively in non-amplified assays (**Gillis et al., 2020a**). Thus, we cannot rule out a critical role for partial agonism in the improved in vivo profile observed for these compounds. Indeed such an interpretation would be more in line with recent publications arguing that it is partial agonism and not bias that accounts for the improved safety profiles of low efficacy MOR agonists (**Gillis et al., 2020a**; **Kliewer et al., 2019**; **Kliewer et al., 2020**; **Gillis et al., 2020b**).

When evaluated in animal models, **MP1207** and **MP1208** demonstrated supraspinal analgesia mediated by MOR and KOR while showing attenuated abuse potential and aversion, as well as lack of respiratory depression. Surprisingly, in contrast to the conventional respiratory depression characteristic of MOR-selective agonists, a modest stimulation was observed. The present data are consistent with evidence suggesting that mixed activation of MOR and KOR may produce potent

analgesia with reduced liabilities (*Brice-Tutt et al., 2020*). Notably, U50,488h *sc and icv* not only lack respiratory depression on their own (*Matthes et al., 1998*), but also reduce DAMGO-induced respiratory depression (*Dosaka-Akita et al., 1993*), supporting a potential role of KOR agonism in alleviating respiratory depression. Moreover, a report examining co-administration of the KOR agonist nalfurafine with the MOR agonist oxycodone noted the reduction of both self-administration as well as respiratory depression, further suggesting that mixed action MOR/KOR ligands (*Townsend et al., 2017*) may have a superior safety profile over either classical or biased ligands at a single subtype, similar to the action ascribed presently to **MP1207** and **MP1208**.

Given the uncertainty over a mechanism by which KOR activity blunts MOR-mediated respiratory depression, future respiratory testing of mixed action ligands with MOR and KOR KO mice will be needed. Such testing may also better resolve whether weak MOR agonism alone or a combination MOR-KOR dual partial agonism, rather than the functional selectivity of **MP1207** and **MP1208** would be sufficient to account for the reduced liabilities presently observed. Similarly, while the present results show more than 40-fold selectivity of both **MP1207** and **MP1208** for MOR and KOR over DOR, it is plausible that pharmacology of these compounds at higher doses may also involve DOR, which is known to modulate MOR mediated behavioral measures. For instance, mixed action MOR-DOR agonists (*Lei et al., 2020*) are more effective analgesics in a chronic pain setting, while MOR agonists-DOR antagonists (*Váradi et al., 2016*) reportedly display less tolerance and physical dependence.

Extending this, buprenorphine, a MOR partial agonist with antagonism at KOR/DOR and weak agonism at NOP shows a ceiling effect in respiratory depression assays (*Grinnell et al., 2016*; *Dahan et al., 2005*) supporting the premise that favorable multifunctional pharmacology and MOR partial agonism may reduce undesired liabilities while synergistically optimizing analgesia. Admittedly, buprenorphine still displays hyperlocomotion (*Marquez et al., 2007*) and an inverted U-shaped dose-response in CPP assays (*Marquez et al., 2007*), and its pharmacology is complicated by its active metabolite norbuprenorphine, which is less active but more efficacious (*Huang et al., 2001*) and is known to show respiratory depression (*Brown et al., 2012*). These limitations point to the value of further refinement in the desired multifunctional pharmacological profile of developed compounds. Here, we find that **MP1207/08** are partial agonists for G protein signaling, and the attenuated respiratory depression and place preference and place aversion could also result, at least in part, from the lower intrinsic efficacy of these ligands at both KOR and MOR (*Gillis et al., 2020a*).

This conjunction of KOR partial agonism with MOR partial agonism may have therapeutic benefits over the more broadly active buprenorphine, for example as shown presently with our probes by blunting MOR mediated respiratory depression.

The structure-based approach in this study allowed rational design of MOR and/or KOR G protein biased ligands with reduced $\beta$-arrestin2 recruitment. Leads **MP1207** and **MP1208** display effective analgesia in vivo with reduced abuse potential and aversion, as well as a lack of respiratory depression. While the relative roles of G protein bias, reduced efficacy at G protein pathways, and the mixed MOR-KOR agonism in the improved profile of these lead compounds are not yet clear and will require further investigation, the new rational design concept and insights gained from the structure-function analysis of these derivatives will help more precise tuning of the pluridimentional functional selectivity profiles of optimal analgesics devoid of opioid liabilities.

## Materials and methods

### Drugs and materials

Opiates were provided by the Research Technology Branch of the National Institute on Drug Abuse (Rockville, MD). **IBNtxA**, **MP1104, MP102** and new compounds (**MP1202, MP1207-MP12108, MP1305**, and **MP1601**) were synthesized. [$^{125}$I]BNtxA was prepared as previously described (*Pickett et al., 2015*). Reagents Na$^{125}$I and [$^{35}$S]GTPγS were purchased from Perkin-Elmer (Waltham, MA). Selective opioid antagonists were purchased from Tocris Bioscience. Miscellaneous chemicals and buffers were purchased from Sigma-Aldrich.

## Chemistry

Reagents purchased from Sigma-Aldrich Chemicals, Fisher Scientific, Alfa Aesar; were used without further purification. While performing synthesis, reaction mixtures were purified by silica gel flash chromatography on E. Merck 230–400 mesh silica gel 60 using a Teledyne ISCO CombiFlash $R_f$ instrument with UV detection at 280 and 254 nm. RediSep $R_f$ silica gel normal phase columns were used with a gradient range of 0–10% MeOH in DCM. Reported yields are isolated yields upon purification of each intermediate. Final clean (purity $\geq$95%, UPLC) compounds were used for the study. NMR spectra were collected using Bruker Avance III 500, or Avance III 600 with DCH CryoProbe instruments. Chemical shifts are reported in parts per million (ppm) relative to residual solvent peaks at the nearest 0.01 for proton and 0.1 for carbon (CDCl$_3$ $^1$H: 7.26, $^{13}$C: 77.1; and CD$_3$OD $^1$H: 3.31, $^{13}$C: 49.0). Peak multiplicity is reported as the NMR spectra were processed with MestreNova software, namely s – singlet, d – doublet, t – triplet, q – quartet, m – multiplet for examples. Coupling constant ($J$) values are expressed in Hz. Mass spectra were obtained at the MSKCC Analytical Core Facility using The Waters Acuity SQD LC MS by electrospray (ESI) ionization. High-resolution mass spectra were obtained using a Waters Acuity Premiere XE TOF LC-MS by electrospray ionization and the accurate masses are reported for the molecular ion [M+H]$^+$. Detail experiments and characterization of the new compounds are included in the supporting information section.

## Mice

Male C57BL/6J mice (24–38 g) were purchased from Jackson Laboratories (Bar Harbor, ME). MOR KO, KOR KO, and DOR KO mice were bred in the McLaughlin laboratory at University of Florida. Progenitors of the colonies for MOR KO and KOR KO were obtained from Jackson Labs, whereas the DOR KO mice were a generous gift of Dr. Greg Scherrer. All mice used throughout the manuscript were opioid naïve. All mice were maintained on a 12 hr light/dark cycle with Purina rodent chow and water available ad libitum and housed in groups of five until testing.

## Radioligand competition binding assays

[$^{125}$I]IBNtxA binding was carried out in membranes prepared from Chinese Hamster Ovary (CHO) cells stably expressing murine clones of mMOR, mDOR, and mKOR, as previously described (Váradi et al., 2015a; Pickett et al., 2015; Váradi et al., 2013). In brief, binding was performed at 25°C for 90 min. Binding in mMOR/CHO was carried out in 50 mM potassium phosphate buffer with 5 mM MgSO$_4$ and 20 µg/mL protein while binding in mKOR/CHO and mDOR/CHO was carried out in 50 mM potassium phosphate pH = 7.0 buffer and 40 µg/mL protein. After the incubation, the reaction was filtered through glass-fiber filters (Whatman Schleicher and Schuell, Keene, NH) and washed (3 × 3 mL of ice-cold 50 mM Tris-HCl, pH 7.4) on a semiautomatic cell harvester. Nonspecific binding was determined by the addition of levallorphan (8 µM) to matching samples and was subtracted from total binding to yield specific binding. Protein concentrations were determined using the Lowry method with BSA as the standard (Lowry et al., 1951). $K_i$ values were calculated by nonlinear regression analysis in GraphPad Prism.

## [$^{35}$S]GTP$\gamma$S functional assay

[$^{35}$S]GTP$\gamma$S binding was performed on membranes prepared from transfected cells stably expressing opioid receptors in the presence and absence of the indicated compound for 60 min at 30°C in the assay buffer (50 mM Tris-HCl, pH 7.4, 3 mM MgCl$_2$, 0.2 mM EGTA, and 10 mM NaCl) containing 0.05 nM [$^{35}$S]GTP$\gamma$S; 2 µg/mL each leupeptin, pepstatin, aprotinin, and bestatin; and 30 µM GDP, as previously described (Bolan et al., 2004). After the incubation, the reactions were filtered through glass fiber filters (Whatman Schleicher and Schuell, Keene, NH) and washed (3 × 3 mL of ice-cold buffer, 50 mM Tris-HCl, pH 7.4) on a semi-automatic cell harvester. Filters were transferred into vials with 3 mL of Liquiscint (National Diagnostics, Atlanta, GA), and the radioactivity in vials was determined by scintillation spectroscopy in a Tri-Carb 2900TR counter (PerkinElmer Life and Analytical Sciences). Basal binding was determined in the presence of GDP and the absence of drug. Data was normalized to 100 nM DAMGO, DPDPE, and U50,488h for mMOR, mDOR, and mKOR binding, respectively. EC$_{50}$, IC$_{50}$, and %E$_{max}$ values were calculated by nonlinear regression analysis in GraphPad Prism.

## cAMP inhibition assay

To measure $G_{\alpha i}$-mediated cAMP inhibition, HEK 293T (ATCC CRL-11268) cells were co-transfected with human opioid receptor (hMOR/hKOR/hDOR) along with a luciferase-based cAMP biosensor (GloSensor; Promega) and assays were performed similar to previously described (*Che et al., 2018*; *Fenalti et al., 2014*). After 16 hr, transfected cells were plated into Poly-lysine coated 384-well white clear bottom cell culture plates in DMEM with 1% dialyzed FBS at a density of 15,000–20,000 cells per 40 µL per well and incubated at 37°C with 5% $CO_2$ overnight. Next day, drug solutions were prepared in freshly prepared buffer [20 mM HEPES, 1 X HBSS, 0.3% bovine serum album (BSA), pH 7.4] at 3X drug concentration. Plates were decanted and received 20 µL per well of drug buffer (20 mM HEPES, 1X HBSS, pH 7.4) followed by addition of 10 µL of drug solution (three wells per condition) for 15 min in the dark at room temperature. To stimulate endogenous cAMP via β adrenergic-Gs activation, 10 µL luciferin (4 mM, final concentration) supplemented with isoproterenol (400 nM, final concentration) were added per well. Cells were incubated in the dark at room temperature for 15 min, and luminescence intensity was quantified using a Wallac TriLux Microbeta (Perkin Elmer) luminescence counter. Results (relative luminescence units) were plotted as a function of drug concentration, normalized to $E_{max}$ of DAMGO and U50,488h for MOR and KOR respectively; and analyzed using 'log(agonist) vs. response' in GraphPad Prism.

## Tango $\beta$-arrestin recruitment assay

The Tango assays were performed as previously described (*Che et al., 2018*). HTLA cells expressing TEV fused-$\beta$-Arrestin2 were transfected with human opioid receptors (hMOR/hKOR/hDOR) Tango construct. The next day, cells were plated in DMEM supplemented with dialyzed FBS (1%) in poly-L-lysine coated 384-well white clear bottom cell culture plates at a density of 10,000–15,000 cells/well in a total of 40 µL. The cells were incubated for at least 6 hr before receiving drug stimulation. Drug solutions were prepared in drug buffer (20 mM HEPES, 1×HBSS, 0.3% BSA, pH 7.4) at 3X and added to cells (20 µL per well) for overnight incubation. The same drug solutions were used for the Tango and cAMP assays. The next day, media and drug solutions were removed and 20 µL per well of BrightGlo reagent (Promega, with 1:20 dilution) was added. The plate was incubated for 20 min at room temperature in the dark before counting using a luminescence counter. Results (relative luminescence units) were plotted as a function of drug concentration, normalized to $E_{max}$ of DAMGO and U50,488h for hMOR and hKOR respectively, and analyzed using 'log(agonist) vs. response' in GraphPad Prism.

## Bioluminescence resonance energy transfer (BRET) assay

The BRET assays were performed by following the protocol published previously (*Kruegel et al., 2016*). In brief, the following cDNA amounts were transfected into HEK-293T cells ($5 \times 10^6$ cells/plate) in 10 cm dishes using polyethylenimine (PEI) in a 1:1 ratio (diluted in Opti-MEM, Life Technologies): for G-protein activation; 2.5 µg mMOR/mKOR/mDOR, 0.125 µg GαoBRLuc8, 6.25 µg β1, 6.25 µg mVenus-γ2; for arrestin recruitment; 2 µg mMOR/mKOR/mDOR, 0.25 µg Rluc8- arrestin3-Sp1, 5 µg mem-linker-citrine-SH3, 5 µg GRK2. Cells were maintained in HEKS44 293T media described above. The media was changed after 24 hr of the transfection and cells were dissociated and re-suspended in phosphate buffered saline (PBS) at 48 hr of transfection. Approximately 200,000 cells/well were added to a black-framed, white well 96-well plate (Perkin Elmer; Waltham, MA). The microplate was centrifuged, and the cells were resuspended in PBS. For agonist experiments, after 5 min, 5 µM of the luciferase substrate coelenterazine H was added to each well. After 5 min, ligands were added, and the BRET signal was measured 5 min later using PHERAstar FS plate reader. For antagonist competition experiments, cells were pre-incubated with the antagonist at varying concentration for 30 min. Coelenterazine H (5 µM) was then added to each well for 5 min. Following coelenterazine H incubation, a fixed concentration of the reference agonist (5x $EC_{50}$) was added, and the BRET signal was measured at 30 min using PHERAstar FS plate reader. The signal was quantified by calculating the ratio of the light emitted by the energy acceptor, mVenus (510–540 nm), or citrine (510–540 nm), over the light emitted by the energy donor, RLuc8 (485 nm). This drug-induced BRET signal was normalized to $E_{max}$ of DAMGO or U50,488h at MOR and KOR respectively. Dose response curves were fit using a three-parameter logistic equation in GraphPad Prism.

## Bias determination

Bias analyses was carried out using the method proposed by *Kenakin et al., 2012.* that is based on the Black and Leff operational method (*Black and Leff, 1983*). For this we followed the step-wise protocol described by Gomes et al recently (*Gomes et al., 2020*). Dose response curves obtained in G protein signaling and arrestin recruitment assays were normalized to that of the standards (DAMGO for MOR and U50,488h for KOR). Data points were fit to the three parameter logistic equation in Prism 7.0 to obtain maximal response ($E_{max}$), $EC_{50}$ values for all ligands for both assays. Transduction coefficients (log ($T/K_A$)) were calculated using the Black and Leff operational model. Log(T/KA) [also referred as Log (RA)] ratios for each ligand in different pathways were determined. Subtract Log(T/KA) ratio of the standard from those of the other ligands to obtain normalized coefficients ΔLog(T/KA). ΔΔLog(T/KA) was determined by subtracting ΔLog(T/KA) ratios from different pathways. The actual value of bias was calculated using anti-Log ΔΔLog(T/KA) values.

## Assessment of off-target activity of MP1207 using PRESTO-Tango GPCR-ome

To identify potential off-target activity of **MP1207**, we used the National Institutes of Mental Health Psychoactive Drug Screen Program. **MP1207** was first tested for activity against 330 non-olfactory GPCRs using the PRESTO-Tango GPCRome screening β-arrestin recruitment assay at 3 μM MP1207. The activity at each receptor was measured in quadruplicate.

Screening of compounds was accomplished using previously described methods with several modifications (*Kroeze et al., 2015*). First, HTLA cells were plated in DMEM with 2% dialyzed FBS and 10 U/mL penicillin-streptomycin. Next, the cells were transfected using an in-plate PEI method (*Longo et al., 2013*). PRESTO-Tango receptor DNAs were resuspended in OptiMEM and hybridized with PEI prior to dilution and distribution into 384-well plates and subsequent addition to cells. After overnight incubation, drugs diluted in DMEM with 1% dialyzed FBS were added to cells without replacement of the medium. The remaining steps of the PRESTO-Tango protocol were followed as previously described.

## Tail-withdrawal assay

The 55°C warm-water tail-withdrawal assay was conducted in mice as a measure of acute thermal antinociception as described previously (*Reilley et al., 2010*). Briefly, each mouse was tested for baseline tail-withdrawal latency prior to drug administration. Following drug administration, the latency for each mouse to withdraw the tail was measured every 10 min until latency returned to the baseline value. A maximum response time of 15 s was utilized to prevent tissue damage. If the mouse failed to display a tail-withdrawal response within 15 s, the tail was removed from the water and the animal was assigned a maximal antinociceptive score of 100%. Data are reported as percent antinociception, calculated by the equation: % antinociception = 100 x [(test latency - baseline latency)/ (15 - baseline latency)]. This was utilized to account for innate variability between mice. Compounds were administered either, interperitoneally (*IP*) or intracerebroventricularly (icv) and the analgesic action of compounds was assessed at as described previously (*Haley and McCORMICK, 1957*). To briefly describe icv administration: mice were anesthetized using isoflurane. A small (3 mm) incision was made in the scalp, and the drug (2 μl/mouse) was injected (using a 10 μL Hamilton syringe fitted to a 27-gauge needle) into the right lateral ventricle at the following coordinates: 2 mm caudal to bregma, 2 mm lateral to sagittal suture, and 2 mm in depth.

## Respiratory and locomotor effects

Respiration rates and spontaneous ambulation rates were monitored using the automated, computer-controlled Comprehensive Lab Animal Monitoring System (CLAMS, Columbus Instruments, Columbus, OH) as described previously (*Reilley et al., 2010*; *Cirino et al., 2019*). Awake, freely moving adult male mice (C57BL6/J wild-type, MOR KO, and KOR KO) were habituated in closed, sealed individual apparatus cages (23.5 cm x 11/5 cm x 13 cm) for 60 min before testing. A baseline for each animal was obtained over the 60-min period before drug injection, and testing began immediately post-injection. Vehicle, morphine (30 mg/kg, IP or 30 or 100 nmol, *icv*), or **MP1207** (30 or 100 nmol, *icv*) or **MP1208** (35 nmol, *icv*) were administered (icv or IP) and five min later mice were confined to the CLAMS testing cages for 200 min. Using a pressure transducer built into the sealed

CLAMS cage, the respiration rate (breaths/min) of each occupant mouse was measured. Infrared beams located in the floor measured locomotion as ambulations, from the number of sequential breaks of adjacent beams. Data are expressed as percent of vehicle control response.

## Conditioned place preference and aversion

Mice were conditioned with a counterbalanced place conditioning paradigm using similar timing as detailed previously (*Váradi et al., 2016*). Groups of C57BL/6J mice (n = 18–24) freely explored a three-compartment apparatus for 30 min. The amount of time subjects spent in each compartment was measured over the 30 min testing period. Prior to place conditioning, the animals did not demonstrate significant differences in their time spent exploring the left vs right compartments. During each of the next 2 days, mice were administered vehicle (0.9% saline) and consistently confined in a randomly assigned outer compartment for 40 min, half of each group in the right chamber, half in the left chamber. Four hours later, mice were administered drugs morphine (30 nmol, icv), U50,488h (100 nmol, icv), **MP1207** (30 and 100 nmol, icv), **MP1208** (100 nmol, icv) or vehicle and were placed to the opposite compartment for 40 min. Conditioned place preference or aversion data are presented as the difference in time spent in drug- and vehicle associated chambers.

## Molecular modeling

The receptor proteins were extracted from the RCSB server for mouse MOR (PDBID: 5c1m), and human KOR (PDBID: 6b73), representing agonist-bound active state of the receptors. All the objects except the receptor protein subunit, the crystallized ligand, and three crystallographic waters important for ligand interactions were deleted from the MOR structure, and the protein was prepared by addition and optimization of hydrogens and optimization of the side chain residues. A similar procedure was also followed for the KOR structure, with an additional step of transplanting and optimizing the three crystallographic water molecules from the active state MOR into active state KOR. Ligands were sketched, assigned formal charges and energy-optimized prior to docking. The ligand docking box for potential grid docking was defined as the whole extracellular half of the protein, and all-atom docking was performed using the energy minimized structures for all ligands with a thoroughness value of 30. The best-scored docking poses, for both chair and boat forms, were further optimized by several rounds of minimization and Monte Carlo sampling of the ligand conformation, including the surrounding side-chain residues (within 5 Å of the ligand) and the three crystallographic water moleculess in the orthosteric sites. All the above molecular modeling operations were performed in ICM-Pro v3.8–5 molecular modeling package. The DFT (B3LYP) QM calculations for boat and chair conformations of ligands were performed using Gaussian03 with two basis sets (LanL2DZ and DGDZVP) using the servers at the High-Performance Computing at the University of Southern California.

The molecular dynamics simulation setup for the **MP1207**-bound KOR (residues 51-340) model was built using CHARMM-GUI web server (*Lee et al., 2016*). The CHARMM General Force Field (*Kim et al., 2017*) was used to generate CHARMM topology and parameter files for MP1207. The ligand-bound receptor system was embedded in a lipid bilayer with a POPC/cholesterol ≈ 9:1 ratio and with an area of 80 Å × 80 Å. The system was solvated with explicit TIP3P water molecules, ionized with 0.15 M $Na^+$ cations, and neutralized with $Cl^-$ ions. The resulting simulation system had a total of 79,258 atoms and occupied an initial volume of 80 Å × 80 Å × 119 Å. The CHARMM36 force field (*Best et al., 2012*) was employed to perform all-atom MD simulations using the GROMACS software package version 2018.1 (*Abraham et al., 2015*). Following the initial energy minimization of the water boxed, lipid embedded and ionized **MP1207**-bound KOR system, six short equilibration runs were carried out while gradually decreasing harmonic constraints on lipid and protein heavy atoms for a cumulative run of 15 ns. The particle mesh Ewald algorithm was utilized to calculate long-range electrostatic interactions, and van der Waals interactions were switched off gradually between 10 Å to 12 Å. Periodic boundary conditions were applied to simulation boxes, and simulations were run with integration time step of 2 fs at 310 K. The resulting trajectories from two independent 1000ns long production runs were analyzed using in-built GROMACS analysis tools. All MD simulations and analyses were performed using the servers at the High-Performance Computing at University of Southern California.

## Chemistry - synthesis

## MP1202, MP1207, MP1208, MP1209, and MP1210

Synthesis of *m*-iodoarylamidomorphinans (MP1202) was achieved, by starting with known codeine phthalimide (*Váradi et al., 2015a*) one in six sequential steps (*Appendix 1—scheme 1A*). The codeine phthalimides one was prepared from morphine in two steps according to the literature procedures. The reduction of codeine phthalimides in the presence of Pd/C and hydrogen followed by phthalimide group removal using excess of hydrazine hydrate gave the β-dihydrocodeine amine 2 (*Crooks et al., 2006*; *Simon et al., 1994*; *Simon et al., 1992*). The β-dihydrocodeine amine two was treated to *m*-iodobenzoic acid in the presence coupling reagent HATU with an organic base DIPEA to furnish corresponding *m*-iodoarylamidomorphinan 3. The *m*-iodoarylamidomorphinan three was treated with DIAD at 65°C in acetonitrile for 20 hr followed by two equivalents of pyridine hydrochloride (Py.HCl) treatment at room temperature to obtain the *m*-iodoarylamidonormorphinan 4 (*Yuan et al., 2013*). *N*-alkylation of **4** was achieved by heating it with (bromomethyl)cyclopropane in the presence of $K_2CO_3$ in DMF to furnish **5**. Finally, *O*-demethylation in **five** was performed using standard $BBr_3$ demethylation protocol to obtain **MP1202** (*Váradi et al., 2015a*). On the other hand, **MP1207–MP1208** were prepared (*Appendix 1—scheme 1B*) using **6** (β-dihydro N-CPM morphine-amine) which was prepared form morphine in seven steps using known protocols (*Simon et al., 1994*; *Simon et al., 1992*). In addition, di-Boc-guanidinomethyl benzoic acid eight was prepared by reacting amino methyl benzoic acid with *N,N'-di*-Boc-1*H*-pyrazole-1-carboxamidine at 50°C (*Robinson and Roskamp, 1997*). Next, *m*-Boc-aminomethyl benzoic acid (*Zhang et al., 2014*) **seven** and *di*-Boc-guanidinomethyl benzoic acid eight were coupled with **6** (β-dihydro N-CPM morphine-amine) in DMF in the presence of HATU and DIPEA to obtain corresponding analogs **9–10**. Finally, deprotection of Boc group at **9–10** using TFA/DCM in the presence of triethyl silane as a cation scavenger furnished the desired compounds; *m*-aminomethyl and *m*-guanidinomethyl arylamidodihydromorphinans **MP1207–MP1208**. *Appendix 1—scheme 2* shows the synthesis of **MP1209** and **MP1210**. Briefly, six was coupled with commercially available 3-(hydroxymethyl)benzoic acid in presence of PyBOP as the coupling agent to give **MP1210**. Coupling of **6** (β-dihydro *N*-CPM morphine-amine) with p-Boc-aminomethyl benzoic acid and deprotection of Boc group gave final product, **MP1209**.

## MP1305

Synthesis of 14-*O*-methyl *m*-iodoarylamidomorphinan **MP1305** was achieved by starting with the known ketal **12** in seven sequential steps (*Appendix 1—scheme 3*). At first, phenolic and ketone groups in naloxone were protected to obtain the ketal **12** prior the methylation of sterically hindered 14-*O* position (*Nagase et al., 2006*). The ketal **12** was treated with an excess of NaH at 0°C in DMF and the mixture was heated with iodomethane at 55°C. Then, the ketal protecting group was removed by treating with aqueous HCl in methanol under mild heating conditions resulting in the known 14-*O*-methyl ketone **13** (*Kobylecki et al., 1982*). Stereoselective reduction of the ketone **13** using lithium selectride in THF at low temperature furnished corresponding **α** alcohol **14**. The stereocenter inversion at C-6 position, with the introduction of a phthalimide moiety, was achieved using DIAD and $PPh_3$ by employing standard Mitsunobu protocol. Next, phthalimide moiety was removed by treating with excess of hydrazine hydrate in methanol to obtain **β** amine **15**. Then, amine **15** was treated to *m*-iodobenzoic acid in the presence HATU and DIPEA in DMF followed by 3-*O*- demethylation of the intermediate using $BBr_3$ in DCM furnished 14-*O*-methyl *m*-iodoarylamidomorphinan **MP1305**.

## MP1601

Synthesis of *m*-iodoarylamido-4,5-deoxymorphinan **MP1601** was achieved from the known ketone **16** (*Appendix 1—scheme 4*). The ketone **16** was synthesized in five steps using naltrexone by following literature reports (*Tius and Kerr, 1992*; *Hupp and Neumeyer, 2010*; *Zhang et al., 2007*). Reductive amination of the ketone **16** using $NH_4OAc/NaCNBH_3$ in methanol gave a racemic mixture of C-6 **α/β** amines (*Majumdar et al., 2011b*). Separation of **β** amine from the **α/β** mixture while work up, was more problematic then anticipated to isolate in optimum yield. However, upon using a mixture of 10% TEA and 1% MeOH in DCM as a column solvent with a silica gel column chromatography, **β** amine **17** was isolated in modest yield. The pattern spectral information in proton NMRs of

the **α** and **β** amines are in agreement with that of close 4,5-epoxymorphinan **α/β** amines (*Jiang et al., 1977*). For instance, upon an introduction of amino moiety at C-6 position, the aromatic proton at C-4 position displays a significant downfield shift in comparison to that of ketone **15** ($\delta$ = 6.80 ppm $C_4$-$H_{Ar}$). The C-4 proton chemical shift ($\delta$) values in **β** and **α** amines are 7.01 and 6.86 ppm, which is about 0.21 and 0.06 ppm downfield shift respectively, indicating that **β** amino group poses lower effect (*Jiang et al., 1977*). Next, the **β** amine **17** was coupled with *m*-iodobenzoic acid using HATU as a coupling reagent in the presence of TEA to obtain 3-methoxy *m*-iodoarylamido-4,5-deoxymorphinan **18**. Finally, deprotection of 3-methyl group in **18** using $BBr_3$ in DCM furnished the desired *m*-iodoarylamido-4,5-deoxymorphinan **MP1601**.

## Preparation and characterization of new compounds

### (7*R*,12*bS*)−9-methoxy-3-methyl-2,3,4,4*a*,5,6,7,7*a*-octahydro-1*H*-4,12-methanobenzo furo[3,2-e]isoquinolin-7-amine (2)

The known phthalimide **1** (3 g, 7.0 mmol) was dissolved in a stirred solution of DCM (15 mL) in methanol (100 mL). Palladium catalyst (10% Pd/C, 149 mg, 0.2 eq.) was added and the mixture was hydrogenated at 50 psi. After the completion of reaction monitored by mass spectrometry, the solution was filtered through celite, concentrated under reduced pressure, and the crude product was purified by silica gel column chromatography (ISCO, 40 g column) using methanol (5–15%) in DCM to get a white solid (2.7 g; Yield 90%) of the desired phthalimide dihydro intermediate whose spectral data matched with the literature reports. Then, hydrazine hydrate (21.5 mL, 34.5 mmol, 10 eq.) was added to the stirred solution of phthalimide dihydro intermediate (1.48 g, 3.4 mmol) in dry methanol (10 mL) at once at rt and the reaction was continued overnight. The reaction mixture was diluted with DCM (40 mL) and the organic layer washed with brine (2 × 20 mL), dried over anhydrous $Na_2SO_4$, filtered and concentrated under reduced pressure. The crude product was purified by silica gel column chromatography (ISCO, 12 g column) using a mixture of methanol in ethyl acetate with small amount of ammonium hydroxide as a base (87%EtOAC/10%MeOH/3%NH₄OH) to get a white solid. Finally, the white solid was re-dissolved in EtOAc, filtered, and precipitated by petroleum ether to get (0.96 g; Yield 93%) of the desired product **2**. The spectral data of the compound 2 was matched with the literature reports (*Váradi et al., 2015a*).

### 3'-Iodo-N-((7R,12bS)−9-methoxy-3-methyl-2,3,4,4a,5,6,7,7a-octahydro-1H-4,12-methanobenzofuro[3,2-e]isoquinolin-7-yl)benzamide (3)

m-Lodobenzoic acid (1.0 g, 4.03 mmol, 1.3 eq.) was added to a stirred solution of *β*−7,8-dihydro-codeine-$NH_2$**2** (932 mg, 3.10 mmol) in DMF (10 mL) at. HATU (1.5 g, 4.03 mmol, 1.3 eq.) was added to the mixture at rt at once and after 5 min, DIPEA (1.62 mL, 9.31 mmol, 3 eq.) was added. After 20 min, the reaction mixture was diluted with EtOAc (80 mL). The EtOAc layer was washed with brine (5 × 50 mL) to remove DMF, dried over anhydrous $Na_2SO_4$, filtered, and concentrated under reduced pressure. The crude product was purified by silica gel column chromatography using a mixture of methanol (0–5%) in DCM to get desired products **3**; (977 mg; Yield 59%). [1]H NMR (600 MHz, $CDCl_3$) $\delta$ = 8.10 (t, *J* = 1.7 Hz, 1H), 7.85–7.77 (m, 1H), 7.72 (dt, *J* = 7.9, 1.3 Hz, 1H), 7.13 (t, *J* = 7.8 Hz, 1H), 6.73 (d, *J* = 8.2 Hz, 1H), 6.66 (d, *J* = 8.2 Hz, 1H), 4.56 (d, *J* = 8.0 Hz, 1H), 3.82 (s, 3H), 3.75 (dq, *J* = 12.6, 3.3 Hz, 1H), 3.15 (s, 1H), 3.02 (d, *J* = 18.3 Hz, 1H), 2.57–2.51 (m, 1H), 2.43 (s, 3H), 2.27–2.14 (m, 2H), 2.05–1.98 (m, 1H), 1.87 (s, 1H), 1.70 (ddd, *J* = 12.3, 3.8, 1.6 Hz, 1H), 1.62–1.49 (m, 1H), 1.38 (qd, *J* = 13.0, 2.5 Hz, 1H), 1.11 (m, 1H). [13]C NMR (151 MHz, $CDCl_3$) $\delta$ = 165.8, 143.9, 143.7, 140.2, 136.6, 136.0, 130.1, 126.2, 119.3, 114.1, 94.1, 92.8, 77.2, 77.0, 76.8, 59.5, 56.7, 53.1, 47.2, 43.4, 42.7, 28.7, 24.1, 20.1; HRMS calcd for $C_{25}H_{27}IN_2O_3$ $[M+H]^+$, 531.1145; found, 531.1140.

### 3'-Iodo-*N*-((7*R*,12*bS*)−9-methoxy-2,3,4,4*a*,5,6,7,7*a*-octahydro-1*H*-4,12-methanobenzofuro[3,2-e]isoquinolin-7-yl)benzamide (4)

The dihydrocodeine iodoaryl amide (952 mg, 1.795 mmol) was added to a stirred solution of DIAD (636 µL. 3.21 mmol, 8 eq.) $CH_3CN$ (15 mL) at rt under an argon and the reaction mixture was heated to 65°C for 20 hr. The reaction mixture cooled to rt and pyridine HCl (415 mg, 3.59 mmol, 2 eq.) was added and the reaction was continued for 3 days. The solvent was removed under reduced pressure and the content was redissolved in DCM (30 mL). The DCM layer was washed with brine (2 × 20 mL), dried over anhydrous $Na_2SO_4$, filtered, and concentrated under reduced pressure. The crude

product was purified by silica gel column chromatography using a mixture of methanol (2–15%) in DCM to get desired products **4**; (607 mg; Yield, 66%). $^1$H NMR (600 MHz, CDCl$_3$) δ = 9.80 (s, 2H), 8.14 (t, *J* = 1.7 Hz, 1H), 7.80 (dt, *J* = 7.9, 1.3 Hz, 1H), 7.75 (dt, *J* = 7.8, 1.3 Hz, 1H), 7.15 (t, *J* = 7.8 Hz, 1H), 6.81 (d, *J* = 8.3 Hz, 1H), 6.75 (d, *J* = 8.2 Hz, 2H), 4.73 (d, *J* = 8.1 Hz, 1H), 4.00 (t, *J* = 4.1 Hz, 1H), 3.84 (s, 3H), 3.79 (ddd, *J* = 12.5, 8.2, 4.6 Hz, 1H), 3.32 (d, *J* = 19.1 Hz, 1H), 3.21 (dd, *J* = 13.5, 4.2 Hz, 1H), 3.06 (dd, *J* = 19.2, 5.9 Hz, 1H), 2.83 (s, 1H), 2.62 (dt, *J* = 12.4, 3.7 Hz, 1H), 2.18 (td, *J* = 13.3, 4.6 Hz, 1H), 1.91–1.80 (m, 1H), 1.67 (dd, *J* = 13.5, 4.1 Hz, 1H), 1.53–1.44 (m, 1H), 1.13–1.03 (m, 1H). $^{13}$C NMR (151 MHz, CDCl$_3$) δ = 165.7, 149.8, 144.4, 144.0, 140.4, 136.2, 136.0, 136.0, 130.2, 127.5, 126.4, 123.7, 123.0, 120.4, 115.5, 94.2, 91.8, 56.9, 53.4, 52.6, 52.4, 42.7, 38.7, 37.9, 32.2, 28.0, 25.7, 23.3; ESI-MS m/z: 517.12 [M+H]$^+$.

### *N*-((7*R*,12*bS*)−3-(Cyclopropylmethyl)−9-methoxy-2,3,4,4*a*,5,6,7,7*a*-octahydro-1*H*-4,12-methanobenzofuro[3,2-e]isoquinolin-7-yl)−3-iodobenzamide (5)

Na$_2$CO$_3$ (92.4 mg, 0.87 mmol, 1.5 eq.) and (bromomethyl)cyclopropane (67.6 µL, 0.69 mmol, 1.2 eq.) were added to a stirred solution of dihydronorcodeine **4** (300 mg, 0.581 mmol) in DMF (1 mL) at rt under argon. The, the reaction mixture was heated to 90℃ overnight. Then the reaction mixture was cooled to rt and was diluted with EtOAc (20 mL). The EtOAc layer was washed with brine (5 × 20 mL) to remove DMF, dried over anhydrous Na$_2$SO$_4$, filtered, and concentrated under reduced pressure. The crude product was purified by silica gel column chromatography using a mixture of methanol (3–10%) in DCM to get desired products **5**; (205 mg; Yield, 62%). $^1$H NMR (600 MHz, CDCl$_3$) δ = 8.09 (t, *J* = 1.8 Hz, 1H), 7.79 (dt, *J* = 7.9, 1.3 Hz, 1H), 7.72 (dt, *J* = 7.8, 1.3 Hz, 1H), 7.13 (t, *J* = 7.8 Hz, 1H), 6.72 (d, *J* = 8.2 Hz, 1H), 6.64 (d, *J* = 8.2 Hz, 1H), 6.40 (s, 1H), 4.55 (d, *J* = 8.0 Hz, 1H), 3.82 (s, 3H), 3.75 (ddd, *J* = 12.4, 8.2, 4.8 Hz, 1H), 3.41 (s, 1H), 2.91 (d, *J* = 18.3 Hz, 1H), 2.82–2.76 (m, 1H), 2.39 (dd, *J* = 87.8, 49.2 Hz, 4H), 1.92 (d, *J* = 16.2 Hz, 1H), 1.70 (ddd, *J* = 12.3, 3.7, 1.7 Hz, 1H), 1.39 (td, *J* = 12.9, 2.5 Hz, 1H), 1.12 (qd, *J* = 12.9, 2.5 Hz, 1H), 0.87 (s, 1H), 0.53 (d, *J* = 8.0 Hz, 2H), 0.15 (s, 2H). $^{13}$C NMR (151 MHz, CDCl$_3$) δ = 171.2, 165.8, 143.8, 140.2, 136.6, 136.0, 130.1, 126.2, 119.2, 94.1, 92.9, 60.4, 59.8, 57.1, 56.6, 53.2, 45.7, 44.0, 28.7, 24.2, 21.1, 20.6, 14.2, 4.0, 3.8; HRMS calcd for C$_{28}$H$_{31}$IN$_2$O$_3$ [M+H]$^+$, 571.1458; found, 571.1474.

### *N*-((7*R*,12*bS*)−3-(Cyclopropylmethyl)−9-hydroxy-2,3,4,4*a*,5,6,7,7*a*-octahydro-1*H*-4,12-methanobenzofuro[3,2-e]isoquinolin-7-yl)−3-iodobenzamide (MP1202)

A solution of BBr$_3$ (7 mL, 7 eq.; 1M in DCM,) was slowly added to a stirred solution of methoxy morphinan **4** (181 mg, 0.31 mmol) in DCM (7 mL) at 0℃ under argon. The reaction mixture was stirred for 10 min at 0℃ and 20 more minutes at rt. The reaction mixture was quenched with excess of ammonia solution (5%) and the mixture was stirred for one hour. Then, the mixture was diluted with DCM (20 mL). The DCM layer was washed with brine (2 × 20 mL), dried over anhydrous Na$_2$SO$_4$, filtered, and concentrated under reduced pressure. The crude product was purified by silica gel column chromatography using a mixture of methanol (7–15%) in DCM to get desired products **MP1202**; (111 mg; Yield, 63%). $^1$H NMR (600 MHz, CDCl$_3$+CD$_3$OD) δ = 8.10 (t, *J* = 1.7 Hz, 1H), 7.74 (m, 2H), 7.11 (t, *J* = 7.8 Hz, 1H), 6.62 (d, *J* = 8.1 Hz, 1H), 6.49 (d, *J* = 8.1 Hz, 1H), 4.38 (d, *J* = 7.5 Hz, 1H), 3.80 (ddd, *J* = 12.7, 7.5, 4.8 Hz, 1H), 3.55 (s, 3H), 3.43–3.23 (m, 2H), 2.82 (d, *J* = 18.3 Hz, 1H), 2.68 (d, *J* = 12.1 Hz, 1H), 2.41 (s, 1H), 2.32 (d, *J* = 18.6 Hz, 2H), 2.08 (d, *J* = 18.1 Hz, 2H), 1.81–1.76 (m, 2H), 1.61 (m, 1H), 1.56–1.50 (m, 1H), 1.35 (qd, *J* = 13.0, 2.6 Hz, 1H), 1.09–0.97 (m, 1H), 0.79 (s, 1H), 0.49 (d, *J* = 8.0 Hz, 2H), 0.10 (dd, *J* = 14.1, 9.2 Hz, 2H). $^{13}$C NMR (151 MHz, CDCl$_3$+CD$_3$OD) δ = 166.7, 142.8, 140.4, 136.2, 136.1, 130.1, 126.4, 119.4, 118.0, 93.9, 93.6, 59.5, 56.8, 52.2, 45.8, 43.7, 2.78, 24.1, 20.3, 3.93, 3.8; HRMS calcd for C$_{27}$H$_{29}$IN$_2$O$_3$ [M+H]$^+$, 557.1301; found, 557.1304.

### (Z)−3-((2,3-*bis*(*tert*-Butoxycarbonyl)guanidino)methyl)benzoic acid (8)

*N*,*N*′-*di*-Boc-1*H*-pyrazole-1-carboxamidine (54 mg, 0.17 mmol, 1.1 eq.) and triethyl amine (67 µL, 0.44 mmol, three eq.) were added to a stirred solution of 3-(aminomethyl) benzoic acid hydrochloride (30 mg, 0.16 mmol, 1 eq.) in MeOH (1 mL) at rt under argon. The reaction was heated to 50℃ and continued for 7 hr. The solvent was removed under reduced pressure, and the content was diluted with EtOAc (15 mL). Water (15 mL) was added and the mixture was acidified with citric acid (to pH ~3). The organic layer was separated and was washed with brine (15 mL), dried over

anhydrous $Na_2SO_4$, filtered, and concentrated under reduced pressure. The crude product was purified by silica gel column chromatography using a mixture of EtOAc (20–50%) in hexanes to get the desired products two as a white solid (46 mg; Yield, 78%); [1]H NMR (600 MHz, Chloroform-d) δ = 11.56 (s, 1H), 8.67 (t, J = 5.4 Hz, 1H), 8.11–7.90 (m, 2H), 7.57 (dt, J = 7.8, 1.4 Hz, 1H), 7.46 (t, J = 7.7 Hz, 1H), 4.71 (d, J = 5.3 Hz, 2H), 1.50 (d, J = 17.8 Hz, 18H). [13]C NMR (151 MHz, CDCl$_3$) δ = 170.9, 163.5, 156.3, 153.2, 138.0, 133.1, 129.7, 129.6, 129.5, 129.0, 83.4, 79.6, 44.5, 28.3, 28.1.

## tert-Butyl (3-(((7R,12bS)−3-(Cyclopropylmethyl)−9-hydroxy-2,3,4,4a,5,6,7,7a-octahydro-1H-4,12-methanobenzofuro[3,2-e]isoquinolin-7-yl)carbamoyl)benzyl)carbamate (9)

The Boc aminomethylbenzoic acid **7** (28 mg, 0.11 mmol, 1.3 eq.) was added to a stirred solution of β-dihydro N-CPM morphineamine **6** (28 mg, 0.08 mmol, 1 eq.) in DMF (0.4 mL) at rt under argon. HATU (39 mg, 0.10 mmol, 1.2 eq.) was added to the mixture at rt at once and after 5 min, DIPEA (45 μL, 0.25 mmol, 3 eq.) was added. After 2 hr, the reaction mixture was diluted with EtOAc (15 mL). The EtOAc layer was washed with brine (5 × 15 mL) to remove DMF, dried over anhydrous $Na_2SO_4$, filtered, and concentrated under reduced pressure. The residue was redissolved in MeOH (0.5 mL) and sodium methoxide in MeOH (0.2 mL, 0.5 M) was added to the mixture. After 15 min, the solvent was removed, redissolved in EtOAc (15 mL), washed with brine, dried over anhydrous $Na_2SO_4$, filtered, and the crude product was purified by silica gel column chromatography using a mixture of methanol (3–10%) in DCM to get nine as a white solid, (31 mg, Yield; 66%); [1]H NMR (600 MHz, Chloroform-d) δ = 7.80 (s, 1H), 7.45 (d, J = 44.7 Hz, 2H), 7.34 (d, J = 7.6 Hz, 1H), 7.19 (t, J = 7.7 Hz, 1H), 6.72 (d, J = 8.0 Hz, 1H), 6.59 (d, J = 8.1 Hz, 1H), 5.33 (s, 1H), 4.69 (s, 1H), 4.25 (dd, J = 15.2, 5.9 Hz, 1H), 4.16 (dd, J = 15.2, 6.1 Hz, 1H), 3.81 (s, 1H), 3.07 (s, 1H), 2.91 (d, J = 18.3 Hz, 1H), 2.69 (s, 4H), 2.47–2.11 (m, 1H), 1.84 (d, J = 12.6 Hz, 1H), 1.71 (s, 1H), 1.56 (d, J = 32.4 Hz, 1H), 1.43 (s, 9H), 1.25 (s, 1H), 1.06 (q, J = 12.7 Hz, 1H), 0.94–0.77 (m, 1H), 0.65 (s, 2H), 0.29 (br, 2H). [13]C NMR (151 MHz, CDCl$_3$) δ = 167.3, 156.2, 143.0, 139.3, 134.3, 130.1, 128.6, 126.7, 125.7, 119.5, 92.4, 79.9, 59.1, 57.5, 52.0, 44.4, 43.1, 31.9, 29.7, 29.6, 29.3, 28.4, 24.0, 22.7, 21.0, 14.1, 4.4. ESI-MS m/z: 560.54 [M+H]$^+$.

## 2,3-Bis(tert-Butoxycarbonyl)(-N-(7R,12bS)−3-(cyclopropylmethyl)−9-hydroxy-2,3,4,4a,5,6,7,7a-octahydro-1H-4,12-methanobenzofuro[3,2-e]isoquinolin-7-yl)−3-(guanidinomethyl)benzamide (5)

The di-Boc-guanoomethylbenzoic acid **8** (32.9 mg, 0.08 mmol, 1.3 eq.) was added to a stirred solution of β-dihydro N-CPM morphineamine **6** (21 mg, 0.06 mmol, 1 eq.) in DMF (0.4 mL) at rt under argon. HATU (29 mg, 0.07 mmol, 1.2 eq.) was added to the mixture at rt at once and after 5 min, DIPEA (33 μL, 0.19 mmol, 3 eq.) was added. After 2 hr, the reaction mixture was diluted with EtOAc (15 mL). The EtOAc layer was washed with brine (5 × 15 mL) to remove DMF, dried over anhydrous $Na_2SO_4$, filtered, and concentrated under reduced pressure. The residue was redissolved in MeOH (0.5 mL) and sodium methoxide in MeOH (0.2 mL, 0.5 M) was added to the mixture. After 15 min, the solvent was removed, redissolved in EtOAc (15 mL), washed with brine, dried over anhydrous $Na_2SO_4$, filtered, and the crude product was purified by silica gel column chromatography using a mixture of methanol (3–10%) in DCM to get **10** as a white solid, (32 mg, Yield; 71%); [1]H NMR (600 MHz, Chloroform-d) δ = 11.52 (s, 1H), 8.82 (d, J = 6.9 Hz, 1H), 7.78 (s, 1H), 7.43 (d, J = 7.5 Hz, 1H), 7.29 (d, J = 7.6 Hz, 1H), 7.19 (t, J = 7.7 Hz, 1H), 7.07 (d, J = 14.4 Hz, 1H), 6.82 (d, J = 8.1 Hz, 1H), 6.63 (d, J = 8.2 Hz, 1H), 4.87 (s, 1H), 4.57 (dd, J = 15.4, 6.7 Hz, 1H), 4.44–4.34 (m, 1H), 3.85 (s, 2H), 3.24 (s, 1H), 2.97 (d, J = 18.1 Hz, 1H), 2.74 (s, 3H), 2.48 (s, 1H), 2.37–2.16 (m, 1H), 2.03–1.91 (m, 1H), 1.83 (s, 1H), 1.62 (dd, J = 30.8, 8.7 Hz, 1H), 1.49 (d, J = 6.2 Hz, 18H), 1.37 (dd, J = 13.6, 8.9 Hz, 1H), 1.32–1.21 (m, 1H), 1.13 (q, J = 12.7 Hz, 1H), 0.93–0.78 (m, 1H), 0.68 (s, 2H), 0.37 (d, J = 54.7 Hz, 2H). [13]C NMR (151 MHz, CDCl$_3$) δ = 167.0, 163.1, 156.3, 153.2, 143.1, 141.4, 137.9, 133.7, 129.8, 128.6, 126.5, 126.2, 120.0, 118.4, 92.4, 83.5, 80.0, 59.2, 57.9, 52.2, 46.7, 44.0, 43.0, 32.0, 29.7, 29.7, 29.6, 29.5, 29.4, 28.5, 28.3, 28.2, 28.1, 28.0, 23.8, 22.7, 21.1, 14.2, 14.2, 4.5. ESI-MS m/z: 702.55 [M+H]$^+$.

### 3-(Aminomethyl)-*N*-((*7R,12bS*)−3-(cyclopropylmethyl)−9-hydroxy-2,3,4,4*a*,5,6,7,7*a*-octahydro-1*H*-4,12-methanobenzofuro[3,2-e]isoquinolin-7-yl)benzamide (MP1207)

Triethylsilane (21 µL, 0.19 mmol, 2.5 eq.) was added to the solution of **9** (30 mg, 0.05 mmol one eq.) in TFA/DCM (1:1, 1 mL) at rt. The reaction was continued for 30 min. Mass spectrometry indicated the reaction was completed. The volatile components were removed under reduced pressure. The content was redissolved in DCM (0.4 mL) and treated with Et$_2$O (3 mL) while shaking resulting in a white precipitation. The precipitate was washed with Et$_2$O (3 mL) and dried under high vacuum to get the desired amino product **MP1207** as a white solid (33 mg, Yield 94%); $^1$H NMR (600 MHz, Deuterium Oxide) δ = 7.79 (dd, J = 7.0, 1.4 Hz, 2H), 7.64 (dt, J = 7.9, 1.4 Hz, 1H), 7.61–7.53 (m, 1H), 6.95–6.85 (m, 1H), 6.85–6.75 (m, 1H), 4.74 (dd, J = 11.0, 8.1 Hz, 1H), 4.23 (s, 3H), 3.77 (ddt, J = 12.2, 8.3, 4.1 Hz, 1H), 3.42 (td, J = 11.6, 9.7, 5.4 Hz, 1H), 3.36–3.26 (m, 1H), 3.20 (d, J = 19.7 Hz, 1H), 3.10–2.95 (m, 2H), 2.76 (td, J = 13.2, 4.0 Hz, 1H), 2.48 (ddd, J = 12.5, 4.6, 2.8 Hz, 1H), 2.14 (td, J = 13.5, 4.7 Hz, 1H), 1.92 (ddd, J = 13.8, 4.2, 1.7 Hz, 1H), 1.87–1.72 (m, 2H), 1.54 (qd, J = 13.1, 2.5 Hz, 1H), 1.19–0.99 (m, 2H), 0.83–0.61 (m, 2H), 0.46–0.28 (m, 2H). $^{13}$C NMR (151 MHz, D$_2$O, without TFA peaks) δ = 170.0, 142.1, 139.8, 134.5, 133.1, 132.2, 129.5, 128.0, 127.6, 122.9, 120.5, 92.1, 58.8, 58.6, 51.6, 46.5, 42.6, 42.4, 42.1, 39.7, 32.4, 27.4, 23.1, 20.4, 5.2, 3.6, 3.4, HRMS calcd for C$_{28}$H$_{34}$N$_3$O$_3$ [M+H]$^+$, 460.2600; found, 460.2585.

### *N*-((*7R,12bS*)−3-(Cyclopropylmethyl)−9-hydroxy-2,3,4,4*a*,5,6,7,7*a*-octahydro-1*H*-4,12-methanobenzofuro[3,2-e]isoquinolin-7-yl)−3-(guanidinomethyl)benzamide (MP1208)

Triethylsilane (35.6 µL, 0.22 mmol, 5 eq.) was added to the solution of **10** (31 mg, 0.04 mmol, 1 eq.) in TFA/DCM (1:1, 1 mL) at rt. The reaction was continued for 30 min. Mass spectrometry indicated the reaction was completed. The volatile components were removed under reduced pressure. The content was redissolved in DCM (0.4 mL) and treated with Et$_2$O (3 mL) while shaking resulting in a white precipitation. The precipitate was washed with Et$_2$O (3 mL) and dried under high vacuum to get the desired amino product **MP1208** as a white solid (28 mg, Yield 87%); $^1$H NMR (600 MHz, Deuterium Oxide) δ = 7.64–7.51 (m, 2H), 7.48–7.28 (m, 2H), 6.86–6.49 (m, 2H), 4.68–4.57 (m, 1H), 4.37 (d, J = 5.7 Hz, 2H), 4.12 (dd, J = 5.7, 2.8 Hz, 1H), 3.66 (ddd, J = 12.7, 8.2, 4.3 Hz, 1H), 3.31 (td, J = 16.6, 15.0, 5.9 Hz, 1H), 3.22–3.15 (m, 1H), 3.10 (dd, J = 19.8, 3.2 Hz, 1H), 3.00–2.85 (m, 2H), 2.65 (td, J = 13.2, 4.2 Hz, 1H), 2.38 (ddd, J = 12.3, 4.4, 2.7 Hz, 1H), 2.04 (td, J = 13.5, 4.6 Hz, 1H), 2.00–1.92 (m, 0H), 1.87–1.76 (m, 1H), 1.76–1.62 (m, 1H), 1.43 (tdd, J = 13.9, 10.4, 5.3 Hz, 1H), 0.98 (tdd, J = 13.7, 8.9, 4.8 Hz, 2H), 0.71–0.58 (m, 2H), 0.34–0.19 (m, 2H). $^{13}$C NMR (151 MHz, D$_2$O, without TFA peaks) δ = 170.3, 156.8, 142.2, 139.7, 136.7, 134.2, 130.3, 129.2, 128.0, 126.3, 125.5, 122.9, 120.5, 118.1, 92.1, 58.8, 58.6, 51.6, 46.5, 44.1, 42.1, 39.7, 32.4, 27.4, 23.1, 20.4, 5.2, 3.6, 3.4. HRMS calcd for C$_{29}$H$_{36}$N$_5$O$_3$ [M+H]$^+$, 502.2818; found, 502.2816.

### 4-(Aminomethyl)-N-((4R,4aR,7R,7aR,12bS)−3-(cyclopropylmethyl)−9-hydroxy-2,3,4,4a,5,6,7,7a-octahydro-1H-4,12-methano[1]benzofuro[3,2-e]isoquinolin-7-yl)benzamide (MP1209)

To a suspension of PyBOP (70 mg, 0.135 mmol, 2.2 eq.) in THF (100 mL) were sequentially added a solution of 4-(((tert-butoxy carbonyl)amino)methyl)benzoic acid (34 mg, 0.135 mmol, 2.2 eq.) in THF (100 mL) and NEt(iPr)$_2$ (24 mL, 0.135 mmol, 2.2 eq.) and the resulting mixture was stirred for 40 min at rt. This solution was subsequently added via cannula to a suspension of solution of *β*-dihydro N-CPM morphineamine **6** (20 mg, 0.061 mmol, 1.0 eq.) in THF (100 mL) and the mixture was stirred at rt overnight. To the crude mixture methanol and potassium carbonate were added and stirring was continued for 2 hr. After filtration the solvent was removed on a rotary evaporator, and the crude product was loaded on a 4 g Silica Gold column. Chromatography was performed with 5% to 10% MeOH (containing 10% concentrated NH$_4$OH solution) gradient in 6 min. The desired Boc protected intermediate eluted around 5–6 min. Deprotection of the Boc protected amine (**11**) was achieved in 3 hr using 4N HCl in dioxane. The product was obtained as a white powder after removal of the solvent and trituration with diethyl ether (25 mg, 76%). $^1$H NMR (400 MHz, CDCl3) δ: 7.92 (dd, J = 8.3, 2.1 Hz, 2H), 7.60–7.53 (m, 2H), 6.79–6.67 (m, 2H), 4.78 (dd, J = 8.0, 1.9 Hz, 1H),

4.18 (s, 3H), 3.88–3.71 (m, 1H), 3.64 (d, J = 1.9 Hz, 2H), 3.51–3.39 (m, 1H), 3.38–3.28 (m, 1H), 3.20 (d, J = 19.0 Hz, 1H), 3.06–2.92 (m, 2H), 2.73 (td, J = 13.0, 4.0 Hz, 1H), 2.63 (d, J = 12.2 Hz, 1H), 2.26 (td, J = 13.5, 4.7 Hz, 1H), 1.95–1.71 (m, 3H), 1.69–1.49 (m, 1H), 1.24–1.06 (m, 3H), 0.78 (dd, J = 10.7, 7.5 Hz, 2H), 0.58–0.43 (m, 2H). $^{13}$C NMR (101 MHz, CDCl3) δ: 164.45, 139.51, 138.14, 133.32, 131.56, 125.44, 124.54, 124.52, 124.29, 117.92, 116.55, 114.77, 87.95, 63.45, 55.79, 55.34, 48.72, 39.16, 39.09, 36.66, 29.62, 24.71, 20.07, 17.26, 2.17. HRMS calcd for $C_{28}H_{33}N_3O_3$ [M+H]$^+$, 460.2594; found, 460.2599.

## N-((4R,4aR,7R,7aR,12bS)−3-(cyclopropylmethyl)−9-hydroxy-2,3,4,4a,5,6,7,7a-octahydro-1H-4,12-methano[1]benzofuro[3,2-e]isoquinolin-7-yl)−3-(hydroxymethyl)benzamide (MP1210)

To a suspension of PyBOP (70 mg, 0.135 mmol, 2.2 eq.) in THF (100 mL) were sequentially added a solution of 3-(hydroxymethyl)benzoic acid (21 mg, 0.135 mmol, 2.2 eq.) in THF (100 mL) and NEt (iPr)$_2$ (24 mL, 0.135 mmol, 2.2 eq.) and the resulting mixture was stirred for 40 min at rt. This solution was subsequently added via cannula to a suspension of solution of β-dihydro N-CPM morphineamine 6 (20 mg, 0.061 mmol, 1.0 eq.) in THF (100 mL) and the mixture was stirred at rt overnight. To the crude mixture methanol and potassium carbonate were added and stirring was continued for 2 hr. After filtration, the solvent was removed on a rotary evaporator, and the crude product was loaded on a 4 g Silica Gold column. Chromatography was performed with 5% to 20% MeOH (containing 10% concentrated NH$_4$OH solution) gradient in 12 min. The desired product eluted around 11–12 min (15 mg, Yield 51%). $^1$H NMR (400 MHz, CDCl3) δ: 8.27 (s, 1H), 8.20 (d, J = 7.7 Hz, 1H), 7.97 (d, J = 7.7 Hz, 1H), 7.89 (dt, J = 10.6, 5.1 Hz, 1H), 7.19 (d, J = 8.3 Hz, 1H), 7.09 (d, J = 8.1 Hz, 1H), 5.14 (s, 2H), 5.09–5.03 (m, 1H), 4.87 (s, 5H), 4.40–4.29 (m, 1H), 4.26 (s, 1H), 3.82 (d, J = 14.1 Hz, 1H), 3.55 (d, J = 11.7 Hz, 1H), 3.44 (s, 1H), 3.32 (s, 1H), 3.20 (s, 1H), 3.16 (s, 1H), 3.03 (s, 1H), 2.91 (s, 1H), 2.68–2.63 (m, 0H), 2.62 (s, 1H), 2.34 (d, J = 13.3 Hz, 1H), 2.25 (d, J = 13.1 Hz, 1H), 2.13 (d, J = 13.1 Hz, 1H), 1.97 (q, J = 13.1 Hz, 1H), 1.63–1.52 (m, 2H), 1.15 (d, J = 8.1 Hz, 2H), 0.83 (d, J = 13.5 Hz, 2H).13C NMR (101 MHz, CDCl3) δ: 169.20, 143.41, 142.19, 141.22, 134.71, 130.51, 128.80, 126.51, 125.92, 120.05, 118.79, 93.14, 64.24, 59.50, 58.21, 52.41, 46.80, 43.52, 40.79, 33.97, 28.81, 24.26, 21.39, 7.49, 4.65, 4.46. HRMS calcd for $C_{28}H_{32}N_2O_4$ [M+H]$^+$, 461.2435; found, 460.2440.

## (4aS,7S,12bS)−3-(Cyclopropylmethyl)−4a,9-dimethoxy-2,3,4,4a,5,6,7,7a-octahydro-1H-4,12-methanobenzofuro[3,2-e]isoquinolin-7-ol (13)

A solution of Li-selectride (1.3 mL, 1.3 mmol, 2 eq. 1M in THF) was slowly added to the stirred solution of known ketone (*Kobylecki et al., 1982*) 12 (320 mg, 0.86 mmol) in THF (5 mL) at −78°C. The reaction mixture was continued at the temperature for 90 min. The reaction mixture was quenched with cold methanol (0.2 mL) at −78°C and the cold bath was removed and is continued stirring for 10 min to warm up to rt. Then, the product was extracted four times (4 × 20 mL) using DCM. The organic layer was dried over anhydrous sodium sulfate, filtered, and concentrated under reduced pressure. The crude product was purified by silica column (ISCO flash column) using methanol (1–2%) in DCM. The product fraction was concentrated and dried under high vacuum to get desired products 13; (240 mg, Yield 75%); $^1$H NMR (600 MHz, Chloroform-d) δ = 6.71 (d, J = 8.2 Hz, 1H), 6.58 (d, J = 8.2 Hz, 1H), 4.66 (dd, J = 4.8, 1.1 Hz, 1H), 4.22 (dt, J = 9.4, 4.6 Hz, 1H), 3.87 (s, 3H), 3.53 (d, J = 6.0 Hz, 1H), 3.29 (s, 3H), 3.12 (d, J = 18.3 Hz, 1H), 2.65 (dd, J = 12.0, 5.2 Hz, 1H), 2.53–2.40 (m, 2H), 2.38 (dd, J = 18.4, 6.1 Hz, 1H), 2.28–2.10 (m, 2H), 1.88–1.73 (m, 1H), 1.66 (ddt, J = 14.7, 8.4, 4.0 Hz, 1H), 1.42 (ddd, J = 12.3, 3.8, 1.6 Hz, 1H), 1.16–1.02 (m, 2H), 0.93–0.84 (m, 1H), 0.62–0.43 (m, 2H), 0.16 – −0.06 (m, 2H). ESI-MS m/z: 372.4 [M+H]$^+$.

## (4aS,7R,12bS)−3-(Cyclopropylmethyl)−4a,9-dimethoxy-2,3,4,4a,5,6,7,7a-octahydro-1H-4,12-methanobenzofuro[3,2-e]isoquinolin-7-amine (14)

Phthalimide (36.4 mg, 0.24 mmol, 2 eq.) and triphenylphosphine (48.7 mg, 0.18 mmol, 1.5 eq.) were added to a stirred solution of alcohol 13 (46 mg, 0.12 mmol) in THF (1 mL.) under argon. Diisopropyl azodicarboxylate (36.8 μL, 0.18 mmol, 1.5 eq.) was added drop wise to the reaction mixture at 0°C and the reaction was continued for overnight. The reaction mixture was quenched with water (0.5 mL) and the mixture was stirred for 10 min at rt. The organic solvent was removed under reduced

pressure. The residue was treated with 2% aqueous citric acid (1 mL), and dilute HCl (0.1 M, 1 mL) solutions. The mixture was washed with Et$_2$O (3 × 2 mL). Then, the aqueous layer was basified (~ pH = 10) using an ammonia solution. The product was extracted in CHCl$_3$ (3 × 5 mL), dried over anhydrous Na$_2$SO$_4$, filtered, and concentrated under reduced pressure. The crude product was purified by silica gel column chromatography using methanol (1–5%) in DCM to obtain white solid of phthalimides intermediates (19 mg, 32%) upon drying. Then, hydrazine hydrate (23.8 µL, 10 eq.) was added to the stirred solution of the phthalimide intermediate in methanol (2 mL) at once heated to 50°C for 90 min to complete the reaction. The reaction mixture was diluted with DCM (4 mL) and the product was extracted in dilute HCl (3 × 2 mL), the aqueous layer was basified with dilute NH$_4$OH (5%) and the amine is extracted in DCM (3 × 2 mL). The DCM layer was dried over anhydrous Na$_2$SO$_4$, filtered and concentrated under reduced pressure to get the desired product **14** (14 mg, yield; quantitative, which was used without further purification); [1]H NMR (600 MHz, Chloroform-*d*) δ 6.69 (d, *J* = 8.2 Hz, 1H), 6.59 (d, *J* = 8.1 Hz, 1H), 4.30 (d, *J* = 6.9 Hz, 1H), 3.86 (s, 3H), 3.55 (d, *J* = 5.1 Hz, 1H), 3.25 (s, 3H), 3.11 (d, *J* = 18.2 Hz, 1H), 2.69–2.52 (m, 2H), 2.52–2.26 (m, 4H), 2.05 (td, *J* = 12.1, 3.9 Hz, 1H), 1.82 (dt, *J* = 14.3, 3.3 Hz, 1H), 1.63 (qd, *J* = 12.8, 2.7 Hz, 1H), 1.52–1.39 (m, 1H), 1.36–1.20 (m, 2H), 1.20–1.05 (m, 1H), 0.87 (tt, *J* = 8.9, 7.1, 3.3 Hz, 1H), 0.64–0.43 (m, 2H), 0.24–0.09 (m, 2H). ESI-MS *m/z*: 371.37 [M+H]$^+$.

### *N*-((4*aS*,7*R*,12*bS*)−3-(Cyclopropylmethyl)−9-hydroxy-4*a*-methoxy-2,3,4,4*a*,5,6,7,7*a*-octahydro-1*H*-4,12-methanobenzofuro[3,2-e]isoquinolin-7-yl)−3-iodobenzamide (MP1305)

m-Iodobenzoic acid (8.86 mg, 0.035 mmol, 1.3 eq.) was added to a stirred solution of *β*-amine **14** (12 mg, 0.032 mmol) in DMF (0.2 mL) at. HATU (14.7 mg, 0.038 mmol, 1.3 eq.) was added to the mixture at rt at once and after 5 min, DIPEA (16.9 µL, 3 eq.) was added and the reaction was continued for 1 hr. Then, the reaction mixture was diluted with EtOAc (5 mL) and the EtOAc layer was washed with brine (5 × 3 mL) to remove DMF. The organic layer was dried over anhydrous Na$_2$SO$_4$, filtered, and concentrated under reduced pressure. The crude product was purified by silica gel column chromatography using a mixture of methanol (1–2%) in DCM to obtain white solid of methoxy intermediates (10 mg, 52%) upon drying. Then, BBr$_3$ solution (116 µL, 0.116 mmol, 7 eq., 1M in DCM) was added to a stirred solution of methoxy intermediate (10 mg, 0.016 mmol) in DCM (2 mL) at 0°C under argon. The reaction mixture was continued for 10 min at 0°C and 20 more minutes at rt. The, reaction mixture was diluted with DCM (8 mL) and quenched with excess of NH$_4$OH (5%, 2 mL) and the mixture was stirred for 1 hr. The DCM layer was washed with brine (2 × 2 mL), dried over anhydrous Na$_2$SO$_4$, filtered, and concentrated under reduced pressure. The crude product was purified by silica gel column chromatography using a mixture of methanol (5–10%) in DCM to get desired products **MP1305** (4.9 mg, Yield 26%); [1]H NMR (600 MHz, Methanol-*d*$_4$) δ = 8.21 (t, *J* = 1.7 Hz, 1H), 8.02–7.75 (m, 1H), 7.25 (t, *J* = 7.8 Hz, 1H), 6.69 (q, *J* = 8.2 Hz, 2H), 4.71 (d, *J* = 7.7 Hz, 1H), 4.09 (s, 1H), 3.87 (ddd, *J* = 12.7, 7.7, 5.2 Hz, 1H), 2.82 (s, 3H), 2.73–2.35 (m, 4H), 2.06 (d, *J* = 14.6 Hz, 1H), 1.87–1.61 (m, 2H), 1.56–1.27 (m, 3H), 1.02 (s, 1H), 0.68 (d, *J* = 55.0 Hz, 3H), 0.37 (s, 3H). [13]C NMR (151 MHz, MeOD) δ = 168.30, 143.52, 141.68, 137.67, 137.48, 131.39, 127.62, 120.51, 119.10, 94.71, 92.55, 76.95, 59.48, 56.11, 54.81, 53.48, 29.51, 24.91, 24.50, 23.97, 5.64, 3.31,−0.03. HRMS calcd for C$_{38}$H$_{31}$IN$_2$O$_4$ [M+H]$^+$, 587.1407; found, 587.1388.

### (4*bR*,6*R*,8*aS*)−6-Amino-11-(cyclopropylmethyl)−3-methoxy-5,6,7,8,9,10-hexahydro-8*aH*-9,4*b*-(epiminoethano)phenanthren-8*a*-ol (16)

NH$_4$OAc (802.4 mg, 10.41 mmol, 20 eq.) was added to a stirred solution of known ketone (*Zhang et al., 2007*) **15** (177.7 mg, 0.52 mmol) in dry MeOH (4 mL) under argon at rt. The mixture was stirred for 8 hr. Then, NaCNBH$_3$ (163.6 mg, 2.60 mmol, 5 eq.) was added to the reaction mixture and the reaction was continued for overnight. The solvent was evaporated under reduced pressure and the content was stirred with aqueous HCl (1M, 10 mL) for 2 hr at rt. The aqueous layer was diluted with water (20 mL), basified with concentrated NH$_4$OH solution (to pH ~10) and then the product was extracted in DCM (3 × 15 mL). The combined DCM layer was washed with brine (10 mL), dried over anhydrous Na$_2$SO$_4$, filtered and concentrated. The crude product was purified by silica gel column chromatography using a mixture of 10% TEA, and 1% MeOH in DCM. The **β** amine is more polar than **α** counterpart on using the condition, which was eluted later and upon drying

furnished a white solid of the desired product **16** (**β**, 90 mg, Yield 51%); [1]H NMR (600 MHz, CDCl$_3$) δ = 6.99 (d, J = 8.4 Hz, 1H), 6.86 (d, J = 2.6 Hz, 1H), 6.69 (dd, J = 8.4, 2.6 Hz, 1H), 4.65 (s, 1H), 3.78 (s, 3H), 3.00–2.91 (m, 2H), 2.77–2.66 (m, 2H), 2.56–2.48 (m, 1H), 2.38–2.27 (m, 2H), 2.15 (dd, J = 13.2, 3.6 Hz, 1H), 2.07–2.00 (m, 2H), 1.80 (dd, J = 13.2, 11.8 Hz, 1H), 1.63 (p, J = 4.8 Hz, 2H), 1.54–1.41 (m, 2H), 1.09–1.00 (m, 1H), 0.86–0.78 (m, 1H), 0.50 (dd, J = 8.1, 1.7 Hz, 2H), 0.12–0.06 (m, 2H). [13]C NMR (151 MHz, CDCl$_3$) δ = 158.4, 142.8, 128.3, 128.1, 111.1, 111.0, 68.7, 60.3, 59.4, 55.4, 46.4, 44.2, 42.3, 40.9, 37.0, 32.6, 31.2, 24.6, 9.6, 4.0, 3.9. ESI-MS m/z: 343.2 [M+H]$^+$.

### *N*-((4*bR*,6*R*,8*aS*)−11-(Cyclopropylmethyl)−8*a*-hydroxy-3-methoxy-6,7,8,8*a*,9,10-hexahydro-5H-9,4*b*-(epiminoethano)phenanthren-6-yl)−3-iodobenzamide (17)

DIPEA (203 µL, 1.16 mmol, 3 eq.) and **β**-amine **16** (80 mg, 0.23 mmol) were added to the stirred solution of *m*-iodobenzoic acid (86.9 mg, 0.35 mmol, 1.5 eq.) dissolved in DMF (1 mL) at rt under an argon atmosphere. The reaction mixture was cooled to 0°C and HATU (133.2 mg, 0.35 mmol, 1.3 eq.) was added to the reaction mixture. After stirring the reaction mixture for 4 hr at 0°C to rt, the reaction mixture was poured into EtOAc (15 mL) and was washed with brine (5 × 15 mL). The EtOAc layer was dried over Na$_2$SO$_4$, filtered; and concentrated under reduced pressure. Then, the residue was purified by silica gel column chromatography using 2–5% MeOH in DCM. The desired product fractions were concentrated under reduced pressure and dried under high vacuum to get amorphous solid of the product **17** (110 mg, Yield 83%); [1]H NMR (600 MHz, MeOD) δ = 8.20 (t, J = 1.7 Hz, 1H), 7.91 (dt, J = 7.9, 1.3 Hz, 1H), 7.83 (dt, J = 7.8, 1.3 Hz, 1H), 7.26 (t, J = 7.8 Hz, 1H), 7.22–7.17 (m, 2H), 6.88 (dd, J = 8.5, 2.5 Hz, 1H), 4.04–3.98 (m, 1H), 3.89 (s, 3H), 3.85 (s, 1H), 3.31–3.21 (m, 3H), 3.02 (d, J = 12.5 Hz, 1H), 2.90 (s, 1H), 2.68 (s, 1H), 2.56–2.43 (m, 2H), 2.13 (t, J = 12.7 Hz, 1H), 2.07–1.98 (m, 1H), 1.82–1.70 (m, 3H), 1.36–1.32 (m, 1H), 1.12 (dd, J = 8.8, 4.2 Hz, 1H), 0.83 (d, J = 8.7 Hz, 1H), 0.76 (td, J = 8.8, 4.6 Hz, 1H), 0.51 (s, 2H). [13]C NMR (151 MHz, MeOD) δ = 168.1, 161.0, 141.5, 138.0, 137.5, 131.3, 130.2, 127.6, 115.1, 111.1, 94.6, 69.6, 62.5, 58.7, 55.9, 46.5, 42.2, 35.8, 31.7, 27.4, 25.4, 6.0, 3.4. ESI-MS m/z: 569.2 [M+H]$^+$.

### *N*-((4*bR*,6*R*,8*aS*)−11-(Cyclopropylmethyl)−3,8*a*-dihydroxy-6,7,8,8*a*,9,10-hexahydro-5H-9,4*b*-(epiminoethano)phenanthren-6-yl)−3-iodobenzamide (MP1601)

A solution of BBr$_3$ (874 µL, 0.87 mmol, 5 eq.) was added to the stirred solution of methyl ether **17** (100 mg, 0.17 mmol) dissolved in DCM (4 mL) at 0°C under an argon atmosphere. The reaction was continued at 0°C for 10 min and then at rt for 20 min. The reaction mixture was diluted with DCM (5 mL) and treated with aqueous NH$_4$OH (5%, 2 mL) for 1 hr. The, the DCM layer was separated and was washed with saturated solution of NaHCO$_3$ (2 × 10 mL), brine (5 mL), dried over anhydrous Na$_2$SO$_4$, filtered; and concentrated under reduced pressure. Then, the residue was purified by silica gel column chromatography using 10–15% MeOH in DCM. The desired product fractions were concentrated under reduced pressure and dried under high vacuum to get amorphous solid of **MP1601**; (72 mg, Yield 74%); [1]H NMR (600 MHz, CDCl$_3$) δ = 8.06 (d, J = 1.8 Hz, 1H), 7.83–7.72 (m, 1H), 7.67 (dt, J = 7.6, 1.2 Hz, 1H), 7.14 (t, J = 7.8 Hz, 1H), 7.03 (d, J = 2.5 Hz, 1H), 6.95 (d, J = 8.3 Hz, 1H), 6.71 (dd, J = 8.3, 2.4 Hz, 1H), 6.09 (s, 1H), 4.02 (tdt, J = 11.9, 8.1, 4.2 Hz, 1H), 3.02 (d, J = 18.2 Hz, 1H), 2.75 (d, J = 18.0 Hz, 1H), 2.59 (s, 1H), 2.45–2.27 (m, 3H), 2.14 (t, J = 15.5 Hz, 1H), 2.04 (d, J = 5.9 Hz, 1H), 1.95–1.74 (m, 3H), 1.63 (td, J = 13.3, 4.4 Hz, 1H), 1.55 (d, J = 13.4 Hz, 1H), 1.31–1.16 (m, 1H), 1.14–1.03 (m, 1H), 0.86 (d, J = 16.2 Hz, 1H), 0.52 (d, J = 8.0 Hz, 2H), 0.13 (s, 2H). [13]C NMR (151 MHz, CDCl$_3$) δ = 165.3, 140.3, 136.6, 136.0, 130.3, 128.6, 126.1, 114.1, 112.0, 94.3, 68.8, 60.2, 59.3, 45.8, 44.1, 41.8, 36.3, 30.8, 27.9, 24.5, 4.0, 3.9. HRMS calcd for C$_{27}$H$_{32}$IN$_2$O$_3$ [M+H]$^+$, 559.1458; found, 559.1457.

## Acknowledgements

SM is supported by funds from NIH grants DA045884, DA046487, AA026949 and W81XWH-17-1-0256 (Office of the Assistant Secretary of Defense for Health Affairs through the Peer Reviewed Medical Research Program) and start-up funds from Center for Clinical Pharmacology, St. Louis College of Pharmacy and Washington University to SM. This research was funded in part through the

NIH/NCI Cancer Center Support Grant P30 CA008748 to MSKCC. Studies were also supported by NIDA grants R33DA038858 (VK) and P01DA035764 (VK and BLR), DA042888 and DA046714 (YXP) and DA007242 and DA006241 (GWP/YXP). T32 MH018870 (SGG), MH54137 and the Hope for Depression Research Foundation (JAJ). This work was also supported by NIH grants (R37DA045657 and RO1MH112205), the NIMH Psychoactive Drug Screening Program Contract, and the Michael Hooker Distinguished Chair of Pharmacology (to B.L.R.) and funds from the University of Florida (J.P. M.).

## Additional information

### Competing interests

Rajendra Uprety: RU have filed a provisional patent on MP1207 and related molecules. Saheem A Zaidi: SZ has filed a provisional patent on MP1207 and related molecules. Ying Xian Pan: YXP is a co-founder of Sparian biosciences. Jay P McLaughlin: JM has filed a provisional patent on MP1207 and related molecules. Bryan L Roth: BLR has filed a provisional patent on MP1207 and related molecules. Gavril W Pasternak: GWP is a co-founder of Sparian biosciences. GWP have filed a provisional patent on MP1207 and related molecules. Vsevolod Katritch: VK has filed a provisional patent on MP1207 and related molecules. Susruta Majumdar: SM. The other authors declare that no competing interests exist.

### Funding

| Funder | Grant reference number | Author |
|---|---|---|
| National Institute on Drug Abuse | DA045884 | Susruta Majumdar |
| National Institute on Drug Abuse | DA046487 | Susruta Majumdar |
| National Institute on Alcohol Abuse and Alcoholism | AA026949 | Susruta Majumdar |
| National Institute on Drug Abuse | DA038858 | Vsevolod Katritch |
| National Institute on Drug Abuse | DA035764 | Bryan L Roth Vsevolod Katritch |
| National Institute on Drug Abuse | DA007242 | Ying Xian Pan Gavril W Pasternak |
| National Institute on Drug Abuse | DA042888 | Ying Xian Pan |
| National Institute of Mental Health | MH018870 | Steven G Grinnell |
| National Institute of Mental Health | MH112205 | Jonathan A Javitch |
| National Institute on Drug Abuse | DA045657 | Jonathan A Javitch |
| National Institute on Drug Abuse | DA006241 | Ying Xian Pan Gavril W Pasternak |
| National Institute on Drug Abuse | DA046714 | Ying Xian Pan |
| National Institutes of Health | W81XWH-17-1-0256 | Susruta Majumdar |
| St. Louis College of Pharmacy and Washington University | | Susruta Majumdar |
| National Institute on Drug Abuse | MH54137 | Jonathan A Javitch |
| Hope for Depression Research Foundation | | Jonathan A Javitch |

| | |
|---|---|
| National Institute of Mental Health | Bryan L Roth |
| University of Florida Foundation | Jay P McLaughlin |

The funders had no role in study design, data collection and interpretation, or the decision to submit the work for publication.

## Author contributions

Rajendra Uprety, Conceptualization, Resources, Data curation, Formal analysis, Supervision, Investigation, Visualization, Writing - original draft, Project administration, Writing - review and editing; Tao Che, Balázs R Varga, Resources, Data curation, Formal analysis, Investigation, Methodology, Writing - review and editing; Saheem A Zaidi, Abdelfattah Faouzi, Conceptualization, Resources, Data curation, Formal analysis, Investigation, Methodology, Writing - original draft, Writing - review and editing; Steven G Grinnell, Conceptualization, Resources, Data curation, Formal analysis, Investigation, Methodology, Writing - review and editing; Samuel T Slocum, Resources, Data curation, Formal analysis, Methodology, Writing - review and editing; Abdullah Allaoa, András Varadi, Resources, Data curation, Writing - review and editing; Melissa Nelson, Conceptualization, Resources, Data curation, Formal analysis, Supervision, Funding acquisition, Investigation, Visualization, Writing - original draft, Project administration, Writing - review and editing; Sarah M Bernhard, Resources, Data curation, Investigation, Writing - review and editing; Elizaveta Kulko, Valerie Le Rouzic, Shainnel O Eans, Chloe A Simons, Amanda Hunkele, Resources, Data curation; Joan Subrath, Resources, Data curation, Formal analysis, Supervision, Funding acquisition, Writing - review and editing; Ying Xian Pan, Conceptualization, Resources, Data curation, Formal analysis, Supervision, Funding acquisition, Writing - original draft, Writing - review and editing; Jonathan A Javitch, Conceptualization, Resources, Data curation, Formal analysis, Supervision, Funding acquisition, Methodology, Writing - original draft, Writing - review and editing; Jay P McLaughlin, Conceptualization, Resources, Data curation, Formal analysis, Supervision, Funding acquisition, Validation, Investigation, Methodology, Writing - original draft, Project administration, Writing - review and editing; Bryan L Roth, Conceptualization, Data curation, Formal analysis, Supervision, Funding acquisition, Validation, Investigation, Methodology, Writing - original draft, Project administration, Writing - review and editing; Gavril W Pasternak, Vsevolod Katritch, Susruta Majumdar, Conceptualization, Resources, Data curation, Formal analysis, Supervision, Funding acquisition, Validation, Investigation, Visualization, Methodology, Writing - original draft, Project administration, Writing - review and editing

## Author ORCIDs

Saheem A Zaidi (iD) https://orcid.org/0000-0001-7531-3587
Balázs R Varga (iD) https://orcid.org/0000-0003-0986-9477
Abdelfattah Faouzi (iD) http://orcid.org/0000-0002-9059-4791
András Varadi (iD) http://orcid.org/0000-0001-5591-377X
Sarah M Bernhard (iD) http://orcid.org/0000-0001-8549-0413
Jonathan A Javitch (iD) http://orcid.org/0000-0001-7395-2967
Vsevolod Katritch (iD) https://orcid.org/0000-0003-3883-4505
Susruta Majumdar (iD) https://orcid.org/0000-0002-2931-3823

## Ethics

Animal experimentation: All animal studies were preapproved by the Institutional Animal Care and Use Committees of University of Florida in accordance with the 2002 National Institutes of Health Guide for the Care and Use of Laboratory Animals. protocols 201808990 and 202011105.

## Decision letter and Author response

Decision letter https://doi.org/10.7554/eLife.56519.sa1
Author response https://doi.org/10.7554/eLife.56519.sa2

## Additional files

### Supplementary files

- Source data 1. Analgesia and CLAMS data.
- Transparent reporting form

### Data availability

All data generated or analysed during this study are included in the manuscript and supporting files.

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

# Appendix 1

*Appendix 1—table 1* shows the receptor affinities of arylamidomorphinans in opioid transfected cell lines. [a] Competition studies were performed with the indicated compounds against $^{125}$IBNtxA (0.1 nM) in membranes from CHO cells stably expressing the indicated cloned mouse opioid receptors. $K_i$ values were calculated from the $IC_{50}$ values and represent the means ± SEM of at least three independent replications. [b]Literature values (*Váradi et al., 2015a*).

**Appendix 1—table 1.** Receptor affinities of arylamidomorphinans in mouse opioid receptor transfected cell lines.

| Compd. | $K_i$ [nM][a] | | |
|---|---|---|---|
| | **mMOR** | **mKOR** | **mDOR** |
| IBNtxA | 0.11 ± 0.02 | 0.03 ± 0.001 | 0.24 ± 0.05 |
| MP1104 | 0.021 ± 0.00 | 0.0064 ± 0.0 | 0.08 ± 0.01 |
| MP1202 | 0.071 ± 0.031 | 0.11 ± 0.064 | 1.3 ± 0.8 |
| MP1207 | 0.23 ± 0.02 | 0.39 ± 0.05 | 15.62 ± 2.64 |
| MP1208 | 0.34 ± 0.01 | 0.28 ± 0.02 | 19.28 ± 6.48 |
| MP1305 | 0.25 ± 0.02 | 2.5 ±0.3 | 11.7 ± 1.4 |
| MP1601 | 0.2 ± 0.01 | 2.13 ± 0.3 | 5.37 ± 0.9 |
| Morphine | 4.60 ± 1.81[b] | – | – |
| DAMGO | 3.34 ± 0.43[b] | – | – |
| U50, 488h | – | 0.73 ± 0.32[b] | – |
| DPDPE | – | – | 1.39 ± 0.67[b] |

*Appendix 1—table 2* shows the [$^{35}$S]GTPγS Functional assays[a] in transfected cell lines.[a]Efficacy data were determined using an agonist induced stimulation of [$^{35}$S]GTPγS binding assay. Efficacy is represented as $EC_{50}$ (nM) and percent maximal stimulation ($E_{max}$) relative to standard agonist DAMGO (mMOR), DPDPE (mDOR), or U50,488H (mKOR) at 1 µM. To determine the antagonistic properties of a compound, membranes were incubated with 100 nM of the appropriate agonist by varying its concentrations. Results are presented as nM ± SEM from three independent experiments performed as triplicate.[b]Buprenorphine data from *Grinnell et al., 2016*. [c]95% CL [d]Full agonist compared to 1 µM DAMGO.

**Appendix 1—table 2.** [$^{35}$S]GTPγS Functional assays in mouse opioid receptor transfected cell lines.

| Compd. | mMOR | | mKOR | | mDOR | | |
|---|---|---|---|---|---|---|---|
| | $EC_{50}$ (nM) | $E_{max}$ (%) | $EC_{50}$ (nM) | $E_{max}$ (%) | $EC_{50}$ (nM) | $E_{max}$ (%) | $IC_{50}$ (nM) |
| IBNtxA | 0.49 ± 0.12 | 101 ± 3 | 0.22 ± 0.02 | 102 ± 4 | 4.08 ± 0.67 | 95 ± 2 | - |
| MP1104 | 0.21 ± 0.03 | 103 ± 2 | 0.027 ± 0.002 | 104 ± 2 | 0.41 ± 0.11 | 88 ± 0 | - |
| MP1202 | 0.32 ± 0.03 | 68 ± 1 | 0.13 ± 0.02 | 94 ± 5 | 4 ± 1.6 | 71 ± 2 | - |
| MP1207 | 1.29 ±0.65 | 41 ± 1 | 1.52 ± 0.07 | 39.3 ± 1.3 | nd | 10-15% | 27.34±1.95 |
| MP1208 | 1.13 ± 0.05 | 54 ± 0.7 | 1.36 ± 0.23 | 43 ± 0.8 | nd | 10-15% | 11.39±0.3 |
| MP1305 | 0.7 ± 0.1 | 81.2 ± 16 | 7.4 ± 1.8 | 42.2 ± 5.3 | 31.7 ± 3.6 | 22± 0.9 | - |
| MP1601 | 0.5 ± 0.2 | 45 ± 4.6 | 3 ± 0.7 | 72 ± 4.5 | 10 ± 1.6 | 67 ± 3.9 | - |
| DAMGO | 3.4 ± 0.2 | - | - | - | - | - | - |
| U50,488h | - | - | 9.5 ± 1.8 | - | - | - | - |
| DPDPE | - | - | - | - | 16.2 ± 5.1 | - | - |
| Morphine | 14.77±3.9 | 102±5 | - | - | - | - | - |
| Buprenorphine[b] | 1.8(1.3,2.3)[c] | Full agonist[d] | - | - | - | - | - |

*Appendix 1—table 3* shows the energies calculated by computational QM calculations.

**Appendix 1—table 3.** Energies for boat and chair conformations calculated by computational QM calculations for ligands.

| Ligand (Basis set) | Energy(Chair – Boat) HF | Energy(Chair – Boat) kcal/mol | Energy(Chair – Boat) kJ/mol |
|---|---|---|---|
| MP1104 (LanL2DZ) | 0.03671637 | 23.0396323 | 96.3988294 |
| MP1104 (DGDZVP) | 0.03493887 | 21.9242457 | 91.7320032 |
| IBNtxA (LanL2DZ) | −0.0014450 | −0.90674184 | −3.7938475 |
| IBNtxA (DGDZVP) | −0.0016583 | −1.040599833 | −4.353869701 |
| MP1202 (LanL2DZ) | −0.01402513 | −8.8009031585 | −36.822978815 |
| MP1202 (DGDZVP) | −0.011652 | −7.31174652 | −30.59234743997 |

*Appendix 1—table 4* showing the best docking scores of each ligand with a chair or a boat conformation at active state human mu and kappa opioid receptors.

**Appendix 1—table 4.** Best docking scores of each ligand with a chair or a boat conformation at active state human mu and kappa opioid receptors.

| Drugs | MP1104 | IBNtxA | MP1202 | MP1207 | MP1208 | MP1305 | MP1601 | 6'GNTI | MP1209 | MP1210 |
|---|---|---|---|---|---|---|---|---|---|---|
| Docking score of Chair/ hMOR | | -36.22 | -52.42 | -55.04 | -69.74 | -38.63 | -40.61 | | -51.03 | -44.17 |
| Boat /hMOR | -48.28 | -31.36 | -49.44 | -50.81 | -67.72 | -28.83 | -38.23 | | -43.67 | -36.18 |
| Chair /hKOR | | -46.93 | -47.82 | -55.99 | -86.24 | -46.77 | -46.08 | -64.86 | -57.63 | -53.91 |
| Boat /hKOR | -55.06 | -49.51 | -52.42 | -45.9 | -65.88 | -44.87 | -47.54 | | -60.77 | -55.43 |

*Appendix 1—table 5* shows the docking scores for proposed analogs of MP1202, where *m*-iodo group is substituted with a polar moiety (R).

**Appendix 1—table 5.** Docking scores for proposed analogs of MP1202, where *m*-iodo group is substituted with a polar moiety (R).

| Serial | R | Chair score | Boat score | Scores for chair preference |
|---|---|---|---|---|
| 1 | -mNH$_2$ | −55.35 | −58.47 | +3.12 |
| 2 | -mN(CH$_3$)$_2$ | −51.81 | −48.34 | −3.47 |
| 3 | -mOH | −54.81 | −57.34 | +2.53 |
| 4 (MP1207) | **-mCH$_2$NH$_2$** | **−55.99** | **−45.9** | **−10.09** |
| 5 | -mCH$_2$CH$_2$NH$_2$ | −59.42 | −51.79 | −7.63 |
| 6 | -mCH$_2$CH$_2$CH$_2$NH$_2$ | −64.07 | −49.91 | −14.16 |
| 7 | -mCH$_2$CH$_2$CH$_2$CH$_2$NH$_2$ | −52.42 | −49.67 | −2.75 |
| 8 | -mgaunidine | −76.29 | −66.34 | −9.95 |
| 9 (MP1208) | **-mCH$_2$guanidine** | **−86.24** | **−65.88** | **−20.36** |
| 10 | -mCOguanidine | −82.09 | −77.53 | −4.56 |
| 11 | -pCH$_2$NH$_2$ | −57.63 | −60.77 | +3.14 |
| 12 | -mCH$_2$OH | −53.91 | −55.43 | +1.5 |

*Appendix 1—table 6* shows the functional studies at DOR using cAMP inhibition and Tango-arrestin and BRET assays. The functional data of each assay using human delta opioid receptor (hDOR) and mouse delta opioid receptor (mDOR) were normalized to $E_{max}$ of corresponding controls DPDPE or DADLE. Results were analyzed using a three-parameter logistic equation in GraphPad Prism and the data are presented as mean $EC_{50}$($pEC_{50} \pm$ SEM) with $E_{max}\% \pm$ SEM for assays run in triplicate; CTRL.; control compound and nd; results could not be determined because of no measurable arrestin recruitment signal.

**Appendix 1—table 6.** Functional data at hDOR and mDOR.

**Functional data at hDOR and mDOR**

*Continued on next page*

*Appendix 1—table 6 continued*

**Functional data at hDOR and mDOR**

| Compd. | cAMP inhibition | | Arrestin recruitment | |
|---|---|---|---|---|
| | EC$_{50}$,nM (pEC$_{50}$± SEM) | E$_{max}$%± SEM | EC$_{50}$,nM (pEC$_{50}$± SEM) | E$_{max}$%± SEM |
| IBNtxA | 0.43 (9.3 ± 0.03) | 106 ± 1 | 14.1(7.8 ± 0.06) | 224±5 |
| DPDPE (CTRL.) | 0.69 (9.1 ± 0.07) | 100 ± 2 | 2.99(8.5 ± 0.04) | 100±1.5 |
| MP1104 | 0.40 (9.4 ± 0.04) | 99 ± 1.1 | 3.73 (8.4 ± 0.06) | 189 ± 5.5 |
| DADLE (CTRL.) | 0.66(9.2 ± 0.05) | 100 ± 1.3 | 0.349 (9.45 ± 0.10) | 100 ± 3.2 |
| MP1202 | 8.18(8.1 ± 0.06) | 99 ± 2.2 | 18.14(7.7 ± 0.25) | 26 ± 3.1 |
| DADLE (CTRL.) | 1.45(8.8 ± 0.06) | 100 ± 2 | 8.41(8.1 ± 0.08) | 100 ± 2.7 |
| MP1207 | 11.4 (7.9 ± 0.1) | 38 ± 2 | 64.06(7.2 ± 0.23) | 34 ± 3.7 |
| MP1208 | 2.49 (8.6 ± 0.13) | 39 ±1.8 | 3624.0(5.5 ± 0.27) | 62±16.7 |
| DADLE (CTRL.) | 0.48 (9.3 ± 0.05) | 100 ±1.5 | 1.41 (8.8 ± 0.07) | 100±2.3 |
| MP1305 | 74.18 (7.1 ± 0.08) | 71 ± 2.4 | 227.5(6.6 ± 0.06) | 89 ± 2.8 |
| MP1601 | 2.76(8.6 ± 0.06) | 106 ± 2.3 | 86.7(7.1 ± 0.06) | 203±5.4 |
| DPDPE(CTRL.) | 0.69 (9.1 ± 0.07) | 100 ± 2 | 2.99(8.5 ± 0.04) | 100±1.5 |

B. Functional data at mDOR using BRET assays

| Compd. | G protein activation | | Arrestin recruitment | |
|---|---|---|---|---|
| | EC$_{50}$,nM (pEC50± SEM) | E$_{max}$%± SEM | EC$_{50}$,nM (pEC50± SEM) | E$_{max}$%± SEM |
| IBNtxA | 0.48(9.3 ± 0.10) | 108 ± 3.7 | 26.6(7.5 ± 0.2) | 88 ± 6.8 |
| DPDPE(CTRL.) | 2.72(8.5 ± 0.09) | 100 ± 3 | 184.3(6.73 ± 0.1) | 100 ± 6.4 |
| MP1104 | 1.4(8.9 ± 0.06) | 91 ± 1.7 | 26.0(7.6 ± 0.11) | 40 ± 1.7 |
| DPDPE(CTRL.) | 1.3(8.9 ± 0.04) | 100 ± 1.2 | 98.0(7.0 ± 0.04) | 100 ± 1.8 |
| MP1202 | 7.03(8.1 ± 0.15) | 103 ± 4.6 | 524.8(6.3 ± 0.20) | 26 ± 2.3 |
| DPDPE(CTRL.) | 2.19(8.6 ± 0.10) | 100 ± 3.5 | 109.3(6.7 ± 0.06) | 100 ± 3 |
| MP1207 | 116.4(6.9 ± 0.12) | 52 ± 2.3 | nd | nd |
| MP1208 | 26.6(7.6 ± 0.20) | 58 ± 3.3 | nd | nd |
| MP1305 | 40.4(7.4 ± 0.20) | 105 ± 5.9 | nd | nd |
| MP1601 | 6.27(8.2 ± 0.20) | 105 ± 6.7 | 9.0 (8.0 ± 0.33) | 16 ± 1.4 |
| DPDPE(CTRL.) | 0.44(9.3 ± 0.13) | 100 ± 2.5 | 21.8 (7.6 ± 0.04) | 100 ± 1.3 |

*Appendix 1—table 7* shows the functional binding data of the compounds at opioid receptors (hMOR, hKOR and mMOR). The functional data of each assay using human mu opioid receptor (hMOR), human kappa opioid receptor (hKOR), and mouse mu opioid receptor (mMOR) were normalized to E$_{max}$ of corresponding controls. Results were analyzed using a three-parameter logistic equation in GraphPad Prism and the data are presented as mean EC$_{50}$(pEC$_{50}$ ± SEM) with E$_{max}$% ± SEM for assays run in triplicate; CTRL.; control compound.

*Appendix 1—table 7.* Gprotein & arrestin pathway potency and efficacy of ligands at hMOR, hKOR and mMOR.

| Receptors | Compounds | cAMP inhibition (Gi) assay | | Arrestin recruitment (Tango) assay | | Figure |
|---|---|---|---|---|---|---|
| | | EC$_{50}$ (pEC$_{50}$± SEM) | E$_{max}$%± SEM | EC$_{50}$ (pEC$_{50}$± SEM) | E$_{max}$%± SEM | |
| hMOR | MP1104 | 0.074 (10.13 ± 0.05) nM | 94±1 | 0.573(9.24±0.08) nM | 90±2.28 | *Figure 2A–B* |
| | DAMGO | 0.84(9.07±0.08) nM | 100 | 13.9 (7.86±0.06) nM | 100 | |

*Continued on next page*

*Appendix 1—table 7 continued*

| Receptors | Compounds | cAMP inhibition (Gi) assay | | Arrestin recruitment (Tango) assay | | Figure |
|---|---|---|---|---|---|---|
| | | $EC_{50}$ (p$EC_{50}$± SEM) | $E_{max}$%± SEM | $EC_{50}$ (p$EC_{50}$± SEM) | $E_{max}$%± SEM | |
| hKOR | MP1104 | 0.00327 (11.49±0.035) nM | 10±1.64 | 0.03944(10.4±0.06) nM | 117 ± 2 | *Figure 2C–D* |
| | U50488h | 0.089(10.05±0.04) nM | 100 | 3.6(8.44±0.04) nM | 100 | |
| hMOR | MP1202 | 0.077 (10.11 ± 0.06) nM | 92±1.4 | 26.8(7.57±0.1) nM | 53±2 | *Figure 2G–H* |
| | DAMGO | 3.78(8.4±0.06) nM | 100 | 58.8(7.23±0.06) nM | 100 | |
| hKOR | MP1202 | 0.00139 (11.86±0.05) nM, | 97.9±2.5 | 0.0457(10.34±0.05) nM | 101±1 | *Figure 2I–J* |
| | U50,488h | 0.006 (10.2±0.056) nM | 100 | 3.6(8.44±0.04) nM | 100 | |
| hKOR | MP1207 | 0.11 (9.98 ± 0.07) nM | 90±1.7 | 3.97(8.4±0.18) nM | 37±2 | *Figure 3A–B* |
| | MP1208 | 0.14 (9.9 ± 0.07) nM | 96±1.9 | 16.41(7.79±0.14) nM | 48±2.4 | |
| | U50,488h | 0.64(9.2±0.06) nM | 100 | 7.55(8.12±0.06) nM | 100 | |
| hMOR | MP1207 | 0.034 (10.47 ± 0.15) nM | 33±1.4 | nd | nd | *Figure 3E–F* |
| | MP1208 | 0.008 (8.73 ± 0.12) nM | 42±1.3 | nd | nd | |
| | U50,488h | 1.86(8.73±0.06) nM | 100 | | | |
| hKOR | MP1209 | 0.024 (10.6 ± 0.05) nM | 100±1.23 | 0.37(9.43 ± 0.19) nM | 67±3 | *Figure 4E–F* |
| | MP1210 | 0.025 (10.6 ± 0.05) nM | 101.1±1.1 | 1.16 (8.94 ± 0.17) nM | 63±3.3 | |
| | U50,488h | 0.05 (10.29±0.06) nM | 100 | 7.85 (8.11±0.1) nM. | 100 | |
| hMOR | MP1209 | 0.25(9.61 ± 0.04) nM | 98.5±0.96 | nd | nd | *Figure 4G–H* |
| | MP1210 | 0.15(9.81 ± 0.05) nM | 94.6±0.98 | nd | nd | |
| | DAMGO | 0.2 (9.7±0.06) nM | 100 | | | |
| hMOR | Methoxycarbonyl fentanyl | 0.099 (10 ± 0.06) nM | 103±1.8 | 18.94 (7.7±0.1) nM | 194±8 | *Appendix 1— figure 8A–B* |
| | DAMGO | 2.58 (8.59±0.07) nM | 100 | 404.1(6.39±0.05) nM | 100 | |
| hMOR | MP102 | 19.7(7.7 ± 0.08) nM | 87±2.66 | nd | nd | *Appendix 1— figure 8F–G* |
| | DAMGO | 2.58 (8.59±0.07) nM | 100 | | | |
| hMOR | Morphine | 21.8 (7.66 ± 0.08) nM | 97±3.06 | 888 (6.05±0.25) nM | 25.32±4.4 | |
| | DAMGO | 8.1 (8.09 ± 0.06) nM | 100 | 140.7 (6.85±0.08) nM | 100 | |
| | Buprenorphine | 0.7 (10.14 ± 0.11) pM | 75±2.36 | 1.79 (8.75±0.13) nM | 43±2 | |
| | DAMGO | 4.36 (8.36 ± 0.08) nM | 100 | 22 (7.64±0.063) nM | 100 | |
| Receptors | Compounds | G protein activation (BRET) assay | | Arrestin recruitment (BRET) assay | | Figure |
| | | $EC_{50}$ (p$EC_{50}$± SEM) | $E_{max}$%± SEM | $EC_{50}$ (p$EC_{50}$± SEM) | $E_{max}$%± SEM | |
| mMOR | MP1207 | 3.61 (8.44 ± 0.26) nM | 42±2.6 | nd | nd | *Figure 3I–J* |
| | MP1208 | 2.27 (8.64 ± 0.29) nM | 41±3 | nd | nd | |
| | DAMGO | 3.27(8.49±0.08) nM | 100 | | | |
| | DAMGO | 9.09 (8.04 ± 0.13) nM | 100 | | | |
| mMOR | Morphine | 9.09 (8.04 ± 0.13) nM | 109±3.1 | | | |
| | Buprenorphine | 1.17 (8.93 ± 0.14) nM | 59±2.4 | | | |

*Appendix 1—table 8* shows the functional binding data of the compounds at rodent opioid receptors (mMOR and rKOR). The functional data of each assay using mouse mu opioid receptor (mMOR) and rat kappa opioid receptor (rKOR) were normalized to $E_{max}$ of corresponding controls. Results were analyzed using a three-parameter logistic equation in GraphPad Prism and the data are presented as mean $EC_{50}$($pEC_{50} \pm$ SEM) with $E_{max}\% \pm$ SEM for assays run in triplicate; CTRL.; control compound.

**Appendix 1—table 8.** Gprotein & arrestin pathway potency and efficacy of ligands at mMOR and rKOR.

| Receptors | Compound | G protein activation (BRET) assay | | Arrestin recruitment (BRET) assay | | Figure |
|---|---|---|---|---|---|---|
| | | $EC_{50}$ ($pEC_{50}\pm$ SEM) | $E_{max}\%\pm$ SEM | $EC_{50}$ ($pEC_{50}\pm$ SEM) | $E_{max}\%\pm$ SEM | |
| mMOR | MP1104 | 0.66 (9.2±0.11) nM | 62±2 | 0.285 (9.55±0.12) nM | 53 ± 1.6 | *Appendix 1—figure 1A* |
| | DAMGO | 7.3 (8.1±0.07) nM | 100 | 31.5 (7.5±0.06) nM | 100 | |
| | IBNtxA | 0.054(10.3 ± 0.02) nM | 59±2.2 | 11.32(4.9±0.3) µM | 75±13.9 | *Appendix 1—figure 1B* |
| | DAMGO | 12.9(7.8±0.06) nM | 100 | 0.77(6.11±0.16) µM | 100 | |
| | MP1202 | 0.63 (9.2±0.09) nM | 61±1.5 | 3140 (5.5±0.28) nM | 53 ± 1.6 | *Appendix 1—figure 1C* |
| | DAMGO | 6.31 (8.2±0.07) nM | 100 | 114 (6.9±0.15) nM | 100 | |
| | MP1207 | 3.61 (8.44 ± 0.26) nM | 42±2.6 | nd | nd | *Appendix 1—figure 1D* |
| | MP1208 | 2.27 (8.64 ± 0.29) nM | 41±3 | nd | nd | |
| | DAMGO | 3.27(8.49±0.08) nM | 100 | | | |
| | MP1305 | 0.74(9.1 ± 0.12) nM | 72±2 | nd | nd | |
| | DAMGO | 3.3(8.4±0.08) nM | 100 | | | *Appendix 1—figure 1E* |
| | MP1601 | 2.2(8.6 ± 0.3) nM | 49±3 | nd | nd | |
| | DAMGO | 0.9(9.0±0.1) nM | 100 | | | |
| rKOR | MP1104 | 0.073 (10.14±0.04) nM | 93±1 | 1.14(8.94±0.07) nM | 89 ± 2.6 | *Appendix 1—figure 2A* |
| | U50, 488h | 1.05(8.98±0.03) nM | 100 | 110(6.95±0.05) nM | 100 | |
| | IBNtxA | 0.064(10.2 ± 0.1) nM | 101±2.3 | 1.23(8.9±0.02) nM | 133±7.3 | *Appendix 1—figure 2B* |
| | U50, 488h | 27.7(7.6±0.01)nM | 100 | 345.6(6.4±0.03) nM | 100 | |
| | MP1202 | 0.134(9.87±0.09) nM | 104±2 | 1.44(8.84±0.25) nM | 77 ± 5 | *Appendix 1—figure 2C* |
| | U50, 488h | 4.79 (8.32±0.07) nM | 100 | 235(6.63±0.18) nM | 100 | |
| | MP1207 | 1.13 (8.95 ± 0.31) nM | 32±2.3 | nd | nd | *Appendix 1—figure 2D* |
| | MP1208 | 1.1(8.97 ± 0.29) nM | 40±2.4 | nd | nd | |
| | U50, 488h | 82.6(7.1±0.09) nM | 100 | | | |
| | MP1305 | 5.04 (8.3 ± 0.32) nM | 35±3 | nd | nd | |
| | U50, 488h | 40.3(7.4±0.07) nM | 100 | | | *Appendix 1—figure 2E* |
| | MP1601 | 8.9(8.1 ± 0.12) nM | 75±3 | 56.5(7.2±0.23) nM | 37±3 | |
| | U50, 488h | 26.7(7.6±0.09) nM | 100 | 169.6(6.7±0.07) nM | 100 | |
| mMOR | IBNtxA | 0.054(10.3 ± 0.02) nM | 59±2.2 | 11.32(4.9±0.3) µM | 75±13.9 | *Figure 3F–G* |
| | DAMGO | 12.9(7.8±0.06) nM | 100 | 0.77(6.11±0.16) µM | 100 | |
| rKOR | IBNtxA | 0.064(10.2 ± 0.1) nM | 101±2.3 | 1.23(8.9±0.02) nM | 133±7.3 | *Figure 3H–I* |
| | U50, 488h | 27.7(7.6±0.01) nM | 100 | 345.6(6.4±0.03) nM | 100 | |
| mMOR | Methoxycarbonyl fentanyl | 0.21(9.7 ± 0.04) nM | 109±0.94 | 1.96(8.71±0.11) nM | 63±1.8 | *Appendix 1—figure 7C–D* |
| | DAMGO | 1.73(8.76±0.05) nM | 100 | 110.1(6.96±0.06) nM | 100 | |
| | MP102 | 404.5 (6.39 ± 0.2) nM | 78±7 | nd | nd | *Appendix 1—figure 7E* |
| | DAMGO | 3.27(8.5±2) nM | 100 | | | |

*Appendix 1—table 9* shows the functional binding data of the compounds at human opioid receptors (hMOR, hKOR and Y312W-hKOR). The functional data of each assay using human mu opioid receptor (hMOR), human kappa opioid receptor (hKOR), and Y312W human kappa opioid receptor mutant (Y312W-hKOR) were normalized to $E_{max}$ of corresponding controls. Results were analyzed using a three-parameter logistic equation in GraphPad Prism and the data are presented as mean $EC_{50}(pEC_{50} \pm SEM)$ with $E_{max}\% \pm SEM$ for assays run in triplicate; CTRL.; control compound.

**Appendix 1—table 9.** cAMP & arrestin potency and efficacy at hMOR, hKOR and Y312W-hKOR of select ligands.

| Receptors | Compounds | cAMP inhibition (Gi) assay | | Arrestin recruitment (Tango) assay | | Figure |
|---|---|---|---|---|---|---|
| | | $EC_{50}$ (pEC$_{50}\pm$ SEM) | $E_{max}\%\pm$ SEM | $EC_{50}$ (pEC$_{50}\pm$ SEM) | $E_{max}\%\pm$ SEM | |
| hMOR | IBNtxA | 0.07(10.2 ± 0.006) nM | 95±2 | 5.86(8.2±0.002) nM | 29±0.02 | *Appendix 1— figure 3A-B* |
| | DAMGO | 0.99(9.0±0.007) nM | 100 | 14.16 (7.9±0.003) nM | 100 | |
| hKOR | IBNtxA | 2.46 (11.6 ± 0.037) pM | 101±1.79 | 0.013 (10.9±0.07) nM | 109±1.8 | *Appendix 1— figure 3C-D* |
| | U50488h | 0.089(10.05±0.042) nM | 100 | 3.63(8.4±0.03) nM | 100 | |
| Y312W-hKOR | MP1202 | 0.21 (10.69 ± 0.07) pM | 101±1.5 | 3.4(8.5±0.14) nM | 55±2.6 | *Appendix 1— figure 7A,C* |
| | U50,488h | 2.7(8.56±0.06)nM | 100 | 0.63(7.2±0.06) nM | 100 | |
| WT-hKOR | MP1202 | | | 0.0457 (10.34±0.05) nM | 101±1 | *Appendix 1— figure 7B,D* |
| | U50,488h | | | 3.6(8.44±0.04) nM | 100 | |
| WT-hMOR | MP1202 | | | 26.8(7.57±0.1) nM | 53±2 | |
| | DAMGO | | | 58.8(7.23±0.06) nM | 100 | |
| Y312W-hKOR | MP1207 | 0.16 (9.8 ± 0.08) nM | 95±1.8 | nd | nd | *Appendix 1— figure 7E,G* |
| | MP1208 | 0.36 (10.44 ± 0.05) pM | 97±1 | nd | nd | |
| | U50,488h | 0.92(9.04±0.04)nM | 100 | 14(7.85±0.1) nM | 100 | |
| WT-hKOR | MP1207 | 3.97(8.4±0.18) nM | 37±2 | nd | nd | *Appendix 1— figure 7F* |
| | MP1208 | 16.41(7.79±0.14) nM | 48±2.4 | nd | nd | |
| | U50,488h | 7.55(8.12±0.06) nM | 100 | | | |
| hKOR | MP1305 | 0.72 (9.14 ± 0.05) nM | 102±1.2 | 25.72(7.6±0.04) nM | 75±1 | *Appendix 1— figure 9A-B* |
| | U50, 488h | 0.076(10.11±0.04) nM | 100 | 3.6(8.44±0.04) nM | 100 | |
| hMOR | MP1305 | 0.12(9.9 ± 0.12) nM | 69±1.8 | 16.4(7.8±0.3) nM | 21±3 | *Appendix 1— figure 9C-D* |
| | DAMGO | 3.88(8.4±0.05) nM | 100 | 168.1 (6.77±0.09) nM | 100 | |
| hKOR | MP1601 | 0.17(9.76 ± 0.05) nM | 109±1 | 3.23(8.49±0.07) nM | 107±3 | *Appendix 1— figure 9E-F* |
| | U50, 488h | 0.077(10.11±0.04) nM | 100 | 3.6(8.44±0.04) nM | 100 | |

*Continued on next page*

*Appendix 1—table 9 continued*

| Receptors | Compounds | cAMP inhibition (Gi) assay | | Arrestin recruitment (Tango) assay | | Figure |
|---|---|---|---|---|---|---|
| | | EC$_{50}$ (pEC$_{50}$± SEM) | E$_{max}$%± SEM | EC$_{50}$ (pEC$_{50}$± SEM) | E$_{max}$%± SEM | |
| hMOR | MP1601 | 0.1(9.99 ± 0.09) nM | 69±1 | 4.02(8.4±0.2) nM | 27±2 | *Appendix 1—figure 9G-H* |
| | DAMGO | 3.88(8.4±0.05) nM | 100 | 168.1 (6.77±0.09) nM | 100 | |

*Appendix 1—table 10* shows the bias analysis of ligands at human opioid receptors (*nd: not determined; ns compared to control*).

**Appendix 1—table 10.** Bias analysis of ligands at human opioid receptors.

| Drug | Receptor | cAMP | Tango | ΔlogRA | ΔlogRA | ΔΔlog RA | Bias factor | Figure |
|---|---|---|---|---|---|---|---|---|
| | | Log RA | LogRA | cAMP | Tango | (cAMP-Tango) | (cAMP-Tango) | |
| U50,488h | hKOR | 10.09±0.035 | 8.29±0.019 | 0±0.049 | 0±0.027 | 0±0.056 | 1 | *Appendix 1—figure 4A* |
| MP1104 | hKOR | 11.44±0.034 | 10.45±0.019 | 1.35±0.048 | 2.16±0.027 | -0.81±0.055 | 0.15 | |
| U50,488h | hKOR | 10.22±0.051 | 8.44±0.004 | 0±0.072 | 0±0.006 | 0±0.070 | 1 | *Appendix 1—figure 4A* |
| MP1202 | hKOR | 11.88±0.050 | 10.30±0.004 | 1.66±0.072 | 1.856±0.006 | -0.19±0.072 | nd | |
| U50,488h | hKOR | 10.06±0.036 | 8.36±0.011 | 0±0.051 | 0±0.015 | 0±0.050 | 1 | *Appendix 1—figure 4A* |
| IBNtxA | hKOR | 11.59±0.036 | 10.89±0.010 | 1.53±0.050 | 2.53±0.015 | -1.002±0.052 | 0.10 | |
| U50,488h | hKOR | 9.175±0.004 | 8.14±0.007 | 0±0.005 | 0±0.010 | 0±0.012 | 1 | *Appendix 1—figure 4A* |
| MP1207 | hKOR | 9.902±0.004 | 7.97±0.024 | 0.73±0.006 | -0.176±0.025 | 0.903±0.026 | 8.00 | |
| U50,488h | hKOR | 9.176±0.004 | 8.14±0.008 | 0±0.005 | 0±0.011 | 0±0.012 | 1 | *Appendix 1—figure 4A* |
| MP1208 | hKOR | 9.83±0.004 | 7.447±0.020 | 0.65±0.005 | -0.697±0.022 | 1.35±0.022 | 22 | |
| U50,488h | hKOR | 10.42±0.06 | 8.37±0.22 | 0±0.084 | 0±0.31 | 0±0.32 | 1 | *Appendix 1—figure 4A* |
| MP1209 | hKOR | 10.55±0.06 | 8.735±0.220 | 0.13±0.084 | 0.368±0.31 | -0.238±0.32 | nd | |
| U50,488h | hKOR | 10.39±0.06 | 8.096±0.084 | 0±0.081 | 0±0.12 | 0±0.144 | 1 | *Appendix 1—figure 4A* |
| MP1210 | hKOR | 10.59±0.06 | 8.576±0.156 | 0.2±0.081 | 0.48±0.18 | -0.28±0.19 | nd | |
| U50,488h | hKOR | 10.010±0.012 | 8.507±0.026 | 0±0.017 | 0±0.037 | 0±0.041 | 1 | *Appendix 1—figure 4A* |
| MP1305 | hKOR | 9.22±0.012 | 7.17±0.025 | -0.79±0.017 | -1.341±0.036 | 0.55±0.04 | 4 | |
| U50,488h | hKOR | 10.03±0.014 | 8.414±0.009 | 0±0.020 | 0±0.012 | 0±0.023 | 1 | *Appendix 1—figure 4A* |
| MP1601 | hKOR | 9.85±0.014 | 8.51±0.009 | -0.19±0.020 | 0.097±0.012 | -0.282±0.023 | 0.52 | |
| DAMGO | hMOR | 9.102±0.009 | 7.925±0.013 | 0±0.012 | 0±0.019 | 0±0.022 | 1 | *Appendix 1—figure 4B* |
| MP1104 | hMOR | 10.12±0.009 | 9.18±0.013 | 1.018±0.012 | 1.26±0.019 | -0.24±0.022 | 0.58 | |

*Continued on next page*

*Appendix 1—table 10 continued*

| Drug | Receptor | cAMP | Tango | ΔlogRA | ΔlogRA | ΔΔlog RA | Bias factor | Figure |
|---|---|---|---|---|---|---|---|---|
| | | Log RA | LogRA | cAMP | Tango | (cAMP-Tango) | (cAMP-Tango) | |
| DAMGO | hMOR | 8.44±0.011 | 7.18±0.001 | 0±0.015 | 0±0.001 | 0±0.015 | 1 | *Appendix 1—figure 4B* |
| MP1202 | hMOR | 9.99±0.012 | 7.24±0.002 | 1.55±0.016 | 0.058±0.002 | 1.49±0.016 | 31 | |
| DAMGO | hMOR | 8.86±0.075 | 7.82±0.060 | 0±0.106 | 0±0.084 | 0±0.135 | 1 | *Appendix 1—figure 4B* |
| IBNtxA | hMOR | 9.99±0.081 | 7.572±0.25 | 1.14±0.11 | -0.25±0.25 | 1.38±0.28 | 24 | |
| DAMGO | hMOR | 8.62±0.054 | 6.78±0.010 | 0±0.076 | 0±0.014 | 0±0.077 | 1 | *Appendix 1—figure 4B* |
| MP1305 | hMOR | 9.79±0.053 | 6.89±0.048 | 1.17±0.076 | 0.109±0.049 | 1.059±0.090 | 11 | |
| DAMGO | hMOR | 8.40±0.001 | 6.78±0.009 | 0±0.002 | 0±0.012 | 0±0.013 | 1 | *Appendix 1—figure 4B* |
| MP1601 | hMOR | 9.84±0.002 | 7.65±0.035 | 1.44±0.002 | 0.87±0.036 | 0.57±0.036 | 4 | |
| U50,488h | WT hKOR | 10.22±0.051 | 8.44±0.004 | 0±0.072 | 0±0.006 | 0±0.070 | 1 | *Appendix 1—figure 7L* |
| MP1202 | WT hKOR | 11.88±0.050 | 10.30±0.004 | 1.66±0.072 | 1.856±0.006 | -0.19±0.072 | nd | |
| U50,488h | Y312W-hKOR | 8.568±0.005 | 7.559±0.051 | 0±0.007 | 0±0.072 | 0±0.073 | 1 | *Appendix 1—figure 7L* |
| MP1202 | Y312W-hKOR | 10.63±0.005 | 8.091±0.051 | 2.062±0.007 | 0.532±0.073 | 1.53±0.073 | 34 | |
| DAMGO | hMOR | 8.563±0.004 | 5.037±0.176 | 0±0.006 | 0±0.249 | 0±0.249 | 1 | *Appendix 1—figure 8J* |
| Methoxycarbonyl fentanyl | hMOR | 10.020±0.004 | 7.546±0.184 | 1.457±0.006 | 2.509±0.255 | -1.052±0.255 | 0.09 | |

*Appendix 1—table 11* shows the bias analysis of ligands at rodent opioid receptors (nd:not determined; ns compared to control).

**Appendix 1—table 11.** Bias analysis of ligands at rodent opioid receptors.

| Drug | Receptor | BRET-G | BRET-Arr | ΔlogRA | ΔlogRA | ΔΔlog RA | Bias factor | Figure |
|---|---|---|---|---|---|---|---|---|
| | | Log RA | LogRA | BRET-G | BRET-arr | (BRETG-BRETArr) | (BRETG-BRETArr) | |
| U50,488h | rKOR | 9.003±0.044 | 7.14±0.034 | 0±0.062 | 0±0.047 | 0±0.078 | 1 | *Appendix 1—figure 4C* |
| MP1104 | rKOR | 10.07±0.047 | 8.84±0.033 | 1.067±0.064 | 1.702±0.047 | -0.64±0.079 | 0.23 | |
| U50,488h | rKOR | 8.312±0.077 | 6.798±0.168 | 0±0.109 | 0±0.238 | 0±0.261 | 1 | *Appendix 1—figure 4C* |
| MP1202 | rKOR | 9.906±0.088 | 8.791±0.171 | 1.594±0.117 | 1.993±0.239 | -0.399±0.27 | nd | |
| U50,488h | rKOR | 7.55±0.103 | 5.988±0.222 | 0±0.146 | 0±0.313 | 0±0.345 | 1 | *Appendix 1—figure 4C* |
| IBNtxA | rKOR | 10.19±0.141 | 8.846±0.266 | 2.64±0.175 | 2.858±0.346 | -0.218±0.388 | nd | |

*Continued on next page*

*Appendix 1—table 11 continued*

| Drug | Receptor | BRET-G | BRET-Arr | ΔlogRA | ΔlogRA | ΔΔlog RA | Bias factor | Figure |
|---|---|---|---|---|---|---|---|---|
| | | Log RA | LogRA | BRET-G | BRET-arr | (BRETG-BRETArr) | (BRETG-BRETArr) | |
| U50,488h | rKOR | 7.593±0.012 | 6.783±0.005 | 0±0.016 | 0±0.007 | 0±0.018 | 1 | *Appendix 1—figure 4C* |
| MP1601 | rKOR | 7.917±0.018 | 7.113±0.052 | 0.324±0.052 | 0.33±0.052 | -0.006±0.057 | nd | |
| | | | | | | | | |
| DAMGO | mMOR | 8.56±0.12 | 7.873±0.06 | 0±0.176 | 0±0.09 | 0±0.197 | 1 | *Appendix 1—figure 4C* |
| MP1104 | mMOR | 8.917±0.12 | 9.196±0.06 | 0.357±0.173 | 1.323±0.09 | -0.966±0.195 | 0.11 | |
| DAMGO | mMOR | 8.713±0.013 | 7.291±0.044 | 0±0.019 | 0±0.063 | 0±0.066 | 1 | *Appendix 1—figure 8J* |
| Methoxycarbonyl | mMOR | 9.697±0.013 | 8.464±0.045 | 0.984±0.019 | 1.173±0.063 | -0.189±0.066 | nd | |
| fentanyl | | | | | | | | |

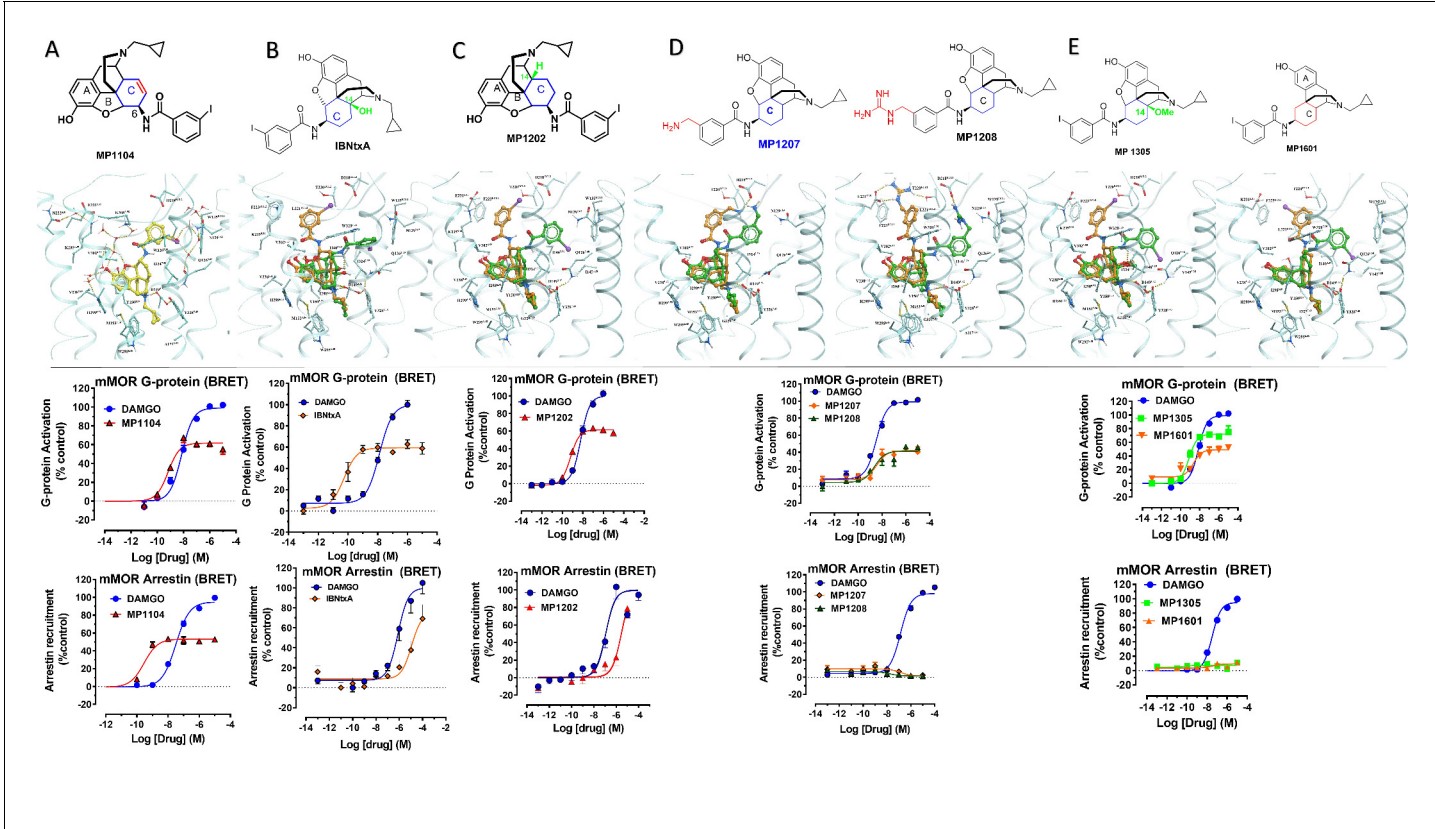

**Appendix 1—figure 1.** Characterization of ligands (MP1104, IBNtxA, MP1202, MP1207, MP1208, MP1305, and MP1601) at mouse mu opioid receptor (mMOR) using BRET assays-chemical structure, docking in MOR, G protein activity, and arrestin recruitment. (**A**) MP1104 targets the TM2-TM3 region and is arrestin-biased at mMOR: The preferred docking pose of MP1104 (boat form, yellow stick) at an active state of MOR is shown. Ring C of MP1104 in boat form forces the iodophenyl moiety to reside in a region between TM2-TM3. MP1104 (red) is a partial agonist in mMOR measuring G protein activation (N = 4) and arrestin recruitment (N = 4) compared to DAMGO (blue) in BRET assays. (**B**) IBNtxA targets the TM5-ECL2 region while showing reduced arrestin potency at mMOR compared to DAMGO: The docking poses of IBNtxA (chair form, orange stick) at an active state of MOR is shown. The saturated ringC in IBNtxA leads to interaction of the ligand in the ECL2 and TM5 region, which leads to a preference of chair form in our docking studies. At mMOR, it is a partial agonist for G protein signaling (N = 4) compared to DAMGO (blue) while showing reduced arrestin potency compared to DAMGO (N = 4) in BRET assays. (**C**) MP1202 targets the TM5-ECL2 region and shows reduced arrestin potency at mMOR compared to DAMGO. The preferred docking pose of MP1202 (chair form, orange stick) at an active state of MOR. The saturated ringC in MP1202 leads to interaction of the ligand in the ECL2 and TM5 region. MP1202 (red) is a partial agonist for G-signaling (N = 5) in mMOR and reduced arrestin potency (N = 5) compared to DAMGO (blue) in BRET assays. (**D**) MP1207 and MP1208 target the TM5-ECL2 region and show no arrestin recruitment at mMOR: Docking results showed that *m*-aminomethyl (MP1207) or *m*-guanidinomethyl (MP1208) moieties (replacing an iodo group in MP1202) forced these compounds in chair form preferred confirmation at MOR (chair form in orange stick and boat form in green stick). MP1207 (orange) and MP1208 (green) are partial agonists at mMOR in BRET assays (N = 3) compared to DAMGO (blue). At MOR, chair forms of MP1207 and MP1208 introduce additional interactions between *m*-amino or *m*-guanidino group and D218[ECL2] and T220[ECL2]. No arrestin recruitment was observed for both agonists (N = 3). (**E**) MP1305 and MP1601 target the TM5-ECL2 region and show no arrestin recruitment at mMOR. Chair and boat forms of MP1305 and MP1601 at an active state MOR (in cyan) are shown. Docking modes of MP1305 and MP1601 are analogous and they both maintain chair confirmation at MOR. In BRET assays using mMOR, MP1305 (green) and MP1601 (red) are partial agonists for G-signaling (N = 3) compared to DAMGO (blue) with no measurable arrestin recruitment (N = 3). In summary, targeting TM5-ECL2 and ring C taking chair form leads to reduced arrestin signaling at MOR in both rodents as well as human receptors with analogs. See *Appendix 1—table 8* for values, *Appendix 1—figure 4* and *Appendix 1—table 11* for bias calculations.

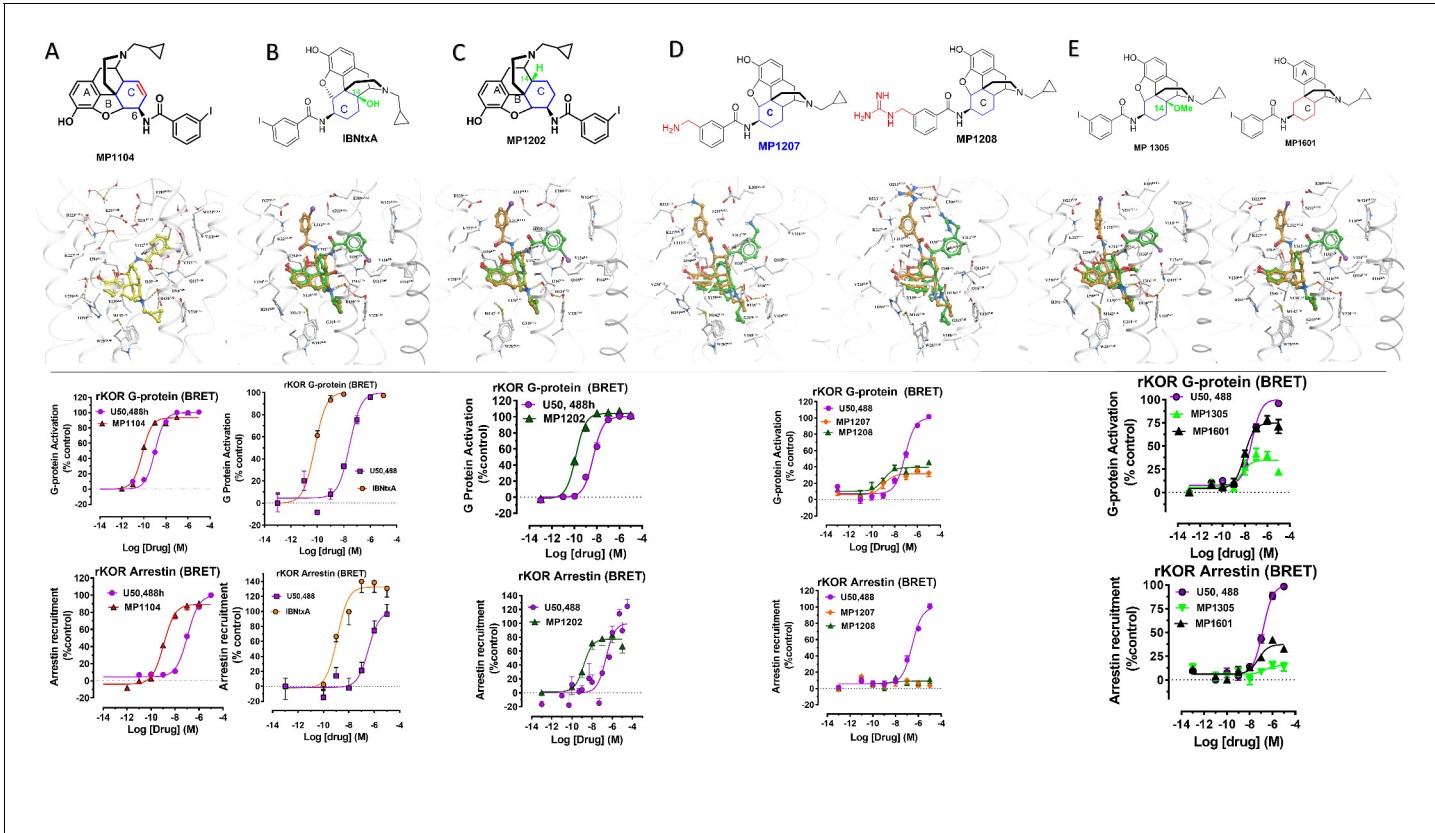

**Appendix 1—figure 2.** Characterization of ligands (MP1104, IBNtxA, MP1202, MP1207, MP1208, MP1305 and MP1601) at rat kappa opioid receptor (rKOR) using BRET assays-chemical structures, docking in MOR, G protein activity, and arrestin recruitment. **A) MP1104** targets the TM2-TM3 region and is arrestin biased at rKOR: The preferred docking pose of **MP1104** (boat form, yellow stick) at an active state of KOR is shown. The iodophenylamido moiety aligns in the hydrophobic pocket between TM2 and TM3 in KOR, a cavity lined with V118$^{2.63}$, W124$^{ECL1}$, and L135$^{3.29}$. In addition, polar residue Q115$^{2.60}$ adopts a slightly different conformation which allows more room for the hydrophobic moiety. Such increase in hydrophobic nature of the KOR binding pocket may well be associated yielding the best docking scores of **MP1104** in its boat conformation. **MP1104** (red) is a full agonist in rKOR in BRET assays measuring G protein activation (N = 6) and arrestin recruitment (N = 6) compared to U50,488H (purple). **(B) IBNtxA** prefers the boat form at KOR and is not biased for any pathway at mKOR: The docking poses of **IBNtxA** (chair form, orange stick) and (boat form, green stick) at an active state of KOR are shown. At rKOR, the iodophenylamido moiety of **IBNtxA** aligns in the hydrophobic pocket between TM2 and TM3 in KOR, a cavity lined with V118$^{2.63}$, W124$^{ECL1}$ and L135$^{3.29}$. In BRET assays using rKOR, **IBNtxA** is a full agonist for G protein (N = 3) as well as Arrestin signaling (N = 3) compared with U50,488H (purple). **(C) MP1202** prefers the boat form at KOR and is not biased for any pathway at rKOR. The preferred docking pose of **MP1104** (boat form, yellow stick) and **MP1202** (chair form, orange stick) at an active state of KOR. The iodophenylamido moiety aligns in the hydrophobic pocket between TM2 and TM3 in KOR, a cavity lined with V118$^{2.63}$, W124$^{ECL1}$, and L135$^{3.29}$. In addition, polar residue Q115$^{2.60}$ adopts a slightly different conformation allowing more room for the hydrophobic moiety. Such increase in hydrophobic nature of the KOR binding pocket may well be associated yielding the best docking scores of both **MP1104, IBNtxA** and **MP1202** with their boat conformation. **MP1202** (green) is a full agonist in rKOR measuring G protein activation (N = 3) and arrestin recruitment (N = 9) compared to U50,488H (purple) in BRET assays. **(D)** *m*-Aminomethyl (**MP1207**) and *m*-guanidinomethyl (**MP1208**) analogs prefer the chair conformation and target the TM5-ECL2 region and show no arrestin recruitment at KOR. Docking results showed that *m*-amino methyl (**MP1207**) or *m*-guanidinomethyl (**MP1208**) moieties (replacing an iodo group in **MP1202**) forced these compounds in chair form preferred confirmation at KOR (chair form in orange stick and boat form in green stick). Unlike boat **MP1202**, chair **MP1207** at KOR may form a new salt bridge interaction between amino group and D223$^{5.35}$ and E209A pulling the amidophenyl moiety away from the hydrophobic pocket between TM2 and TM3. Likewise, chair **MP1208** forms salt bridge interactions between guanidino group and D223$^{5.35}$ as well as with E209$^{ECL2}$. **MP1207** (orange) and **MP1208** (green) are partial agonists for G-signaling (N = 3) at rKOR in BRET assays compared to U50,488H (purple). No arrestin recruitment was observed for both agonists (N = 5) against U50,488H as a control at rKOR. **(E) MP1305** showed no arrestin recruitment and **MP1601** was not biased for any pathway at rKOR: Chair and boat forms of MP130) and **MP1601** at an active state KOR (in gray) are shown. Both **MP1305** (green) and **MP1601** (black) are partial agonists for G-signaling (N = 3) at rKOR in BRET assays compared to U50,488H (blue). No arrestin recruitment was observed for **MP1305** (N = 6) while **MP1601** was a partial agonist for arrestin signaling pathway (N = 6). In summary, targeting TM5-ECL2 and ring C taking chair form leads to preference for G protein pathway at KOR in both rodents as well as human receptors with analogs. See *Appendix 1—table 8* for values, *Appendix 1—figure 4* and *Appendix 1—table 11* for bias calculations.

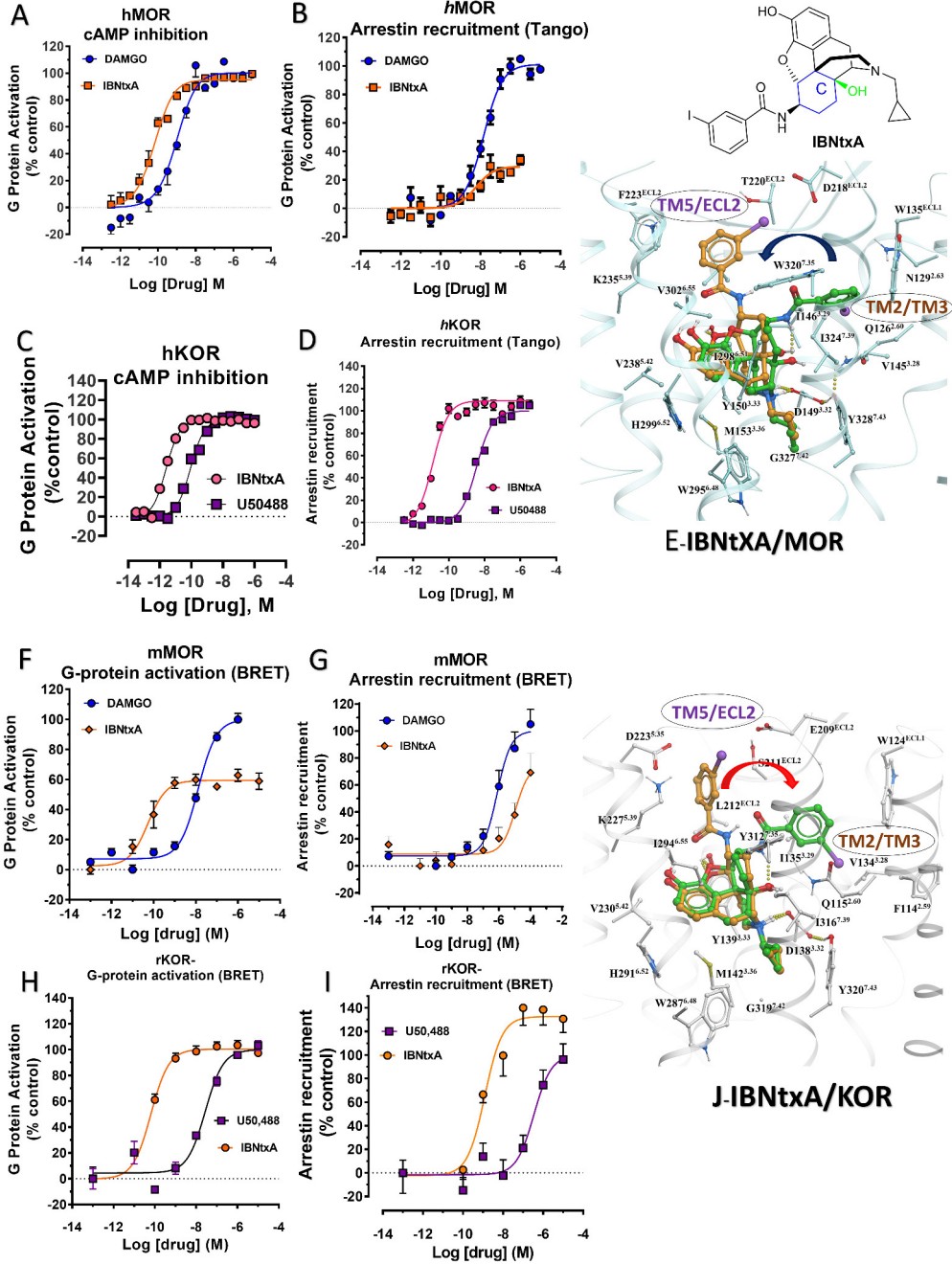

**Appendix 1—figure 3.** IBNtxA is arrestin biased at KOR but shows G protein biased agonism at MOR. (**A–B**) At hMOR, IBNtxA (orange) is a G protein biased agonist compared with DAMGO (blue) (N = 4). (**C–D**) At hKOR, IBNtxA (pink) is a full agonist in cAMP inhibition (N = 3) and Tango-arrestin recruitment assays (N = 3) compared to U50,488H (purple). (**F–G**) However, at mMOR, it is a partial agonist in G-signaling assays (N = 4) and shows reduced arrestin potency (N = 4) compared to DAMGO (blue) in BRET assays. (**H–I**) In BRET assays using rKOR, IBNtxA is a full agonist in both G-signaling (N = 3) and arrestin signaling assays (N = 3) compared with U50,488H (purple). (**E and J**) The docking poses of IBNtxA (chair form, orange stick) and (boat form, green stick) at an active state of MOR and KOR are shown. At MOR, the saturated ring C in IBNtxA leads to interaction of the ligand in the ECL2 and TM5 region leading to a preference of chair form shown by a red arrow. At KOR, the iodophenylamido moiety of IBNtxA aligns in the hydrophobic pocket between TM2 and TM3 in KOR, a cavity lined with V118$^{2.63}$, W124$^{ECL1}$ and L135$^{3.29}$ similar to MP1202. This flip of ring

*Appendix 1—figure 3 continued on next page*

*Appendix 1—figure 3 continued*

C conformation from chair to boat is shown by a blue arrow. Preference for chair form correlates with G protein bias while preference for boat form correlates with increased arrestin signaling or no preference for any pathway. See *Appendix 1—table 9* for values, *Appendix 1—figure 4* and *Appendix 1—tables 10–11* for bias calculations.

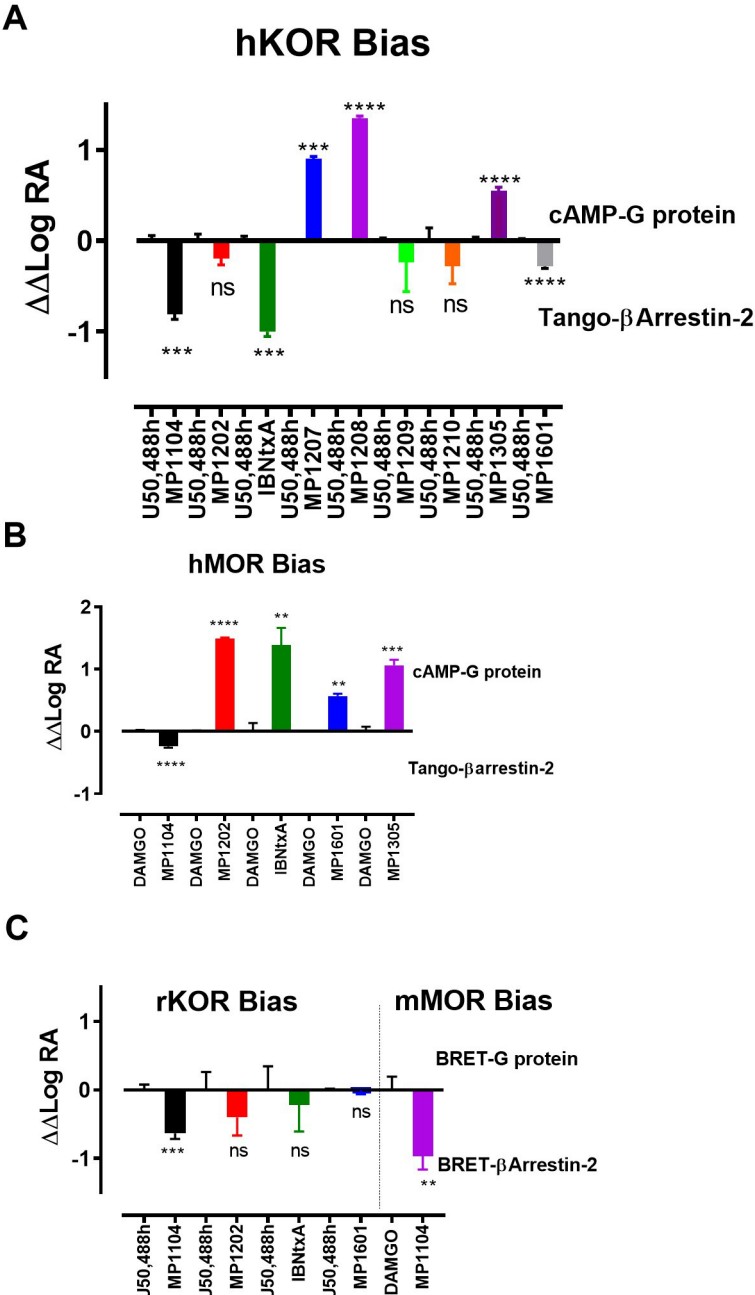

**Appendix 1—figure 4.** Bias plots for ligands at hKOR (**A**), hMOR (**B**) using cAMP and Tango assay and at rKOR /mMOR (**C**) using BRET assays. Bias analysis for signaling was performed as described in Materials and methods. Data analyzed against DAMGO and/or U50,488h for each ligand using unpaired t-test with Welch's correction. At hKOR, MP1104 (**p=0.0015); IBNtxA (***p=0.0002) showed preference for arrestin over G-signaling while MP1209, 1210, 1305 and 1601 showed no

*Appendix 1—figure 4 continued on next page*

*Appendix 1—figure 4 continued*

preference for any signaling pathway. MP1209 (***p=0.0001) and MP1208 (****p<0.0001) were G-biased. At hMOR, MP1104 (*p=0.0427) preferred arrestin pathway while MP1202 (****p=<0.0001); IBNtxA (**p=0.0089); MP1601 (**p=0.0017) and MP1305 (***p=0.001) were G-biased. At rKOR, MP1104 (***p=0.0002) preferred arrestin pathway while IBNtxA, MP1202, 1601 and 1305 showed no preference for any signaling pathway. At mMOR, MP1104 (**p=0.0021) showed preference for arrestin pathway. To summarize bias factors suggest that in spite of differences in assays and species tested, ligands which engage TM5-ECL2 region of MOR/KOR and where ringC takes chair conformation.

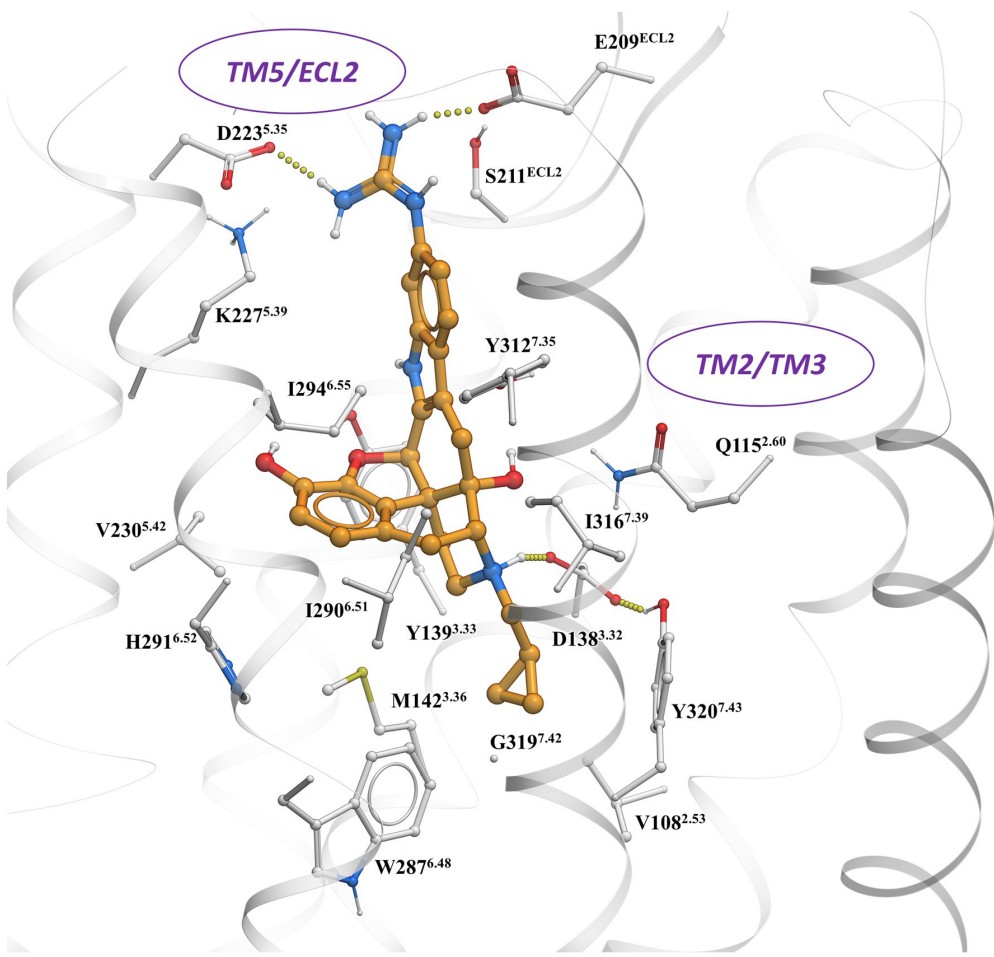

**Appendix 1—figure 5.** The preferred docking pose of known KOR biased ligand 6'GNTI (chair form, orange stick) at an active state of KOR with the guanidino group engaging a region between TM5 and ECl2. Note: possible engagement of residues D223 and E209 similar to MP1207 and MP1208 in TM5-ECL2 region.

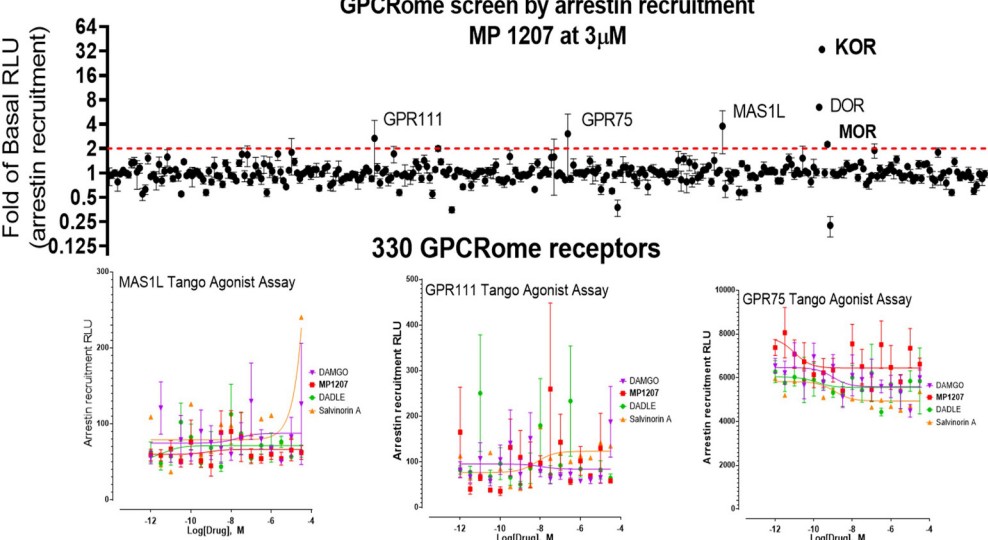

**Appendix 1—figure 6.** MP1207 is selective for opioid receptors in the GPCRome screen. MP1207 was screened against 330 non-olfactory GPCRs for agonism in the arrestin recruitment TANGO assay. Each point shows luminescence normalized to basal level at a given GPCR at 3 µM MP1207 dose (>3000 higher dose than the binding affinity at opioid receptors), with vertical lines indicating the standard error of the mean. MP1207 induces an increase in signal twofold over basal at opioid receptors predominantly at KOR and DOR and much less at MOR. Results show selectivity for opioid receptors over non-opioid targets when tested at >10µM and >1500 fold higher than the binding affinity and agonistic potency at opioid receptors. The low signal at MOR is consistent with null arrestin recruitment at MOR. Several potential targets (GPR111, MAS1L, GPR75) did not show dose-dependent increase in signal and probably represent screening false positives.

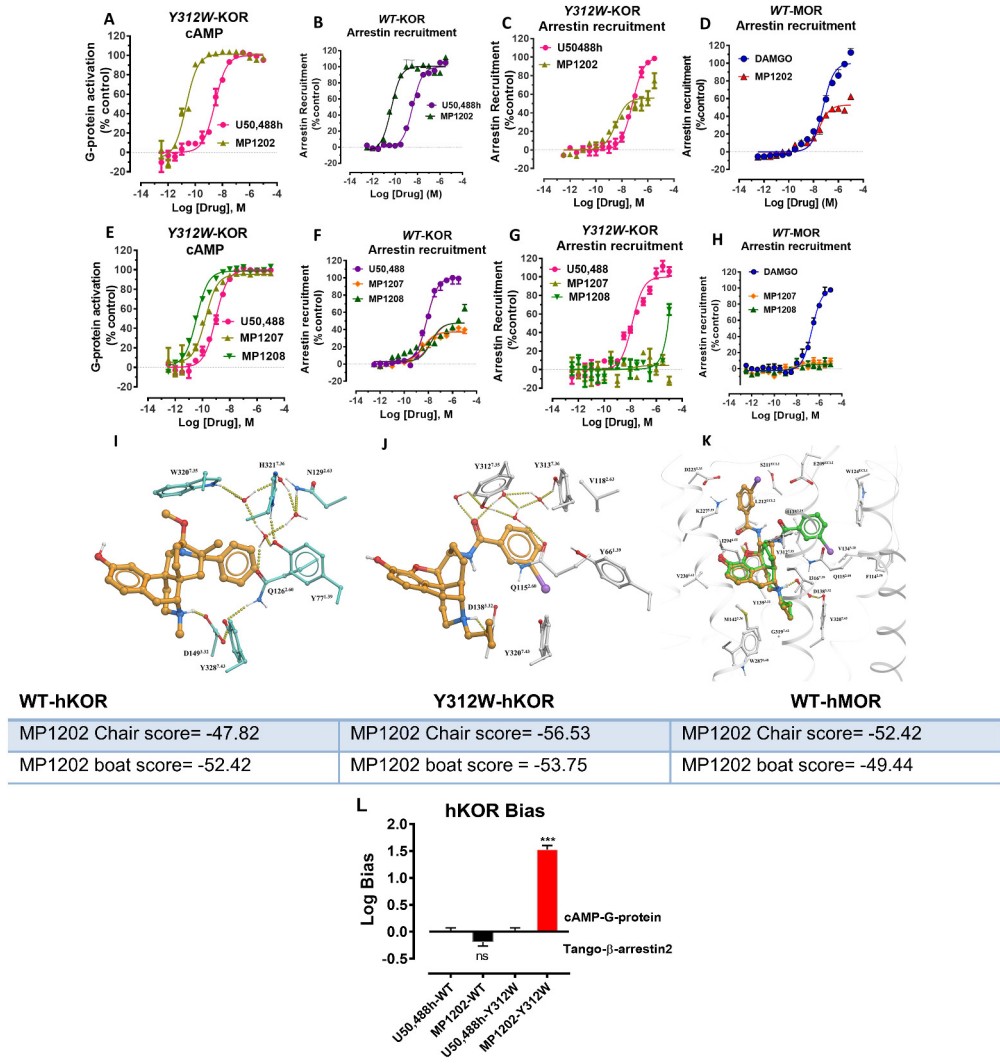

**Appendix 1—figure 7.** Mutation of KOR Y312W leads to a receptor mimicking MOR arrestin recruitment. MP1202 flips to a G protein biased agonist and arrestin recruitment for MP1207 and MP1208 is reduced. (**A and C**) At Y312W-hKOR, MP1202 (light green) is a G-biased agonist in cAMP inhibition (N = 3) (**A**) and Tango-arrestin recruitment assays (N = 3) (**C**) compared to U50,488H (pink). (**B**) At WT-hKOR, MP1202 (purple) acts as a full agonist for arrestin recruitment (N = 3). (**D**) At WT-hMOR, MP1202 (red) acts as a partial agonist for arrestin recruitment (N = 3). (**E and G**) At Y312W-hKOR, MP1207 (light green) and 1208 (dark green) show full agonism in cAMP inhibition (N = 3) (**E**) and reduced arrestin measurement in Tango-arrestin recruitment assays (N = 3) (**G**) compared to U50,488H (pink). (**F**) At WT-KOR, MP1207 (orange) and MP1208 (green) act as partial agonists for arrestin recruitment (N = 3) compared to control U50,488H (purple). (**H**) At WT-MOR, no arrestin recruitment (N = 3) was observed for both agonists MP1207 and MP1208. (**I**) Conformation of selected residues seen in high resolution active state MOR structure along with crystallographic waters around TM2-TM3 region and crystallized ligand (BU72). MOR, conformation of conserved $Q^{2.60}$ is maintained by a rather extensive hydrogen-bonding network mediated by $W^{7.35}$ and at least four tightly bound waters as found in the crystal structure. (**J**) Conformation of selected residues and MP1202 seen in active state KOR structure, along with modeled waters around TM2-TM3 region. In KOR, no crystallographic waters were resolved in the structure, the non-conserved residues, including $Y^{7.35}$ instead of $W^{7.35}$, would rearrange water network and change conformation of $Q^{2.60}$, paramount for ligand binding. A theoretical water network was modeled using SampleFlood method of ICM Molsoft in the orthosteric ligand site, and resulting waters were

*Appendix 1—figure 7 continued on next page*

*Appendix 1—figure 7 continued*

optimized *via* several rounds of extensive conformational sampling. As a result of significant difference observed in the water network, compared to MOR, KOR's Q115$^{2.60}$ moves further inwards and is positioned between D138$^{3.32}$ and Y312$^{7.43}$, and this new position of Q115$^{2.60}$ is stabilized *via* a water-mediated hydrogen bond with Y312$^{7.35}$. (**K**) Docking poses of chair and boat conformations of MP1202 in Y312W KOR mutant. The chair form (−56.53) is favored over boat form (−53.75) at this mutant receptor for MP1202. (**L**) Bias plots for U50,488h and MP1202 at hKOR and Y312W hKOR mutant. MP1202 is not significantly different from U50,488h with respect bias for G over βarrestin-2 signaling at wild type receptor while at the hKOR mutant it shows G-biased signaling. Data are mean± SEM from N = 3 replicates. Data analyzed using unpaired t-test with Welch's correction,\*\*\*p=0.0001. See *Appendix 1—table 9* for values, *Appendix 1—figure 4* and *Appendix 1—table 10* for bias calculations.

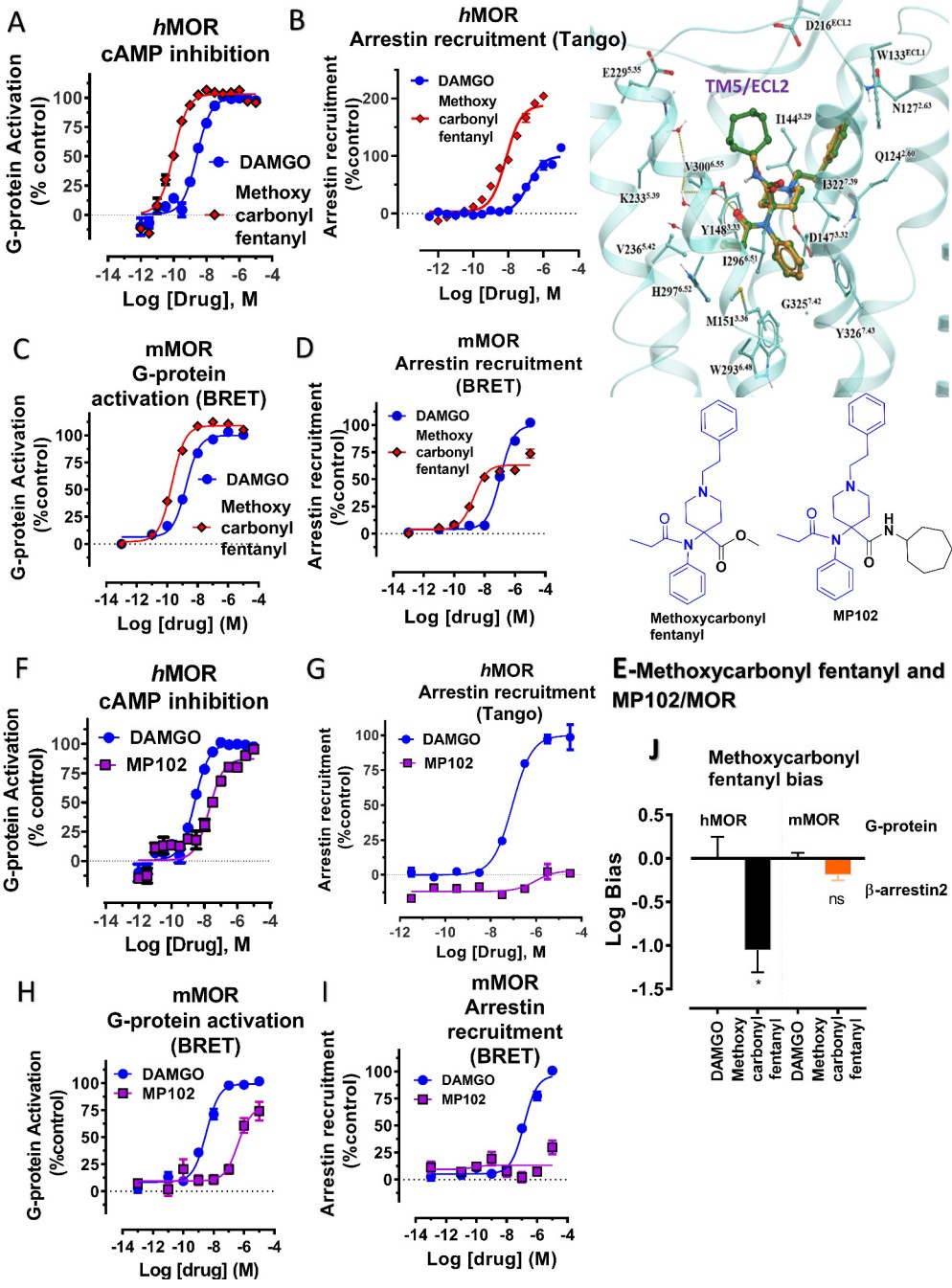

**Appendix 1—figure 8.** Methoxycarbonyl fentanyl amide analog MP102 targeting TM5 in MOR shows no arrestin recruitment compared to methoxycarbonyl fentanyl at mMOR in BRET assays. (**A-B**) Methoxycarbonyl fentanyl (red) is a full agonist at hMOR in cAMP (N = 3) and arrestin recruitment assays (N = 3) compared to DAMGO (blue). (**C–D**) Methoxycarbonyl fentanyl (red) is a full agonist at mMOR in BRET G protein activation (N = 4) and partial agonist in arrestin recruitment assays (N = 4) compared to DAMGO (blue). (**E**) Molecular docking of MP102 (green stick) engaging TM5 region and methoxycarbonyl fentanyl (yellow stick) not occupying TM5 region in *hMOR*. and chemical structures of methoxycarbonyl fentanyl and methoxycarbonyl fentanyl amide, MP102. (**F-G**) MP102 (purple) is a full agonist at mMOR in cAMP assays (N = 3) and shows no arrestin recruitment assays (N = 3) compared to DAMGO (blue). (**H–I**) Similarly, MP102 (purple) is a partial agonist at mMOR in BRET G protein activation assays (N = 3) and shows no arrestin recruitment assays (N = 6) compared

*Appendix 1—figure 8 continued on next page*

to DAMGO (blue). (**J**) Bias plots for DAMGO and methoxycarbonyl fentanyl at hMOR and mMOR. Methoxycarbonyl fentanyl is arrestin biased at hMOR and shows no preference for any pathway at mMOR. Data are mean± SEM from N = 3 replicates and is analyzed using unpaired t-test with Welch's correction,***p=0.0001. See *Appendix 1—tables 7– eight* for values, and *Appendix 1— table 10* and *Appendix 1—table 11* for bias calculations.

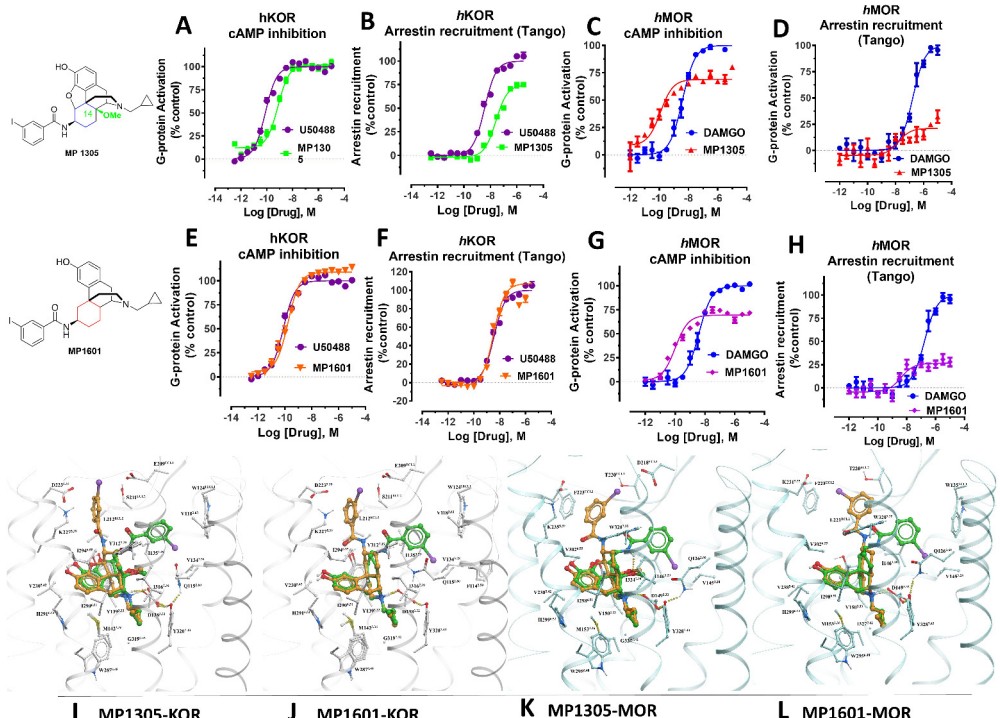

**Appendix 1—figure 9.** At human receptors, MP1305 is G-biased at MOR and KOR whereas MP1601 is G-biased at MOR and arrestin biased at KOR. (**A-B**) MP1305 (green) is a full agonist at hKOR in cAMP inhibition (N = 6) and partial agonist in Tango-arrestin recruitment assays (N = 6) compared to U50,488H (purple) at hKOR. (**C–D**) MP1305 (red) is a partial agonist at hMOR in cAMP inhibition (N = 3) and partial agonist in Tango-arrestin recruitment assays (N = 3) compared to DAMGO (blue). (**E–F**) MP1601 (orange) is a full agonist at hKOR in cAMP inhibition (N = 6) and Tango-arrestin recruitment assays (N = 6) compared to U50,488H (purple). (**G–H**) Similarly, MP1601 (purple) is a partial agonist at hMOR in cAMP inhibition (N = 3) and Tango-arrestin recruitment assays (N = 3) compared to DAMGO (blue). (**I–J**) Molecular docking of MP1305 and MP1601 with a chair (orange stick) or a boat confirmation (green stick) at hKOR and hMOR. (**K–L**) Chair and boat forms of MP1305 and MP1601 at an active state hKOR (in gray) and Chair and boat forms of MP1305 and MP1601 at an active state hMOR (in cyan). Preferred docking modes of MP1305 and MP1601 are analogous and they both maintain chair confirmation at hMOR and are biased toward G protein, while being balanced agonists at hKOR taking boat conformation. See *Appendix 1— table 9* for values, *Appendix 1—figures 4* and *Appendix 1—table 10* for bias calculations.

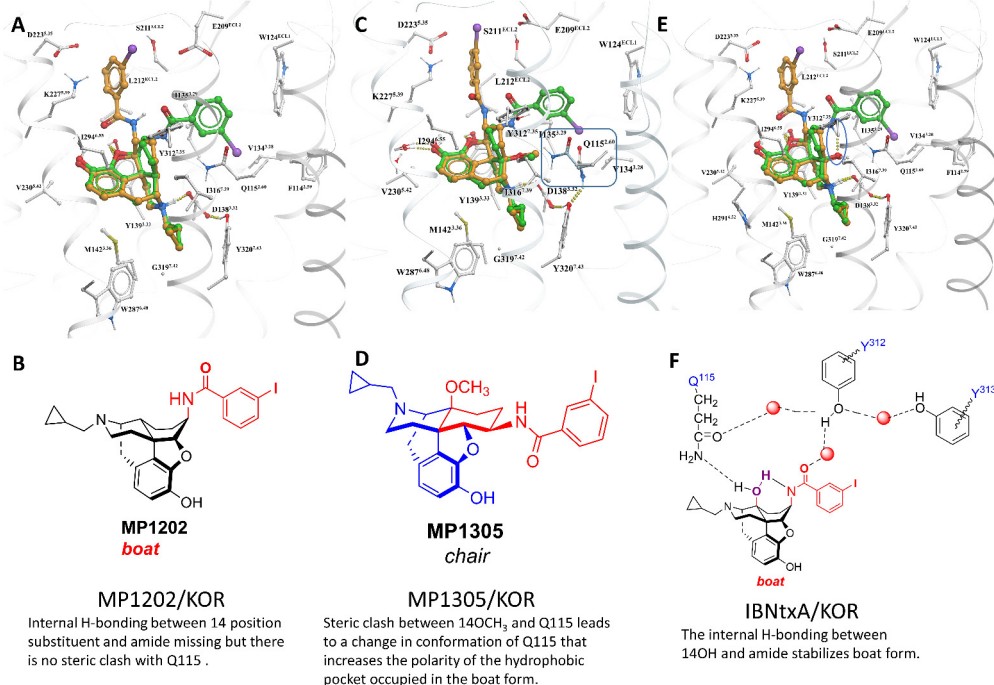

**Appendix 1—figure 10.** MP1305 is G protein biased at KOR. Docking pose for ligands inside active state KOR shown in white carbon sticks and white ribbon representations. (**A–B**) MP1202 in KOR favors boat conformation. (**C–D**) MP1305 in KOR favors chair conformation. 14th-position-Methoxy of MP1305 causes sidechain of Q115$^{2.60}$ residue to undergo a conformational change, when compared to ligands with non-Methoxy substitutions at 14th-position (shown in cyan colored carbon sticks representation). (**E**) **IBNtxA** in KOR favors boat conformation. (**F**) Stabilization of **IBNtxA** in boat form through internal hydrogen-bonding between 14-OH and amide and with Q115$^{2.60}$ is shown. The docked and energy minimized pose of MP1305 shows the substitution of 14-OH with larger methoxy group displaces the conserved Q115$^{2.60}$ of KOR. In ligands with 14-OH, the Q115$^{2.60}$ residue of KOR is directed toward the ligand and forms hydrogen bonds with the 14-OH. In the case of MP1305, the methoxy group pushes Q115$^{2.60}$ residue away to avoid steric clashes, and the polar terminus of the side chain ends up in the previously hydrophobic TM2-TM3 sub-pocket of KOR. Furthermore, due to internal ligand sterics, the internal hydrogen bond between the amide and 14-OH, as seen in compounds such as **IBNtxA**, is not possible for MP1305. The combination of these two factors, increasing polarity of the hydrophobic TM2-TM3 sub-pocket (an effect similar to Y312W$^{6.35}$ KOR sub-pocket mutation- and lack of an internal hydrogen bond stabilizing the boat form, shifts the equilibrium toward the chair form for MP1305).

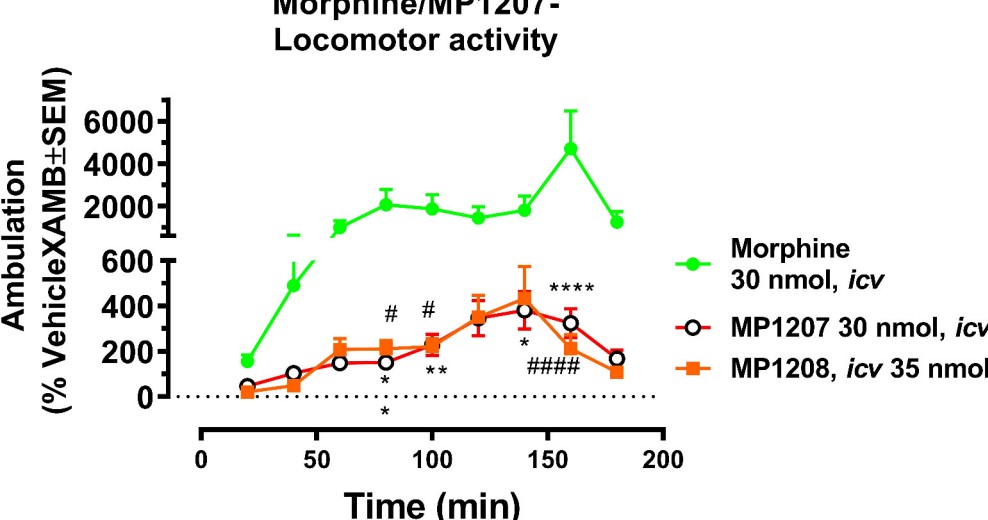

**Appendix 1—figure 11.** MP1207 and 1208 show significantly less hyperlocomotion compared to morphine. Mice were administered *icv* either morphine (30 nmol; n = 18), MP1207 (30 nmol; n = 26) or MP1208 (35 nmol; n = 10) and the ambulation of each group of mice monitored using the CLAMS/Oxymax system. **Morphine** (30 nmol, *icv*) significantly increased forward ambulation in comparsion to MP1207 at 80 min (**p=0.002), 100 min (**p=0.01), 140 min (*p=0.03), and 160 min (****p<0.0001) as determined by two-way ANOVA followed by Dunnett's multiple-comparison test. Similarly, significant morphine-induced increases in ambulation as compared to the response **of MP1208** were observed at 80 min (#p=0.0249), 100 min (#p=0.0497), and 160 min (####p<0.0001) as determined by two-way ANOVA followed by Dunnett's multiple-comparison test.

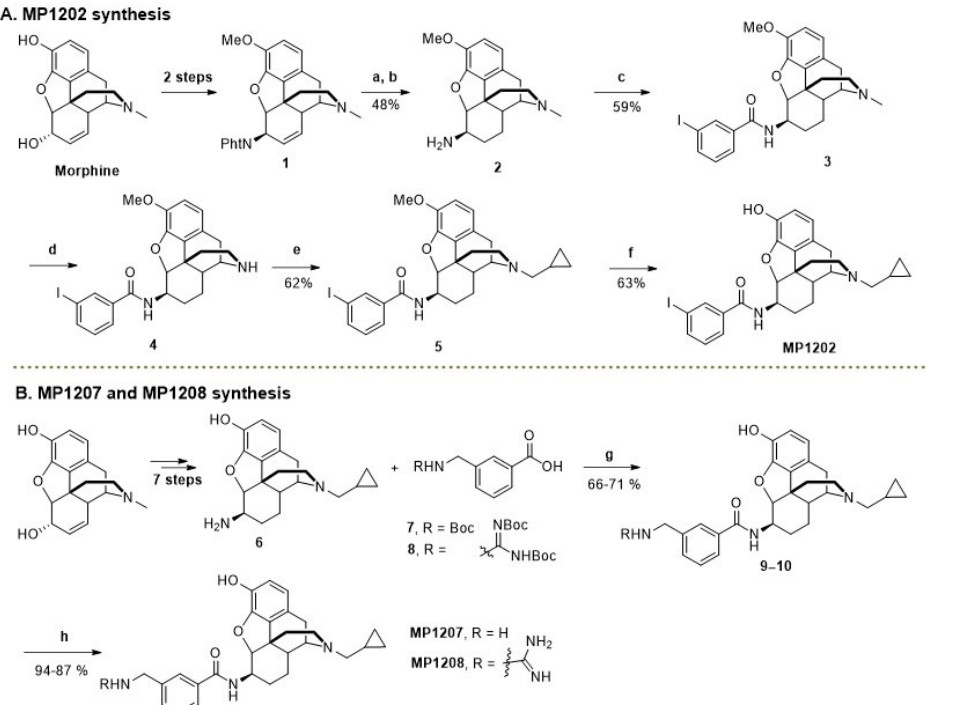

**A. MP1202 synthesis**

**B. MP1207 and MP1208 synthesis**

**Reagents/condition: a)** Pd/C, $H_2$, 100 Psi 24 h, MeOH; **b)** $NH_2$-$NH_2$*$H_2O$, MeOH; **c)** *m*-IBA, HATU, DIPEA, DMF, 0 °C-rt; **d)** i. DIAD, AcN, 65 °C, ii. Py.HCl/ACN; **e)** $Na_2CO_3$, R-X, DMF, 90 °C; **f)** $BBr_3$, DCM 0 °C- rt; **g)** HATU, DIPEA, DCM, 0 °C-rt; **h)** TES, TFA/DCM, rt.

**Appendix 1—scheme 1.** Synthesis of m-arylamido dihydromorphinans MP1202 (**A**), MP1207, MP1208 (**B**).

**MP1209 and MP1210 synthesis**

**Reagents/condition: a)** PyBOP, DIPEA, THF, rt; **b)** 4N HCl in dioxane, rt.

**Appendix 1—scheme 2.** Synthesis of m-arylamido dihydromorphinans MP1210 and p-arylamido dihydromorphinan analog, MP1209.

**Reagents/condition**: **a**) NaH, MeI, DMF, 55 °C; **b**) aq. HCl, MeOH; **c**) Li-selectride, THF, -78 °C-rt; **d**) PhtNH, PPh$_3$, DIAD, THF, 0 °C-rt; **e**) NH$_2$-NH$_2$*H$_2$O, MeOH; **f**) $m$- IBA, HATU, DIPEA, DMF, 0 °C-rt; **g**) BBr$_3$, DCM 0 °C- rt.

**Appendix 1—scheme 3.** Synthesis of 14-$O$-methyl $m$-iodoarylamidomorphinan MP1305.

**Reagents/condition: a**) NH$_4$OAc,NaCNBH$_3$, MeOH, rt; **b**) $m$-IBA, HATU, DIPEA, DMF, rt; **c**) BBr$_3$, DCM, 0 °C-rt.

**Appendix 1—scheme 4.** Synthesis of $m$-iodoarylamido-4,5-deoxymorphinan MP1601.

