## [Decision Letter]

**Acceptance summary:**

This paper will be of broad interest to scientists studying GPCR signaling, opioid pharmacology, and pain and addiction. The authors use computational modeling to develop dual G protein biased MOR-KOR agonists with reduced side effect profiles. The premise is novel, and the data support well the key conclusions.

**Decision letter after peer review:**

Thank you for submitting your article "Controlling opioid receptor functional selectivity by targeting distinct subpockets of the orthosteric" for consideration by *eLife*. Your article has been reviewed by three peer reviewers, and the evaluation has been overseen by Olga Boudker as the Senior and Reviewing Editor. The reviewers have opted to remain anonymous.

The reviewers have discussed the reviews with one another and the Reviewing Editor has drafted this decision to help you prepare a revised submission.

To date, the majority of biased drug development strategies have focused on increasing bias at individual receptors in order to reduce adverse effects while maintaining or enhancing the beneficial effects of receptor activation. Although conceptually promising, the clinical effects of single receptor biased ligands have been equivocal. The approach used in this manuscript is to developed selective ligands that are G protein biased at two different receptors, the mu and kappa opioid receptor (MOR, KOR). The premise is that G protein bias at each receptor may minimize adverse effects associated with MOR or KOR activation individually. Furthermore, co-activation of KOR will enhance the pain-relieving properties and minimize the rewarding properties of MOR activation. The crystal structure of activated MOR and KOR is used to identify regions within these receptors that confer G protein bias vs. a more balanced profile. The computational modeling shows that engagement of the TM2-TM3 pocket correlates with balanced agonists, while the TM5-ECL2 site correlates with G protein biased agonists at MOR and KOR. Also, the use of mutational analysis adds a level of rigor to the experimental design. Importantly, you have made several MOR-KOR G protein biased ligands, and show that one of these (MP1207) has effective antinociceptive properties without reward-seeking behaviors, and rather than respiratory depression it produces stimulation. This work could have significant utility in guiding future efforts to develop safer analgesics. Overall, all reviewers were enthusiastic about the work. Several significant weaknesses were identified and would need to be addressed. Below is a summary of the major points followed by the comments from individual reviewers.

1) All reviewers have agreed that the in vivo work has significant weaknesses. However, it is essential for the paper. Mostly, the reviewers were concerned that you are overly focused on the G protein bias while ignoring efficacy. It remains unclear whether G protein bias might confer benefits as opposed to other mechanisms such as partial agonism. A key issue is that comparisons are made between different opioid drugs that are administered differently and used at different relative doses. This makes it more challenging to make conclusions about each agonist's relative propensity to cause a particular effect. Three possible solutions to this:1) you provide further explanations/existing data as to why this experimental approach was taken and why you can draw the conclusions you make; 2) you repeat experiments using more consistent doses/administration routes; 3) you revise the discussion of these data so that any statement regarding the relative effect of agonists is qualified by the limitations we have highlighted.

2) The physiological experiments use the partial agonist morphine as a standard for comparison, while the cell-based and in-vitro assays all use the full agonists DAMGO or U50,488 as references. Morphine and/or buprenorphine should be included in the cell-based assays or the GTPγS assay to get an idea of the relative efficacy and affinity compared to these compounds. The comparison would allow for a less ambiguous interpretation of the data. Additionally, the GTPγS activity table (Appendix 1—table 2) is important and should be in the main manuscript- with morphine included.

3) There are some questions about the potential rewarding effects of MP-120. Locomotor effects of MP-120 appear to be highest at the lower dose and at a later time point. Locomotor stimulation often corresponds with increased mesolimbic dopamine release, which can also underlie the rewarding properties of a ligand. It would be important to reconcile this dose and timing with the conditioning experiment to more clearly determine if MP-120 has rewarding properties.

4) Data analysis in the in vitro experiments is not satisfactory. The finding that by targeting a sub-pocket in the receptors can allow the “rational” design of biased ligands is potentially important. However, unless you provide details about how these biased values are derived and provide errors and statistical analysis it is impossible to say if these findings are robust.

*Reviewer 1:*

1) For the tail flick experiment could you please include the time course in the supplementary data? It would be good to see the peak time for anti-nociceptive effects relative to respiratory depression and locomotor stimulation.

2) For the CPP experiments I think you should test 30 nmol of MP1207 since this lower dose had a greater effect on locomotor stimulation which is associated with mesolimbic dopamine release. Also the timing for conditioning should be discussed. Locomotor stimulation peaks around 120 min, while conditioning for CPP was done 0-40 min following infusion.

3) The authors hypothesize that the respiratory stimulation induced by MP1207 is due to KOR activation. What does a "standard" KOR agonist like U50,488 do in this assay? Also, could you please discuss what respiratory stimulation might mean clinically.

4) The data with the MOR and KOR knockouts add a level of rigor to the study. However, I'm not sure that it necessarily adds clarity. I think expressing the data in Figure 6F and G, as area under the curves as well would help with the analysis. Also, in the KOR KO, MP1207 doesn't show respiratory depression as a traditional MOR agonist would. Could you please discuss this further? Is there are reason to not do these types of experiments with MOR and KOR antagonists?

*Reviewer 2:*

1) The physiological experiments use the partial agonist morphine as a standard for comparison while the cell-based and in-vitro assays all use the full agonists DAMGO or U50,488 as references. Morphine should be included in the cell-based assays, or the GTPγS assay at a minimum, to get an idea of the relative efficacy and affinity compared to morphine. This would allow for a more complete interpretation of the data. Additionally, the GTPγS activity table (Appendix 1—table 2), is important and should be in the main manuscript- with morphine included.

2) The respiratory depression data is confounded by the increased locomotion rate (measured at the same time) induced by the new compounds, perhaps due to KOR agonist actions. Increased movement will increase the respiratory rate and introduce artifacts into the respiration measurements. This is clear with 100 nmol morphine in Figure 6D. It is not clear how these complications were addressed.

3) It's not clear that the doses chosen for respiratory depression and CPP are equianalgesic with morphine with respect to MOR activity. A full dose response in the KOR knockout (Figure 6B) would help with interpretation of the respiratory depression data.

4) Comparing IP morphine with icv MP1207 in the CPP assay is also questionable.

5) Recent work with a "G protein biased" MOR mouse (Kliewer et al., 2019) is an elegant study that raises serious questions about the G protein bias and on-target opioid effects. Is there any evidence that arrestin signaling can actually cause respiratory depression or place preference? This should be addressed.

6) It's unclear if there is any statistical threshold for calling an agonist "biased". The bias factor provides a nice number showing an extent of apparent bias but it's not clear what the line is between biased and not biased as it seems like a continuum with an arbitrary cutoff.

7) It's also unclear if statistical tests were performed for the respiration and locomotion data in Figure 6D and E. Were none of these points significant? That would affect interpretation of these data.

*Reviewer 3:*

1) The final sentence of the Abstract that suggests that dual G protein biased MOR/KOR signalling provides better therapeutic profiles does not entirely fit with the data set or indeed, the discussion of these data by the authors. I suggest the authors rewrite this such that while dual Mu/Kappa engagement appears to be beneficial there is no definitive causal link with bias (or indeed evidence that it is not important).

2) The Introduction quotes bias factors from different studies. No explanation is given in the Introduction text as to what these values are or what they might mean. Values based on methodologies using the Black and Leff operational model developed by Kenakin, Christopoulos and Ehlert to generate log[Tau/KA] are relative values subject to the experimental system used. Different studies will have used different cell backgrounds, different receptor expression levels, different assays/approaches to measure signalling events and even different reference agonists. The comparison of bias factors across different studies is likely not meaningful and such quantitative comparisons should be removed from the Introduction.

3) Bias factors are quoted throughout the manuscript but the work up to these values is not provided nor an explanation as to what they mean. Presumably, a bias factor >1 means bias toward cAMP inhibition and < 1 means bias towards arrestin recruitment. I notice that often ligands with a bias factor of < 1 are described as balanced rather than arrestin biased? The authors are using a methodology based on the operational model of agonism. We need to see the various steps that have been used to calculate these values including the Log[tau/KA], deltaLog[tau/KA] (which should be compared with appropriate statistical methods and deltadeltaLog[tau/KA] values). The bias factors are presented without error associated with them so it is impossible to evaluate the significance of such values, and ultimately this means that we cannot assess the robustness of these findings. This information is needed for publication. I would also like to see a more detailed description of the use of this analysis in the Materials and methods rather than the citation of the Kenakin and Christopoulos methodology paper.

4) Some of the curve fitting in some of the figure panels needs to be addressed. For example, in Figure 1C the lower asymptote of MP1104 is not defined at all and the concentration-response curve appears to start below zero. How was the lower asymptote of this curve defined? The same issue is true for the IBtxA curve at the hKOR G protein assay (S3C) and MP1202 in Figure 2I. In Appendix 1—figure 1 the maximal response of IBNtxA and MP1202 in the arrestin assay are not defined so it is not clear how this not has been fitted or how EC50 and Emax have been estimated. The same is true for IBNtxA in S3G. Fitting data in this way will likely impact bias calculations too. In some cases (eg. IBtxA) there are some compelling reversals of the order of potency with DAMGO as qualitative indicators of bias but in some cases I am not sure if reliable quantitative estimates of bias can be derived from these data.

5) DAMGO is less potent in the arrestin recruitment assay as compared to the cAMP assay indicating the former is more weakly coupled to the MOR and/or less amplified. The same is true for the reference agonist U50,488 at the KOR. In such systems, a partial agonist may give a robust response in the highly coupled endpoint but a weak response or no response in the weakly coupled endpoint. From these data, I would say that the authors cannot definitively call such examples “G protein biased”. This includes MP1207 and MP1208 at the MOR (Figure 3 and Appendix 1—figure 1), MP1209 and 1210 “G protein biased” at the MOR (Figure 4), MP1306 and MP1601 at the mMOR (Appendix 1—figure 1). Indeed, the [35S]GTPgS data for some of these ligands does indicate that these ligands are partial agonists for G protein activation too. The possibility that partial agonism might account for these observations is acknowledged in the Discussion. I recommend that the descriptions of these ligands as G protein biased in the Results text is toned down to be consistent with this ambiguity. MP1207 and MP1208 have been assigned bias values (8 and 23) in S2D but, given that no response is observed in the arrestin assay, it is not clear how these bias factors have been derived. The boxes in various figures that define such ligands as “G protein-biased” is at odds with the above ambiguity and should be removed.

6) The authors should provide further characterization/description of the double mutant is needed in order to interpret their data. Is it efficiently expressed at the cell surface and how does this expression compare to WT? How does the affinity and potency of the reference agonist change at this mutant? Indeed, given that bias is always expressed relative to the action of the reference agonist can the authors be sure that these mutations have altered the bias of MP1207/8 but not U50,488? How does U50,488 bind the receptor in terms of interaction with TM2/ECL2?

7) The figures legends are excessively long due to the inclusion of potency data for each agonist in each panel. These data should be presented in a table as the authors have done for their DOR data. In the figures more detail is needed regarding the data shown. Is this grouped data or is this an individual experiment? The authors should clarify if the numbers in parentheses are pEC50 plus minus SEM. The authors should provide the number of individual experiments and the number of replicates within each experiment for each data set. This information is available for some but not all data sets.

8) All in vivo data is shown as % decreases or as %MPE. Baseline values for each group should be given to show that they were not different in both analgesia and respiratory depressant assays. I would strongly encourage including raw data from these experiments as a supplementary table.

9) Drug administration route: In the in vivo experiments there are instances in which drugs are compared in the same assay but are administered differently. The reason for this needs further explanation. The basis for chosen doses needs further clarification. In the CPA/CPP experiments it is not clear why IP morphine/U50,488h is compared to ICV MP1207 drug administration. Unless there is evidence that this route results in appreciable concentrations of MP1207 in the circulating plasma you are not comparing like with like. It is a significant confound in this experiment when trying to demonstrate improvement over morphine in particular. Routes of administration and factors such as ability to cross the blood-brain barrier will, naturally, determine the measured effects of an opioid. This may be problematic when comparing the effects of opioid administered via different routes. Indeed, given the importance of peripheral opioid receptors for analgesia, tolerance and constipation it is difficult to predict the relative therapeutic benefits of a drug based on ICV data alone. This should be discussed.

10) In measurements of respiratory depression, a dose of 10 mg/kg morphine IP is used. Without a correlate in the other assays it is not clear why this is a useful comparator for the action of ICV MP1207. Similarly, U50,488h is also administered IP in this assay but ICV in the antinociception assay. The 30 nmol and 100 nmol ICV dose of morphine used in respiration are in excess of the maximal concentration of morphine used in the tail-flick assay. 10 nM ICV morphine would seem a more logical concentration to use for comparison with 30 nmol MP1207. This is also true for measurements of locomotor activity.

11) Data was taken at the point of peak antinociception for each drug, but it is not clear if these drugs peak at the same timepoint. The duration of an analgesic effect is also important information to know. The authors should supply supplementary data showing the time course of analgesia.

[Editors' note: further revisions were suggested prior to acceptance, as described below.]

Thank you for submitting your article "Controlling opioid receptor functional selectivity by targeting distinct subpockets of the orthosteric site" for consideration by *eLife*. Your article has been reviewed by three peer reviewers, and the evaluation has been overseen by Olga Boudker as the Senior and Reviewing Editor. The following individual involved in review of your submission has agreed to reveal their identity: Amynah A Pradhan (Reviewer #1).

The reviewers have discussed the reviews with one another and the Reviewing Editor has drafted this decision to help you prepare a revised submission.

Summary:

This paper will be of broad interest to scientists studying GPCR signaling, opioid pharmacology, and/or pain and addiction. The authors use computational modeling to develop dual G protein biased MOR-KOR agonists with reduced side effect profiles. The premise is novel and the key claims are well supported by the data.

Revisions:

All reviewers agree that the manuscript has been significantly improved. Only minor additional concerns listed below were raised:

1) Since these agonists appear to be slightly more partial than buprenorphine (adding this reference agonist was helpful), it would be useful to discuss what is known about buprenorphine in respiratory depression, CPP and hyperlocomotion assays- with the caveat that the buprenorphine metabolite norbuprenorphine may be biologically relevant. This may help strengthen the case that partial KOR agonism is beneficial relative to the weak MOR agonism KOR antagonism of bup.

2) It appears there is some weak DOR agonism/ antagonism and that both MP1207/1208 can bind to DOR with only slightly lower affinity than MOR and KOR. DOR antagonism is known to modulate some MOR-mediated behaviors, so there is the potential for DOR agonism to play a role in some of the behaviors reported, which should be acknowledged.

3) "Together, these results support that dual MOR and KOR agonism may offset the liabilities characteristic of receptor-selective agonists".

4) The results are consistent with the dual MOR/KOR agonism hypothesis. Still, they might not differentiate weak MOR agonism from MOR/KOR agonism unless they are also done in the KOR knockout mice as was done for analgesia. Therefore, the language should be toned down.

5) Throughout the manuscript, the authors refer to bias factors. For example, MP1104 is said to have bias factors of 0.58 and 0.15 at MOR and KOR. Bias factors are not absolute values and are relative to the reference agonist used. Readers unfamiliar with bias factors will have no idea what these values mean unless they go to the Materials and methods section and then read a couple of papers cited there. The first time these bias factors are used, the authors need to explain more clearly how they were derived and what exactly they mean. Specifically: A) That at the MOR this is relative to the action of the agonist DAMGO and at the KOR, this is relative to the action of the agonist U50,488h – this is in the Materials and methods but needs to be stated in the text. B) That in all cases in the manuscript, you are comparing G protein signaling with arrestin recruitment so that bias factors greater than 1 means bias towards G protein and less than 1 means bias towards arrestin. C ) Where your values of Log bias are not significantly different from zero (so your bias factors are around 1) you need to state that the ligand is not biased.

6) In some cases, the analysis shows that some of the ligands show bias towards arrestin, but this is not highlighted in the text. Instead, it seems to be stated that the ligand shows robust at recruiting arrestin (e.g., MP1104, IBNtxa @ the KOR). This is not the case for G protein biased ligands. Why the difference? It is less confusing if bias is described the same way in both directions.

7) The statement that IBNtxa induces minimal arrestin recruitment at the MOR is at odds with the Appendix 1—figure 3, which shows that this ligand can induce a robust level of recruitment to 70% of that of DAMGO – the interesting thing is that the ligand is more potent than DAMGO in the G protein assay but less potent than DAMGO in the arrestin assay.

8) Similarly, the statement that MP1202 shows diminished arrestin recruitment is at odds with Appendix 1—figure 1C as the maximum effect of the ligand at arrestin recruitment is not reached at the highest concentration used – indeed, it would appear that MP1202 is less potent in the arrestin assay, but relative to DAMGO, may be more efficacious in this assay as compared to the cAMP assay.

9) Some of the concentration-response curves appear to be biphasic and not well fitted to the concentration-response curve shown in the relevant figure: Figure 3B, 1208; Figure 4D, 1209 and 1210, Figure S7C, 1202; Figure S7F, 1208; maybe Figure S7G, 1208; Figure S9D, 1305. Can the authors provide an explanation for this? Can they rule out some sort of non-specific effect occurring when these ligands are used at very high concentrations? It seems to be something observed in the arrestin assays in particular.

10) A bias factor for MP1202 at the mutant Y312W is stated as 34 and this is shown in figure S7L. The authors should provide the data/calculations in a supplementary table from which these values were derived, similar to those tables S10/11 that they have already provided for ligands at WT ORs.

11) Abstract: arrestin instead of arestin.

---

## [Author Response]

[…] This work could have significant utility in guiding future efforts to develop safer analgesics. Overall, all reviewers were enthusiastic about the work. Several significant weaknesses were identified and would need to be addressed. Below is a summary of the major points followed by the comments from individual reviewers.1) All reviewers have agreed that the in vivo work has significant weaknesses. However, it is essential for the paper. Mostly, the reviewers were concerned that you are overly focused on the G protein bias while ignoring efficacy. It remains unclear whether G protein bias might confer benefits as opposed to other mechanisms such as partial agonism. A key issue is that comparisons are made between different opioid drugs that are administered differently and used at different relative doses. This makes it more challenging to make conclusions about each agonist's relative propensity to cause a particular effect. Three possible solutions to this:1) you provide further explanations/existing data as to why this experimental approach was taken and why you can draw the conclusions you make; 2) you repeat experiments using more consistent doses/administration routes; 3) you revise the discussion of these data so that any statement regarding the relative effect of agonists is qualified by the limitations we have highlighted.

We thank the reviewers and the editor for pointing out partial agonism at MOR as an alternative pharmacological mechanism that may explain in vivo results for our ligands, which specifically demonstrated effective analgesia with reduced side effects, including lack of respiratory depression in mice. We are acutely aware of the ongoing debate about whether β-arrestin2 is indeed responsible for MOR-mediated respiratory depression. As the reviewers noted, our lead ligands, designed as dual MOR/KOR agonists with reduced (KOR) or abolished (MOR) arrestin recruitment, also behave as partial agonists in MOR cAMP and BRET assays. This comprises a complex multidimensional pharmacological profile for our ligands, and we demonstrate in vivo that it is beneficial. But we agree that it would be a great oversimplification to ascribe these reduced side effects results to any specific property of our new ligands, whether its dual, biased, or partial agonism, or a combination of these properties. Moreover, it is unlikely that any specific in vivo experiment would be sufficient to unambiguously commit to any conclusion here. Therefore, we revised the discussion of the in vivo data to include partial agonism at MOR and KOR as a possible factor in MP1207’s as well as 1208’s (now fully characterized in vivo) favorable attenuated opioid side-effect profile. Also, to address some inconsistencies noted by reviewers, we added new icv morphine/U50,488h analgesia, respiration, and CPP/CPA data as described in the answers below.

2) The physiological experiments use the partial agonist morphine as a standard for comparison, while the cell-based and in-vitro assays all use the full agonists DAMGO or U50,488 as references. Morphine and/or buprenorphine should be included in the cell-based assays or the GTPγS assay to get an idea of the relative efficacy and affinity compared to these compounds. The comparison would allow for a less ambiguous interpretation of the data. Additionally, the GTPγS activity table (Appendix 1—table 2) is important and should be in the main manuscript- with morphine included.

Morphine and buprenorphine data has been added for GTPgS (Table S2) and cAMP assays as well as BRET assays (Appendix 1—table 7). We prefer to keep the GTPgS data in the supplementary section to maintain the flow of the manuscript. Two in vivo figures are now part of the main manuscript (Figures 5-6). MP1208 has also been fully characterized pharmacologically and similar to MP1207 shows no CPP/CPA and attenuated respiratory depression.

3) There are some questions about the potential rewarding effects of MP-120. Locomotor effects of MP-120 appear to be highest at the lower dose and at a later time point. Locomotor stimulation often corresponds with increased mesolimbic dopamine release, which can also underlie the rewarding properties of a ligand. It would be important to reconcile this dose and timing with the conditioning experiment to more clearly determine if MP-120 has rewarding properties.

The editor raises a good point here. We find that hyperlocomotion and the rewarding effect of drugs as reflected by CPP seem to be separable. New data with MP1207 CPP/CPA at 30 nmol and MP1208 CPA/CPP at 100 nmol icv have been added, showing that no place preference and/or place aversion is seen at either dose. Furthermore, at equianalgesic doses, both MP1207 and 1208 show far less hyperlocomotion (See Appendix 1—figure 11) compared to icv morphine, and morphine shows CPP whereas neither 1207 nor 1208 does so.

4) Data analysis in the in vitro experiments is not satisfactory. The finding that by targeting a sub-pocket in the receptors can allow the “rational” design of biased ligands is potentially important. However, unless you provide details about how these biased values are derived and provide errors and statistical analysis it is impossible to say if these findings are robust.

As requested by the editor to accurately estimate bias we have added a new Appendix 1—figure 4 showing a graphical representation of bias with errors associated with the bias of ligands. Bias calculations are shown in newly added Appendix 1—table 10 and 11.

Reviewer 1:1) For the tail flick experiment could you please include the time course in the supplementary data? It would be good to see the peak time for anti-nociceptive effects relative to respiratory depression and locomotor stimulation.

Time course of analgesia for both icv administered MP1207 (Figure 5C) and MP1208 (Figure 5D) has now been added. Note the peak effect for analgesia and respiratory effects do not match for both drugs compared to morphine. This is not unexpected; there are many suggestions in the literature that the analgesic effects of opioids may be dissociated from those doses inducing other effects, such as reward (De Vry et al., Eur J Pharmacol, 1989; Wilson et al., Peptides, 2000) or respiration (Brice-Tutt et al., 2020 from our lab). It is further possible that receptor occupancy required to produce analgesia is different from that producing respiratory effects when polypharmacology involving two receptors is involved, a topic outside the current scope.

2) For the CPP experiments I think you should test 30 nmol of MP1207 since this lower dose had a greater effect on locomotor stimulation which is associated with mesolimbic dopamine release. Also the timing for conditioning should be discussed. Locomotor stimulation peaks around 120 min, while conditioning for CPP was done 0-40 min following infusion.

We appreciate this critique from the reviewer. CPP of MP1207 at 30 nmol and MP1208 at 100 nmol (Figure 5B) has been added. Both drugs show no CPP and/or CPA. We have added more animals to the morphine group for both respiration (Figure 6B) and locomotor experiments (Figure 6E). Morphine at 30 and 100 nmol showed hyperlocomotor activity. Furthermore, morphine shows respiratory depression as well as CPP while MP1207 doesn’t at the highest dose tested. There is a discrepancy in peak effects for MP1207 and MP1208 analgesia vs respiration and locomotor effects which we are unable to explain at this point (although see response to 1; above). However, while comparisons of this sort are common (for instance, see Varadi et al., 2016), and a 40 min conditioning session still matches our analgesic time course to demonstrate pharmacological activity, we now offer the caveat of conditioning duration in the Discussion.

3) The authors hypothesize that the respiratory stimulation induced by MP1207 is due to KOR activation. What does a "standard" KOR agonist like U50,488 do in this assay? Also, could you please discuss what respiratory stimulation might mean clinically.

We have added U50,488h groups. icv U50,488h 30 nmol and 100 nmol (Figure 6B), showing it has no effect on respiration when administered icv. Dr. McLaughlin’s group has previously shown that U50,488h administered i.p. shows no respiratory stimulation up to 10 mg/kg, IP (PMID: 31258480). There is evidence from another paper published from the McLaughlin group that suggests KOR agonism can negate MOR mediated respiratory depression (PMID: 32562259), a finding reported previously with individual treatments of U50,488h and MOR agonists; Dosaka-Akita et al., 1993. It stands to reason that this respiratory stimulation would serve clinically to offset the respiratory effects of morphine and other MOR agonists used for pain management, potentially rendering safer analgesia.

4) The data with the MOR and KOR knockouts add a level of rigor to the study. However, I'm not sure that it necessarily adds clarity. I think expressing the data in Figure 6F and G, as area under the curves as well would help with the analysis. Also, in the KOR KO, MP1207 doesn't show respiratory depression as a traditional MOR agonist would. Could you please discuss this further? Is there are reason to not do these types of experiments with MOR and KOR antagonists?

We have removed the KO results on respiration and locomotor effects from this manuscript. Notably, reviewer 2 wanted us to evaluate MP1207 respiration at multiple doses in KOR KO mice, and we agree that would be ideal to evaluate the role of MOR/KOR towards respiration/locomotor effects to add further rigor to our work. KO mice are cleaner for testing as compared to using antagonists, which have defined durations of activity and compete with the compounds for receptor sites, making the KO mice more attractive for our work. Unfortunately, our KO colonies at UF were ordered to be much reduced during the Covid crisis and have not yet recovered, precluding additional experiments with KO animals.

Reviewer 2:1) The physiological experiments use the partial agonist morphine as a standard for comparison while the cell-based and in-vitro assays all use the full agonists DAMGO or U50,488 as references. Morphine should be included in the cell-based assays, or the GTPγS assay at a minimum, to get an idea of the relative efficacy and affinity compared to morphine. This would allow for a more complete interpretation of the data. Additionally, the GTPγS activity table (Appendix 1—table 2), is important and should be in the main manuscript- with morphine included.

As requested by the reviewer morphine and buprenorphine has been added to the GTPgS table (Appendix 1—table 2) as well as cAMP assays and BRET (Appendix 1—table 7). The buprenorphine published data (PMID: 27223691) is from the laboratory of the late Dr. Pasternak. Both drugs appear as full agonists in the GTPgS assays, suggesting the presence of substantial receptor reserve. Numerous other papers from Pasternak’s lab (PMID: 15939800 in mMOR, PMID: 15893644 in hMOR, PMID: 14579421 in mMOR) have shown morphine to act as a full agonist. In addition, morphine has been found to be a full agonist by others (PMID: 32234959 using BRET-Gαi2, PMID: 20647394 and PMID: 31501422 using cAMP and Gαi2 activation). In cAMP assay morphine is a full agonist while buprenorphine is a partial agonist (E_Max_=77% of DAMGO). In Gi-1-BRET assay at hMOR, morphine behaves as a full agonist while buprenorphine is a partial agonist (Author response image 1).

Similarly at mMOR in BRET assays morphine was a full agonist (E_max_=109%) while buprenorphine was a partial (E_Max_=59%). See Appendix 1—table 7. We also note that both drugs have been shown in the literature to be partial agonists (PMID: 16436499 in C6 glioma transfected with MOR and buprenorphine in mMOR-CHO; PMID: 9580589 ) suggestive partial vs full agonism is cell line and assay dependent.Taken together, it is clear out our in vitro assays are showing substantial receptor reserve, making it difficult to capture the partiality of some compounds such as morphine.

However, the use of the operational model mitigates against this by considering shifts in both potency and efficacy. Furthermore, the literature clearly suggests that receptor reserve is present in vivo as well. Note that buprenorphine is a fully effective analgesic in rodents (PMID: 27223691) and humans, despite a “true” efficacy of ~25% in unamplified systems ( PMID: 32234959). We have decided to keep the GTPΥS data in the supplementary section so that we don’t interrupt the flow of the manuscript. Accordingly, we think it best to keep the GTPgS data in the supplementary section for those readers wishing to review it.

2) The respiratory depression data is confounded by the increased locomotion rate (measured at the same time) induced by the new compounds, perhaps due to KOR agonist actions. Increased movement will increase the respiratory rate and introduce artifacts into the respiration measurements. This is clear with 100 nmol morphine in Figure 6D. It is not clear how these complications were addressed.

We share the reviewer’s concern. We have added more animals to the icv morphine group for both respiration (Figure 6B) as well as locomotor (Figure 6D). Data on MP1208 at 35 nmol (Figure 6C) is now also included and a similar increase in breath rates is seen, consistent with the pharmacology of MP1207, a mixed KOR/MOR agonist. However, please note that KOR agonism is well documented to impair locomotor activity (including our own demonstration in Brice-Tutt et al., 2020). Moreover, in spite of inducing hyperlocomotion in rodents (as shown in Figure 6F), morphine still shows respiratory depression well documented to be mediated by MOR agonism. Collectively, our current data suggest that activity at KOR plays a role in the enhancement of breath rate with both MP1207/08. Also note, morphine produces far more hyperlocomotion compared to MP1207 and MP1208 at equianalgesic doses (See Appendix 1—figure 11). The data is also consistent with PMID: 32562259.

3) It's not clear that the doses chosen for respiratory depression and CPP are equianalgesic with morphine with respect to MOR activity. A full dose response in the KOR knockout (Figure 6B) would help with interpretation of the respiratory depression data.4) Comparing IP morphine with icv MP1207 in the CPP assay is also questionable.

We have added icv morphine (30 nmol) and icv U50488h (100 nmol) CPP/CPA data at equianalgesic doses (Figure 5B). icv analgesia of morphine was determined instead of relying on the literature values. The analgesic ED_50_ is 4.77 nmol for morphine (Figure 5A). At approximately 5x analgesic ED_50_ doses morphine (30 nmol) shows CPP and MP1207 (30 nmol, ED_50_=6.1 nmol) does not. Similarly, icv U50,488h (ED_50_=8.8 nmol) at approximately 15x analgesic ED_50_, U50,488h (100 nmol) shows CPA while MP1207 (analgesic ED_50_=6.1 nmol) and MP1208 (analgesic ED_50_=7.2 nmol) do not show CPA or CPP.

We are however unable to add a KOR KO dose response curve. We do agree this would add rigor as well as allow us to evaluate the role KOR plays in negating MOR induced respiratory depression. Unfortunately, as noted in response to reviewer 1 above, our KO colonies at UF were effectively eliminated during the Covid crisis and have not yet recovered, precluding additional experiments with KO animals. Respecting the wishes of the reviewer, we have accordingly decided to remove the KO experiments evaluating respiration of MP1207 at 30 nmol from the manuscript.

5) Recent work with a "G protein biased" MOR mouse (Kliewer et al., 2019) is an elegant study that raises serious questions about the G protein bias and on-target opioid effects. Is there any evidence that arrestin signaling can actually cause respiratory depression or place preference? This should be addressed.

This is indeed an elegant paper. The role of arrestin in opioid induced respiratory depression has also been questioned by investigators from the same group (PMID: 32052419) and bioRxiv 2020, 2020.08.28.272575. Both these papers and the preprint are now cited in our Discussion where we have included a role for partial agonism as a rationale for MP1207/08 decreased adverse effects compared to classical opioid agonists. A manuscript on low intrinsic efficacy playing a role towards low respiratory depression is also cited (PMID: 32234959). A preprint challenging this hypothesis is also now included in our citations (https://www.biorxiv.org/content/10.1101/2020.11.19.390518v1).

We also note that our compounds are biased and partial at KOR as well as MOR so different receptor mechanisms with poly pharmacology are also at play. Evidence for a role of arrestin signaling for CPP is low; we agree with the reviewer on this. However, compounds with mixed MOR/KOR profile haven’t been thoroughly evaluated for Gi vs arrestin signaling as well as partial agonism at MOR/KOR and their in vivo outcomesrespiration vs addiction potential is thereby less well characterized for compounds with activity at KOR

Reviewer 3:1) The final sentence of the Abstract that suggests that dual G protein biased MOR/KOR signalling provides better therapeutic profiles does not entirely fit with the data set or indeed, the discussion of these data by the authors. I suggest the authors rewrite this such that while dual Mu/Kappa engagement appears to be beneficial there is no definitive causal link with bias (or indeed evidence that it is not important).

We share the reviewer’s concerns, which is generally discussed in response to the Editors’s request at the beginning of this letter. We have toned down our focus on biased signaling as a rationale for the improved therapeutic benefit of MP1207/08 and added that dual engagement of mu as well as kappa receptor in addition to partial agonism may also be necessary. The focus of our manuscript is structure based design leading to biased agonists at KOR, using the crystal structure of MP1104. We computationally designed and synthesized 12 compounds each requiring multi-step synthesis, evaluated signaling independently in two labs and used MD simulations to validate receptor subpocket activity. However, we certainly cannot rule out a role for partial agonism as a mechanism of action., To address the reviewer’s comments, we have in addition to our lead compound MP1207 added MP1208 pharmacology (analgesia, respiration, locomotor effects and CPP/CPA) too in vivo and both drugs attenuated side-effects at highest doses tested possibly because of mixed actions at MOR and KOR.

2) The Introduction quotes bias factors from different studies. No explanation is given in the Introduction text as to what these values are or what they might mean. Values based on methodologies using the Black and Leff operational model developed by Kenakin, Christopoulos and Ehlert to generate log[Tau/KA] are relative values subject to the experimental system used. Different studies will have used different cell backgrounds, different receptor expression levels, different assays/approaches to measure signalling events and even different reference agonists. The comparison of bias factors across different studies is likely not meaningful and such quantitative comparisons should be removed from the Introduction.

As suggested, the literature-based bias factors for all ligands in the Introduction section have been removed.

3) Bias factors are quoted throughout the manuscript but the work up to these values is not provided nor an explanation as to what they mean. Presumably, a bias factor >1 means bias toward cAMP inhibition and < 1 means bias towards arrestin recruitment. I notice that often ligands with a bias factor of < 1 are described as balanced rather than arrestin biased? The authors are using a methodology based on the operational model of agonism. We need to see the various steps that have been used to calculate these values including the Log[tau/KA], deltaLog[tau/KA] (which should be compared with appropriate statistical methods and deltadeltaLog[tau/KA] values). The bias factors are presented without error associated with them so it is impossible to evaluate the significance of such values, and ultimately this means that we cannot assess the robustness of these findings. This information is needed for publication. I would also like to see a more detailed description of the use of this analysis in the Materials and methods rather than the citation of the Kenakin and Christopoulos methodology paper.

Requested changes have been made. The steps involved in calculating bias factors and errors related to bias factors have now been included. Please see Appendix 1—figure 4. Appendix 1—table 10-11 show Log[tau/KA], deltaLog[tau/KA] and deltadeltaLog[tau/KA] values. Only compounds with a measurable arrestin efficacy/potency and those with bias factors>1 are called G protein biased and ones with bias factor<1 are designated as not G-biased or in most cases referred to as compounds showing arrestin recruitment. The word balanced agonist is not used or used only for DAMGO and U50,488h. Note our computational, structural design complemented by ligand synthesis cannot distinguish between arrestin biased ligands and balanced agonists. We admit this is a limitation at our end. The focus of the manuscript is to develop predictive models that give a bias factor>1 or towards preferably G-biased signaling.

Please note there are differences between bias calculated between assays. For instance, MP1601 and IBNtxA are balanced agonists at rKOR (in BRET assay) while being arrestin biased at hKOR(in the cAMP/Tango determinations). In spite of these differences (which could be because of rat vs human receptor or assay used (BRET vs cAMP/Tango) or receptor reserve), the results are predictive towards a model which states that the alignment of the amidophenyl arm towards TM2-TM3 will lead to preference away from G protein signaling.

4) Some of the curve fitting in some of the figure panels needs to be addressed. For example, in Figure 1C the lower asymptote of MP1104 is not defined at all and the concentration-response curve appears to start below zero. How was the lower asymptote of this curve defined? The same issue is true for the IBtxA curve at the hKOR G protein assay (S3C) and MP1202 in Figure 2I. In Appendix 1—figure 1 the maximal response of IBNtxA and MP1202 in the arrestin assay are not defined so it is not clear how this not has been fitted or how EC50 and Emax have been estimated. The same is true for IBNtxA in S3G. Fitting data in this way will likely impact bias calculations too. In some cases (eg. IBtxA) there are some compelling reversals of the order of potency with DAMGO as qualitative indicators of bias but in some cases I am not sure if reliable quantitative estimates of bias can be derived from these data.

Changes have made and lower asymptote and maximal response defined in the legend. New cAMP data for IBNtxA, MP1202 and MP1104 at hKOR have been added and concentration-response curve now start at zero. The arrestin potency of IBNtxA and MP1202 at mMOR is now not reported as maximal response and potency could not be determined and bias factors for these are now not reported. Bias factors are now only reported for compounds with a measurable arrestin potency. As discussed above, compounds with bias factors >1 relative to either DAMGO for MOR or U50488H for KOR are designated as biased while ones with bias factor<1 are designated as not biased.

5) DAMGO is less potent in the arrestin recruitment assay as compared to the cAMP assay indicating the former is more weakly coupled to the MOR and/or less amplified. The same is true for the reference agonist U50,488 at the KOR. In such systems, a partial agonist may give a robust response in the highly coupled endpoint but a weak response or no response in the weakly coupled endpoint. From these data, I would say that the authors cannot definitively call such examples “G protein biased”. This includes MP1207 and MP1208 at the MOR (Figure 3 and Appendix 1—figure 1), MP1209 and 1210 “G protein biased” at the MOR (Figure 4), MP1306 and MP1601 at the mMOR (Appendix 1—figure 1). Indeed, the [35S]GTPgS data for some of these ligands does indicate that these ligands are partial agonists for G protein activation too. The possibility that partial agonism might account for these observations is acknowledged in the Discussion. I recommend that the descriptions of these ligands as G protein biased in the results text is toned down to be consistent with this ambiguity. MP1207 and MP1208 have been assigned bias values (8 and 23) in S2D but, given that no response is observed in the arrestin assay, it is not clear how these bias factors have been derived. The boxes in various figures that define such ligands as “G protein-biased” is at odds with the above ambiguity and should be removed.

Requested changes have been made. Only compounds where measurable arrestin efficacy was observed are characterized as G-biased while others are noted as agonists without measurable β-arrestin2 recruitment. This includes MP1207/08/09/10 at MOR (mouse and human) and MP1305, MP1601 at mMOR. Bias factors for MP1207 and MP1208 at rKOR have been removed and boxes with G-biased/not biased have been removed from all figures throughout the manuscript, We have toned down the discussion of bias and included a discussion on partial agonism for ligands that do not recruit arrestin.

6) The authors should provide further characterization/description of the double mutant is needed in order to interpret their data. Is it efficiently expressed at the cell surface and how does this expression compare to WT? How does the affinity and potency of the reference agonist change at this mutant? Indeed, given that bias is always expressed relative to the action of the reference agonist can the authors be sure that these mutations have altered the bias of MP1207/8 but not U50,488? How does U50,488 bind the receptor in terms of interaction with TM2/ECL2?

This is an excellent question. Although all points raised by the reviewer are fair, the loss of Dr. Pasternak’s lab leaves us unable to carry out any additional experiments to resolve this. In light of the reviewer’s concerns, we have taken out this figure and added an MD simulation, which computationally determines that the polar amine group binds between TM5 and ECL2 region. MD results in combination with MP1209 and MP12010 synthesis and kappa bias results together suggest that MP1207 requires acidic residues in TM5-ECL2 region to mediate its G-biased agonism. We are not able to answer the reviewer’s question of whether the mutation changes cell surface expression and if it affects potency and efficacy of the reference ligand vs MP1207/08. We add that U50,488H does not bind in the TM2/ECL2 region based on our computational studies. Furthermore, we have carried out BRET assays to evaluate whether the efficacy of MP1207 and 1208 is altered by TM5-ECl2 region mutations. We find that the efficacy of MP1207/08 is not altered by E209AD223A. We have retained the Y312W hKOR mutation results because mutant was extensively characterized in our previous publication (PMID: 29307491).

7) The figures legends are excessively long due to the inclusion of potency data for each agonist in each panel. These data should be presented in a table as the authors have done for their DOR data. In the figures more detail is needed regarding the data shown. Is this grouped data or is this an individual experiment? The authors should clarify if the numbers in parentheses are pEC50 plus minus SEM. The authors should provide the number of individual experiments and the number of replicates within each experiment for each data set. This information is available for some but not all data sets.

In order to reduce the length of the figure legend as requested by the reviewer, we have now included all in vitro potency and efficacy data in a table. See Appendix 1—table 7-9. Data shown is grouped in cohorts to increase rigor. We now clarify in the figure legend that the data represented is EC50 ± SEM. Legends of each figure now further include the number of individual replicates for each figure.

8) All in vivo data is shown as % decreases or as %MPE. Baseline values for each group should be given to show that they were not different in both analgesia and respiratory depressant assays. I would strongly encourage including raw data from these experiments as a supplementary table.

We agree this increases rigor. All analgesia and respiration raw data has been added as source data with the manuscript as requested by the reviewer. We did not however add the baseline values to the figures as they become too crowded and busy.

9) Drug administration route: In the in vivo experiments there are instances in which drugs are compared in the same assay but are administered differently. The reason for this needs further explanation. The basis for chosen doses needs further clarification. In the CPA/CPP experiments it is not clear why IP morphine/U50,488h is compared to ICV MP1207 drug administration. Unless there is evidence that this route results in appreciable concentrations of MP1207 in the circulating plasma you are not comparing like with like. It is a significant confound in this experiment when trying to demonstrate improvement over morphine in particular. Routes of administration and factors such as ability to cross the blood-brain barrier will, naturally, determine the measured effects of an opioid. This may be problematic when comparing the effects of opioid administered via different routes. Indeed, given the importance of peripheral opioid receptors for analgesia, tolerance and constipation it is difficult to predict the relative therapeutic benefits of a drug based on ICV data alone. This should be discussed.

Being charged at physiological pH, both MP1207/08 were not expected to be active peripherally and hence were evaluated after icv administration. This is also a well-established route of administration for initial testing of novel compounds, for which the supply is understandably limited. Still, to address the direct comparison requested by the reviewer, icv morphine and U50,488h controls for analgesia, respiration, CPP/CPA are now added wherever appropriate to facilitate direct comparison of the compounds by the same route of administration. See Figures 5 and 6. Notably, our conclusions were not changed by this revision.

We appreciate the reviewer’s concern about tolerance and constipation being dependent on peripheral opioid receptors. However, analgesia (at least in part), respiratory depression, locomotor effects and addiction/aversion potential are well established to be supraspinally mediated. Accordingly, only those side-effects were evaluated here. Indeed, with improved design and possibly structures of these biased drugs, the next generation of MP1207 analogs will be aimed at developing systemically active analogs where other side-effects can be evaluated in great detail and mechanisms of MOR/KOR G/arrestin signaling and partial agonism in vivo will be probed.

10) In measurements of respiratory depression, a dose of 10 mg/kg morphine IP is used. Without a correlate in the other assays it is not clear why this is a useful comparator for the action of ICV MP1207. Similarly, U50,488h is also administered IP in this assay but ICV in the antinociception assay. The 30 nmol and 100 nmol ICV dose of morphine used in respiration are in excess of the maximal concentration of morphine used in the tail-flick assay. 10 nM ICV morphine would seem a more logical concentration to use for comparison with 30 nmol MP1207. This is also true for measurements of locomotor activity.

We had added animals and determined that morphine icv ED_50_ is 4.77 nmol (Figure 5A) which is similar to icv MP1207 ED_50_ 6.1 nmol. Hence determining respiration at 30 and 100 nmol for morphine is appropriate (Figure 6B). We have added icv morphine/U50,488h data for CPP and respiration also (Figure 5B).

11) Data was taken at the point of peak antinociception for each drug, but it is not clear if these drugs peak at the same timepoint. The duration of an analgesic effect is also important information to know. The authors should supply supplementary data showing the time course of analgesia.

Data has been added showing the time course of analgesia. See Figures 5C for MP1207 and Figure 5D for MP1208. We note the peak analgesic effects and peak increase in respiration and locomotor activity vary as discussed above.

[Editors' note: further revisions were suggested prior to acceptance, as described below.]

Revisions:All reviewers agree that the manuscript has been significantly improved. Only minor additional concerns listed below were raised:1) Since these agonists appear to be slightly more partial than buprenorphine (adding this reference agonist was helpful), it would be useful to discuss what is known about buprenorphine in respiratory depression, CPP and hyperlocomotion assays- with the caveat that the buprenorphine metabolite norbuprenorphine may be biologically relevant. This may help strengthen the case that partial KOR agonism is beneficial relative to the weak MOR agonism KOR antagonism of bup.

As requested by the reviewer we have added references for hyperlocomotion, CPP (PMID: 17367825), respiration (PMID: 15833777) and role of norbuprenorphine (PMID: 11303059 and PMID: 22739506) in mediating some of buprenorphine’s effects. We have added a line distinguishing our probes MP1207/08 from buprenorphine. Briefly, buprenorphine causes hyperlocomotion, shows CPP and is known to have lower respiratory depression compared to classical mu opioiod agonists. Buprenorphine also metabolizes to norbuprenorphine further complicating its pharmacology. Norbuprenorphine has lower potency but higher efficacy over buprenorphine and norbuprenorphine shows respiratory depression. MP1207/08 in contrast shows attenuated CPP as well as respiratory depression at least in part because of dual low intrinsic efficacy at both KOR and MOR. This conjunction of KOR partial agonism with MOR partial agonism may have therapeutic benefits over the more broadly active buprenorphine, for example as shown presently with our probes by blunting MOR mediated respiratory depression.

2) It appears there is some weak DOR agonism/ antagonism and that both MP1207/1208 can bind to DOR with only slightly lower affinity than MOR and KOR. DOR antagonism is known to modulate some MOR-mediated behaviors, so there is the potential for DOR agonism to play a role in some of the behaviors reported, which should be acknowledged.

MP1207/08 certainly can label DOR at high doses and it is possible that DOR partial agonism and/or antagonism may impact the overall pharmacology of MP1207/08 in mice. We have added references for mixed MOR-DOR ligands, ligands with MOR agonists-DOR antagonists are known to have less tolerance as well as physical dependence (PMID: 27556704) while ligands with mixed MOR agonism-DOR agonism (PMID: 31201990) are known to be more active in chronic pain models.

3) "Together, these results support that dual MOR and KOR agonism may offset the liabilities characteristic of receptor-selective agonists".

Requested change has been made.

4) The results are consistent with the dual MOR/KOR agonism hypothesis. Still, they might not differentiate weak MOR agonism from MOR/KOR agonism unless they are also done in the KOR knockout mice as was done for analgesia. Therefore, the language should be toned down.

As requested a line stating the above noted limitation has been added in the Discussion section. Briefly, given the uncertainty over a mechanism by which KOR activity blunts MOR-mediated respiratory depression, future respiratory testing of mixed action ligands with MOR KO and KOR KO mice will be needed. Such testing may also better resolve whether weak MOR agonism alone or a combination MOR-KOR dual partial agonism, rather than the functional selectivity of MP1207 and MP1208 would be sufficient to account for the reduced liabilities presently observed.

5) Throughout the manuscript, the authors refer to bias factors. For example, MP1104 is said to have bias factors of 0.58 and 0.15 at MOR and KOR. Bias factors are not absolute values and are relative to the reference agonist used. Readers unfamiliar with bias factors will have no idea what these values mean unless they go to the Materials and methods section and then read a couple of papers cited there. The first time these bias factors are used, the authors need to explain more clearly how they were derived and what exactly they mean. Specifically: A) That at the MOR this is relative to the action of the agonist DAMGO and at the KOR, this is relative to the action of the agonist U50,488h – this is in the Materials and methods but needs to be stated in the text. B) That in all cases in the manuscript, you are comparing G protein signaling with arrestin recruitment so that bias factors greater than 1 means bias towards G protein and less than 1 means bias towards arrestin. C ) Where your values of Log bias are not significantly different from zero (so your bias factors are around 1) you need to state that the ligand is not biased.6) In some cases, the analysis shows that some of the ligands show bias towards arrestin, but this is not highlighted in the text. Instead, it seems to be stated that the ligand shows robust at recruiting arrestin (e.g., MP1104, IBNtxa @ the KOR). This is not the case for G protein biased ligands. Why the difference? It is less confusing if bias is described the same way in both directions.

Thank you for this thoughtful suggestion. Requested changes under items 5 and 6 has been made. All ligands with bias factor >1 are named as G protein biased; ones with bias factor less <1 are arrestin biased while compounds with bias factor lacking statistically significant deviation from 1 are deemed unbiased throughout the manuscript wherever applicable. Definition of how bias factor is calculated in relative terms is now illustrated clearly.

7) The statement that IBNtxa induces minimal arrestin recruitment at the MOR is at odds with the Appendix 1—figure 3, which shows that this ligand can induce a robust level of recruitment to 70% of that of DAMGO – the interesting thing is that the ligand is more potent than DAMGO in the G protein assay but less potent than DAMGO in the arrestin assay.8) Similarly, the statement that MP1202 shows diminished arrestin recruitment is at odds with Appendix 1—figure 1C as the maximum effect of the ligand at arrestin recruitment is not reached at the highest concentration used – indeed, it would appear that MP1202 is less potent in the arrestin assay, but relative to DAMGO, may be more efficacious in this assay as compared to the cAMP assay.

We thank the reviewers for pointing this out. The efficacy for both ligands is lower than DAMGO in arrestin as well as G protein assay. The potency in the G protein assay is greater than DAMGO for both ligands and as pointed out by the reviewers, ligands trend towards lower potency in the arrestin assay relative to DAMGO. While the exact reason for this is not known, we have made changes and used the phrase “trending towards lower arrestin potency” versus G protein potency compared to DAMGO in this case.

Note, we did not calculate the bias factors in this case because we cannot accurately determine the EC50 for ligands at βarrestin2 because the curves don’t show a plateau. Therefore we are not claiming these to be biased in this assay. We do however see a clear preference for G protein pathway with same ligands using cAMP/Tango assays. The bias analysis of all ligands is covered by Appendix 1—figure 4 and calculations of bias which cover efficacy and potency at both signaling pathways is shown in Appendix 1—tables 10-11.

9) Some of the concentration-response curves appear to be biphasic and not well fitted to the concentration-response curve shown in the relevant figure: Figure 3B, 1208; Figure 4D, 1209 and 1210, Figure S7C, 1202; Figure S7F, 1208; maybe Figure S7G, 1208; Figure S9D, 1305. Can the authors provide an explanation for this? Can they rule out some sort of non-specific effect occurring when these ligands are used at very high concentrations? It seems to be something observed in the arrestin assays in particular.

We thank the reviewers for pointing this out. We do not believe this effect is non-specific, because it was not seen for the same compounds in otherwise identical conditions in MOR Tango assays. Furthermore, it is not a general property of the KOR Tango assays as it was also not seen when testing U50,488h. The most plausible explanation for the observed biphasic response could be compounds at high concentrations hitting intracellular receptors, either with a basal pool of internal receptors or with receptors that have been internalized in response to agonist addition. Consistent with this interpretation, we and other groups have shown that a high density of receptors do exist intracellularly (Stoeber et al., 2018; Che et al., 2020). We have now included this discussion in the main text.

10) A bias factor for MP1202 at the mutant Y312W is stated as 34 and this is shown in Figure S7L. The authors should provide the data/calculations in a supplementary table from which these values were derived, similar to those tables S10/11 that they have already provided for ligands at WT ORs.

The requested calculations are shown Appendix 1—table 10. It is possible reviewers had missed seeing it.